# Inhibitory control of frontal metastability sets the temporal signature of cognition

**Vincent Fontanier[1]\*[†], Matthieu Sarazin[2]\*[†], Frederic M Stoll[3], Bruno Delord[2][‡], Emmanuel Procyk[1][‡]**

[1]Univ Lyon, Université Lyon 1, Inserm, Stem Cell and Brain Research Institute U1208, Bron, France; [2]Sorbonne Université, CNRS, Institut des Systèmes Intelligents et de Robotique, ISIR, F-75005, Paris, France; [3]Nash Family Department of Neuroscience and Friedman Brain Institute, Icahn School of Medicine at Mount Sinai, New York, United States

**Abstract** Cortical dynamics are organized over multiple anatomical and temporal scales. The mechanistic origin of the temporal organization and its contribution to cognition remain unknown. Here, we demonstrate the cause of this organization by studying a specific temporal signature (time constant and latency) of neural activity. In monkey frontal areas, recorded during flexible decisions, temporal signatures display specific area-dependent ranges, as well as anatomical and cell-type distributions. Moreover, temporal signatures are functionally adapted to behaviourally relevant timescales. Fine-grained biophysical network models, constrained to account for experimentally observed temporal signatures, reveal that after-hyperpolarization potassium and inhibitory GABA-B conductances critically determine areas' specificity. They mechanistically account for temporal signatures by organizing activity into metastable states, with inhibition controlling state stability and transitions. As predicted by models, state durations non-linearly scale with temporal signatures in monkey, matching behavioural timescales. Thus, local inhibitory-controlled metastability constitutes the dynamical core specifying the temporal organization of cognitive functions in frontal areas.

**\*For correspondence:**
vincent.fontanier@gmail.com
(VF);
matthieu.sarazin@live.fr (MS)

[†]These authors contributed
equally to this work
[‡]These authors also contributed
equally to this work

**Competing interest:** The authors
declare that no competing
interests exist.

**Reviewing Editor:** Timothy E
Behrens, University of Oxford,
United Kingdom

## Editor's evaluation

The paper investigates the temporal signatures of single-neuron activity (the autocorrelation timescale and latency) in two frontal areas, MCC and LPFC. These signatures differ between the two areas and cell classes, and form an anatomical gradient in MCC and, moreover, the intrinsic timescales of single neurons correspond with their coding of behaviorally relevant information on different timescales. The authors develop a detailed biophysical network model which suggests that after-hyperpolarization potassium and inhibitory GABA-B conductances may underpin the potential biophysical mechanism that explains diverse temporal signatures observed in the data. The proposed relationship between the intrinsic timescales, coding of behavioral timescales, and anatomical properties (e.g., the amount of local inhibition) in the two frontal areas is novel, the use of the biophysically detailed model is creative and interesting and the claims are convincingly supported by the data.

## Introduction

Large-scale cortical networks are anatomically organized in hierarchies of interconnected areas, following a core-periphery structure (*Markov et al., 2013*). Within this large-scale organization, the dynamical intrinsic properties of cortical areas seem to also form a hierarchy in the temporal domain (*Chaudhuri et al., 2014*; *Murray et al., 2014*). The temporal hierarchy arises from increasing

timescales of spiking activity from posterior sensory areas to more integrative areas including notably the lateral prefrontal and midcingulate cortex (MCC). Intrinsic areal spiking timescales are defined from single unit activity autocorrelation (*Murray et al., 2014*). Long spiking timescales potentially allow integration over longer durations, which seems crucial in the context of higher cognitive functions, learning, and reward-based decision-making (*Bernacchia et al., 2011*). Recent studies uncovered links between single unit working memory and decision-related activity and spiking timescales in the lateral prefrontal cortex (LPFC) (*Cavanagh et al., 2018*; *Wasmuht et al., 2018*). However, the mechanisms that causally determine the timescale of cortical neuron firings and their role in the functional specificity of areas remain to be described.

To address this question, we recorded in the MCC and LPFC, because these two frontal areas both display particularly long spiking timescales and are functionally implicated in cognitive processes operating over extended timescales. These interconnected regions collaborate in monitoring performance and in integrating the history of outcomes for flexible decisions (*Kennerley et al., 2006*; *Khamassi et al., 2015*; *Kolling et al., 2018*; *Medalla and Barbas, 2009*; *Rothé et al., 2011*; *Seo and Lee, 2007*; *Womelsdorf et al., 2014a*). Recent anatomical and physiological investigations revealed that the cingulate region has relatively higher levels of synaptic inhibition on pyramidal neurons than LPFC, with higher frequency and longer duration of inhibitory synaptic currents (*Medalla et al., 2017*), suggesting that excitatory and inhibitory cell types differentially contribute to the specific dynamics of distinct frontal areas. Moreover, MCC also seems to have a longer spiking timescale than the LPFC (*Cavanagh et al., 2018*; *Murray et al., 2014*).

In this context, we sought to understand the relationship between temporal features of spiking activity, local neural network dynamics, and the computations implemented by frontal neural networks. We focused on whether and how different temporal features play distinct roles in different frontal areas. To this aim, we addressed the following questions: what are the exact differences in the temporal organization of spiking in the LPFC and MCC? How do they relate to the distinct roles of excitation and inhibition? Do they reflect cognitive operations, and can they be adjusted to current task demands? Can they be accounted for by local biophysical circuit specificities? If so, do distinct collective network neurodynamics emerge from such areal biophysical characteristics and what are their functional implications?

We examined the contribution of single unit temporal signatures to dynamical differences between LPFC and MCC in monkeys. After clustering units based on spike shape (putative fast spiking [FS] and regular spiking [RS] units), we computed spike autocorrelograms and their temporal signatures (time

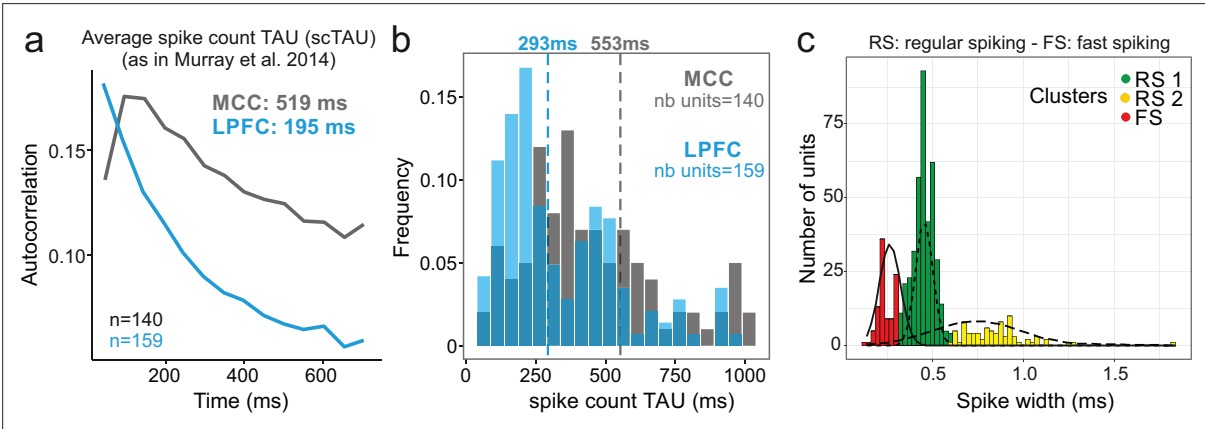

**Figure 1.** Midcingulate cortex (MCC) and lateral prefrontal cortex (LPFC) spike count autocorrelograms. (**a**) Population exponential fit. Autocorrelograms were computed for each unit and the fit was performed on all the units of the MCC (dark grey) and the LPFC (blue) to extract the decay parameter scTAU (as in *Murray et al., 2014*). (**b**) Single unit fits were used to capture individual spiking timescales and produce the distribution of scTAU values for each region. Dotted lines represent the median of scTAU. (**c**) Clustering of spike shape. After extracting the spike width and amplitude from each unit average waveform, we performed a hierarchical clustering revealing the presence of three groups of units (coloured groups RS1, RS2, FS; see Materials and methods). Fitting Gaussian mixed model on the population (lines) confirmed the presence of the three clusters. In the paper, units with narrow spike width were termed as fast spiking (FS), whereas units with broader waveform were marked as regular spiking (RS: RS1 + RS2).

constant and latency). We discovered that LPFC and MCC differed not only in average time constant, but also specifically in the autocorrelogram latency of their RS units.

Regular and FS MCC neurons showed different temporal signatures. Remarkably, through these signatures, neurons contributed to encoding information at different timescales, that is, information relevant between trials or across multiple trials. Exploring constrained biophysical recurrent network models, we identified the ionic after-hyperpolarization potassium (AHP) and inhibitory GABA-B receptor conductances as critical determinants mechanistically accounting for the difference in spiking temporal signatures between LPFC and MCC. The models predicted how differences in temporal signature amounts to the ability of networks to undergo metastable states with different properties. Indeed, we found, in monkey data, long-lasting states in primate MCC activity but not in the LPFC.

Critically, we show that by controlling states stability and transitions, local inhibition – rather than synaptic excitation (*Chaudhuri et al., 2015*) – is the major factor setting temporal signatures. Moreover, inhibitory-mediated temporal signatures did not require specific disinhibition between molecularly identified subnetworks of interneurons but naturally emerged from inhibitory weight variability (*Wang, 2020*).

## Results

We analysed 570 units recorded in MCC and LPFC (298 and 272 units, respectively). Using the autocorrelogram of binned spike counts (see Materials and methods), we were able to extract population spiking timescales for a subset of this population (140 and 159 units, respectively, for MCC and LPFC) and observed population autocorrelograms similar to those obtained with other datasets (*Cavanagh et al., 2018*; *Murray et al., 2014*; *Wasmuht et al., 2018*; *Figure 1a*). At the population level, the characteristic timescale of spiking fluctuation over time, scTAU (the time constant from the exponential fit of the spike count autocorrelogram), was longer for MCC than for LPFC (MCC = 519 ± 168 ms, LPFC = 195 ± 17 ms). In addition, MCC single units exhibited longer individual scTAU than LPFC units (medians, MCC = 553 ms, LPFC = 293 ms; two-sided Wilcoxon signed-rank test on log(scTAU), $W$ = 15,192, p<10$^{-8}$; *Figure 1b*), as in previous datasets (Figure 1c in *Cavanagh et al., 2018*). Aside from being characterized by a slow decay (long scTAU), the MCC population autocorrelation displayed a distinctive feature: a positive slope at the shortest time lags equivalent to a latency in the autocorrelogram, that can be observed in previous publications (see Figure 1c in *Murray et al., 2014*, Figure 1d in *Cavanagh et al., 2018*). However, the method we employed above (derived from *Murray et al., 2014*) cannot resolve the fine dynamics of neuronal activity at short time lags. To improve upon this approach, we developed a method based on the spike autocorrelogram of individual units from all spike times (named spike autocorrelogram below), which provides high temporal precision in parameter estimation and is computed using the spike time series in the entire or subset segments of recordings (see Materials and methods).

One basic assumption to explain local dynamical properties is that interactions between cell types (e.g. pyramidal cells and interneurons) might induce specific dynamics in different areas (*Medalla et al., 2017*; *Wang, 2020*; *Womelsdorf et al., 2014b*). To separate putative cell populations in extracellular recordings, we clustered them using single unit waveform characteristics (*Nowak et al., 2003*). Although associating spike shapes to cell types is not a fully reliable methods for cell-type identification (*Vigneswaran et al., 2011*), several studies have shown that on population data different cell types and coding properties can be clustered in this way (*Krimer et al., 2005*; *Trainito et al., 2019*). Clustering our dataset discriminated three populations, with short, large, and very large spikes (*Figure 1c*). The results below were obtained using two clusters (small, and large + very large), as detailed analyses showed no clear difference between large and very large spike populations (see *Figure 2—figure supplement 1*). We classified units as FS (short spikes; $n_{MCC}$ = 41, $n_{LPFC}$ = 57 units) or RS (long spikes; $n_{MCC}$ = 257, $n_{LPFC}$ = 215 units) which, in previous studies, were associated to putative interneurons and pyramidal cells, respectively. In the rest of the paper, and especially for the purpose of modelling, we thus assume simplistically an equivalence between FS vs. RS and interneurons vs. excitatory neurons.

## MCC temporal signatures differ for RS units

From spike autocorrelograms we extracted multiple metrics, namely the peak latency (LAT; the time lag of the peak of the autocorrelogram) and time constant (TAU) (see Materials and methods). Together, TAU and LAT constituted the temporal signature of single neurons spiking dynamic. The success rate of fitting an exponential function on spike autocorrelograms using the whole recordings was 91.4% and largely outperformed the alternative method (see Materials and methods). *Figure 2a* shows comparative examples. All subsequent analyses of this study were performed on this pool of units ($n_{MCC-FS}$=39, $n_{MCC-RS}$=225, $n_{LPFC-FS}$=55, $n_{LPFC-RS}$=202). Note that because of the methodological criteria on spike numbers required for good fitting, the sample size of units can change depending on the analysis, especially when restricting recordings to specific time periods. Note also that in the pool of neurons where TAU was successfully extracted using both methods (*n*=280, see Materials and methods for criteria), we found a correlation between the two measures (Murray methods – scTAU – vs. spike autocorrelograms; Spearman's correlation: rho(282) = 0.46, p<$10^{-15}$) although scTAU were overall larger, as observed by another recent study using a different method (*Spitmaan et al., 2020*). Importantly, TAU was not correlated with firing rate across units (*Figure 2b*, *Figure 2—figure supplement 2*).

TAU was higher on average in MCC than in LPFC for both RS and FS cells (medians ± sd: MCC FS = 284.7 ± 132 ms, RS = 319.5 ± 199 ms, LPFC FS = 175.1 ± 67 ms, RS = 191.6 ± 116 ms; linear model fit on Blom transformed TAU for normality, TAU = area * unit type, area: $F$(1,520)=18.36, p<$10^{-4}$, unit type: $F$(1,520)=2.72, p=0.12, interaction: $F$(1,520)=0.19, p=0.79) (*Figure 2c*; individual monkey data in *Figure 2—figure supplement 3*).

Additionally, our new approach allowed us to extract LAT, which captures other aspects of neurons temporal dynamics. Importantly, it differed significantly between MCC and LPFC for RS but not for FS units, with MCC RS units having particularly long latencies (median ± sd: MCC FS = 48.5 ± 30 ms, RS = 108.7 ± 64 ms, LPFC FS = 48.5 ± 35 ms, RS = 51.9 ± 46 ms; linear model fit on Blom transformed LAT for normality, LAT = area * unit type, interaction: $F$(1,520) = 11.81, p<0.005) (*Figure 2c*).

TAU and LAT both reflect temporal dynamics, but those measures were significantly correlated only in LPFC RS units (Spearman's correlations with Bonferroni correction, only significant in LPFC RS: rho(203) = 0.29, p<$10^{-3}$). The absence of correlation suggested TAU and LAT likely reflect different properties of cortical dynamics. Moreover, the data also suggested that the different temporal signatures of RS units could reflect differences in the physiology and/or local circuitry determining the intrinsic dynamical properties of MCC and LPFC.

## MCC temporal signatures are modulated by current behavioural state

A wide range of temporal signatures might reflect a basic feature of distributed neural processing (*Bernacchia et al., 2011*). But do different temporal signatures play distinct roles in terms of neural processing in different areas? And, are these signatures implicated differentially, depending on task demands? These questions are unresolved, although recent studies suggest a lack of relationship between individual neuron timescale and selectivity to task-relevant signals (*Spitmaan et al., 2020*). As single units were recorded while monkeys performed a decision-making task (described in *Stoll et al., 2016*; *Figure 3a*), we extracted each unit's temporal signature separately for periods in which monkeys were either engaged in the cognitive task or were pausing from performing the task (*Figure 3b*). TAU extracted during engage and pause periods were significantly correlated across neural populations (MCC FS *n*=19, LPFC FS *n*=21, MCC RS *n*=80, LPFC RS *n*=97, Pearson correlation: *r*(215)=0.20, p=3.0e-3), indicating that TAU reflects stable temporal properties across conditions (corrected from time-on-task, see Materials and methods). The MCC RS population exhibited a significant modulation of TAU, expressing longer TAU during engage periods compared to pause periods, suggesting that engagement in cognitive performance was accompanied by a lengthening of temporal dynamics for RS neurons in MCC (*Figure 3c* top) (Wilcoxon signed-rank test with Bonferroni correction, only significant for MCC RS: MD = 1.06, *V*=2467, p=3.9e-7). To control for a time-on-task effect on such timescale modulation, we contrasted pause periods with engaged periods that occurred at similar times within sessions (i.e. considering only engaged periods occurring after the first pause – see limits in *Figure 3b*, red marks) ($n_{MCCFS}$ = 19, $n_{LPFCFS}$ = 21, $n_{MCCRS}$ = 80, $n_{LPFCRS}$ = 97, Wilcoxon signed-rank test with Bonferroni correction, only significant for MCC RS: MD = 1.06, *V*=2467, p=3.9e-7).

We observed no significant variation of LAT with task demands.

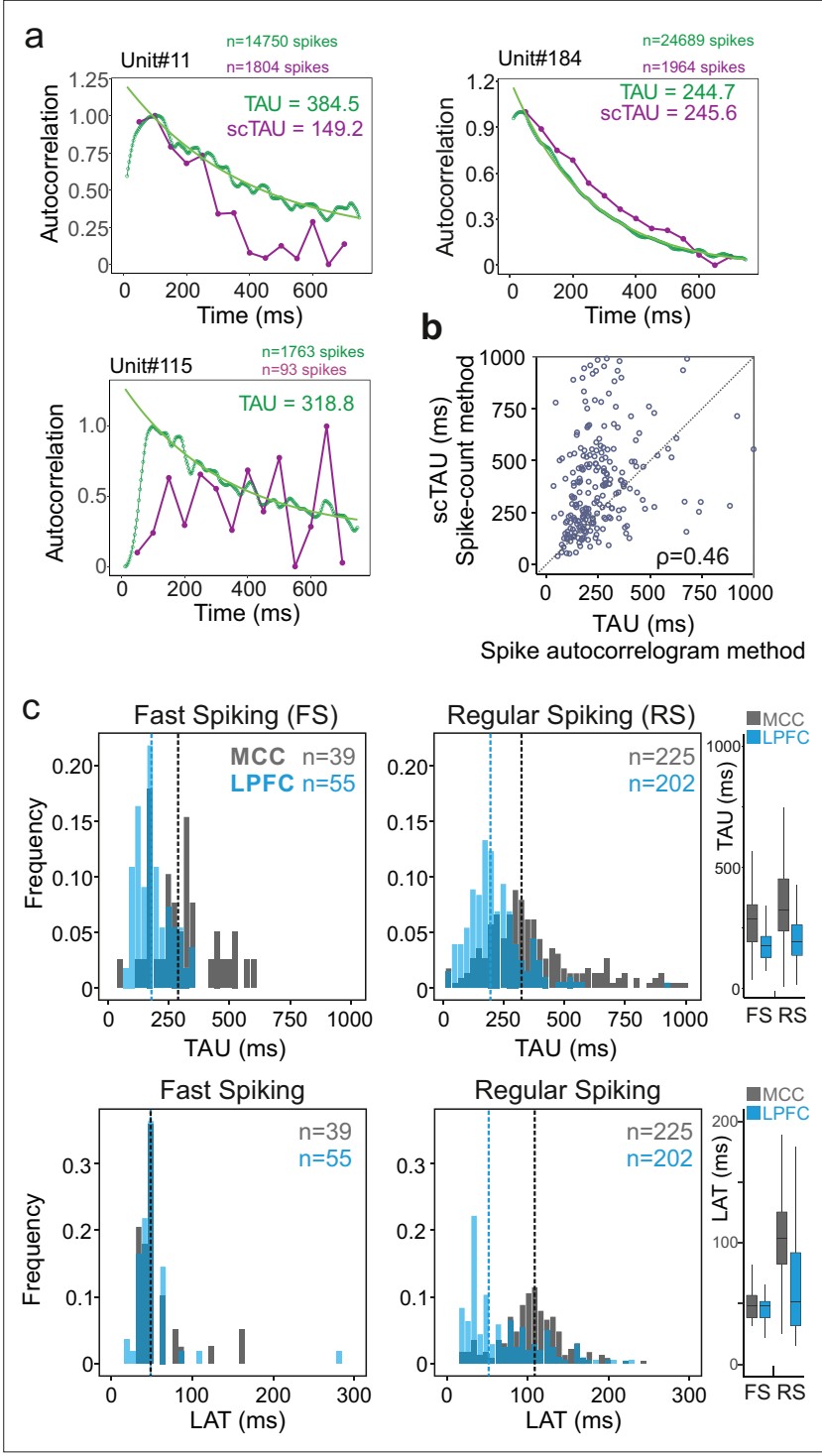

**Figure 2.** Spike autocorrelogram and temporal signatures in midcingulate cortex (MCC) and lateral prefrontal cortex (LPFC). (**a**) Three single examples of spike count (purple, scTAU) vs. normalized spike autocorrelograms (green) contrasting the outcome of the two methods. The measured time constant (TAU) is indicated for both when possible. Numbers of spikes used for each method is also indicated. (**b**) TAU values extracted from each methods are significantly correlated (*n*=280, Spearman's rho(282) = 0.46, p<10⁻¹⁵). (**c**) Distributions of TAUs (upper histograms) and peak latencies (LAT – lower histogram) for fast spiking (FS) (left) and regular spiking (RS) (right) units. '*n*' indicates the number of units. Vertical dashed lines indicate medians of respective populations. Boxplots on the right show the respective population data. TAU values were longer in MCC (dark grey) than in LPFC (blue) for both FS and RS (linear model fit on BLOM transformed TAU for normality, TAU = region * unit type, region:

*Figure 2 continued on next page*

*Figure 2 continued*

$t$=−4.68, p<10$^{-6}$, unit type: ns, interaction: ns). Peak latencies significantly differed between MCC and LPFC for RS but not for FS units (medians: MCC FS = 48.5 ms, RS = 102.0 ms, LPFC FS = 48.5 ms, RS = 51.8 ms; linear model fit on BLOM transformed latency for normality, latency = region * unit type, interaction: $t$-value=−3.57, p<10$^{-3}$).

The online version of this article includes the following figure supplement(s) for figure 2:

**Figure supplement 1.** Spike autocorrelogram features considering the three clusters of populations (fast spiking [FS], regular spiking 1 [RS1] and RS2) in the midcingulate cortex (MCC) and lateral prefrontal cortex (LPFC).

**Figure supplement 2.** Relationship between firing rate and temporal signatures.

**Figure supplement 3.** Average TAU for each cell type, area, and for each animal separately.

## Temporal signatures are linked to cognitive processing

Contrary to MCC, LPFC temporal signatures were not modulated by engagement in the task. Multiple cognitive models propose a functional dissociation between MCC and LPFC and indeed empirical data reveal their relative contribution to feedback processing, shifting, and decision-making (*Khamassi et al., 2015*; *Kolling et al., 2018*; *Stoll et al., 2016*). One important question is thus whether temporal signatures observed for a given area and/or cell type contribute to selected aspects of cognitive processing. For example, temporal signatures might be adjusted to the current functional context and timescale required to perform a task. In our experiment monkeys gained rewards by performing trials correctly in a categorization task while each success also brought them closer to obtaining a bonus reward (*Figure 3a*, right panel, see Materials and methods for task description). By touching a specific lever at trial start, animals could either enter a categorization trial or check the status of a visual gauge indicating the proximity of the bonus reward availability. The number of rewards (i.e. correct categorization trials) needed to get the bonus, and thus the speed of the gauge increase, varied across blocks (i.e. either fast or slow). Previous analyses revealed that feedback influenced the likelihood of checking

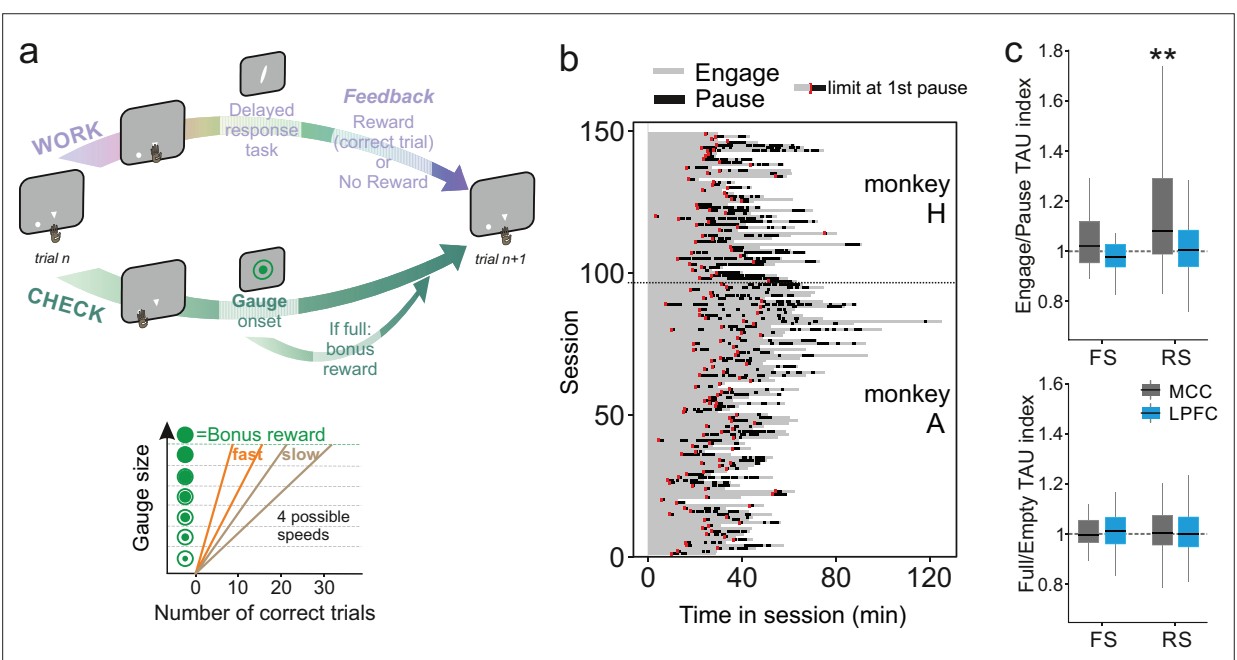

**Figure 3.** Behavioural engagement in task and spiking timescale changes. (**a**) Schematic representation of the task. At the start of each trial, animals can either initiate a delayed response task (WORK option) which can lead to one reward delivery, or use the CHECK option to check the current size of the gauge (or collect the bonus reward). Each reward in the task contributes to increase the gauge size and bring the bonus availability closer. The graph (bottom) schematized the speed of increase of the gauge size which varies between blocks (fast or slow blocks). (**b**) Distribution of pauses in sessions. Each line represents the time course of behaviour for one session for monkey H and monkey A. Grey zones represent engagement in the task, black zones represent pauses in work. Red marks indicate the start limit of the first pause in session which defined the beginning of the period taken for control analyses. (**c**) Boxplots of indices for each unit type and region calculated to estimate potential changes in TAU between engage and pause (top), and between empty and full gauge (bottom). TAUs increased in engage vs. pause only for midcingulate cortex (MCC) regular spiking (RS) units.

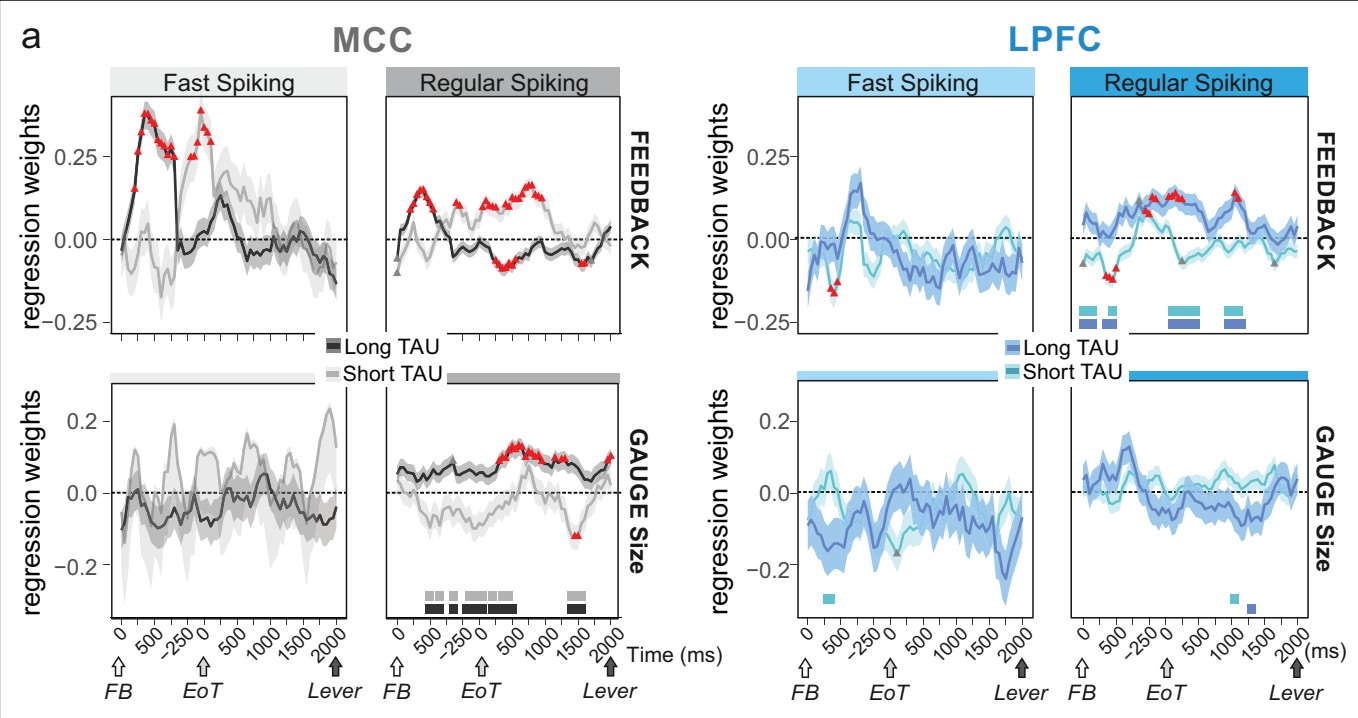

**Figure 4.** Encoding of feedback and gauge size for different unit types and spiking timescales and rostro-caudal distribution. (**a**) Regression weights ($\beta$-coefficients) for the midcingulate cortex (MCC) (grey) and lateral prefrontal cortex (LPFC) (blue) unit populations obtained from time-resolved *glmm* for feedback (reward vs. no reward; top graphs) and gauge size (bottom) (see group analyses using *glmm'* in Materials and methods). Regression weights are obtained at successive time points covering the entire intertrial period between feedback onset and the lever onset in the following trial. Significant effects are indicated by a red triangle (p<0.05 corrected) when more than two successive bins are concerned, shadings indicate standard deviations. Positive values depict a population activity bias towards negative feedback (top) and positive slope of linear coding for gauge size (bottom). Data are presented for fast spiking (FS) and regular spiking (RS) units (left and right respectively for each panel) and have been obtained on subpopulations with short or long TAU values (determined by a median split). Short and long TAU populations are represented by light and dark colour intensity, respectively. Thick bars above the x-axes indicate significance of the coloured corresponding data compared to a null distribution generated through permutations of median split unit identity. Note in particular the dissociation for RS MCC units with short and long TAU respectively coding for feedback and gauge size.

The online version of this article includes the following figure supplement(s) for figure 4:

**Figure supplement 1.** Time-resolved single unit analyses of gauge size and feedback valence encoding between the feedback time (0) and lever onset.

**Figure supplement 2.** Anatomical heterogeneity of TAU.

in the following trial (***Stoll et al., 2016***). Thus, feedback can be considered as information used on a short timescale (within the intertrial period). The animals also built an estimation of the gauge size that was updated upon checking in order to regulate the frequency of checks during blocks, allowing animals to seek and collect the bonus in a cost-efficient manner (***Stoll et al., 2016***). Gauge size can thus be considered as information used and carried over long timescales.

We first hypothesized that blocks of different speeds and/or gauge encoding could engage neurons and modulate their spiking timescale. This was not the case. TAU values were not significantly modulated depending on the state of the gauge (less vs. more than half full, ***Figure 3c***, bottom), nor related to different speeds (Wilcoxon signed-rank test, median = 1) with Bonferroni correction, for gauge state and gauge speed, all (p>0.6).

Conversely, we assessed whether temporal signatures observed for certain cell types contributed to code-specific aspects of the task. We used mixed effect models on groups of single units to test the contribution of population activity to encoding task-relevant information: feedback in categorization trials (i.e. reward vs. no-reward), and gauge size. The rationale was that feedback information was relevant within the intertrial period, whereas gauge information was relevant across trials between two successive checks. Previous analyses had revealed that both MCC and LPFC units encode such information, although MCC units showed greater contributions (***Stoll et al., 2016***). We used data from

the whole recordings (all periods) and classified both FS and RS units as either short *or* long TAU units using a median split. We used a time-resolved generalized mixed linear models (*glmm*; *Figure 4a*) to reveal notable dissociations between these populations that we complemented using a more classical approach at the single unit level, using Poisson *glm* and weighted proportion of variance explained (wPEV; *Figure 4—figure supplement 1*).

Early phases of feedback encoding recruited MCC long TAU populations for both FS and RS units (*Figure 4a*, upper graphs). This discrepancy was confirmed by a difference between early feedback coding in short and long TAU population at the single unit level (*Figure 4—figure supplement 1*). Interestingly FS units in the MCC were mostly engaged in the first second after feedback onset, with a strong bias towards encoding negative feedback (*Figure 4a*, upper left, positive estimates). Effects were more transient and involved short TAU units in the LPFC (*Figure 4a*).

During the intertrial interval, feedback valence was represented in different directions between short and long TAU RS populations in MCC, coding being positive for short TAU populations (higher activity for incorrect feedback) and becoming negative for long TAU populations (higher activity for correct feedback). Conversely, solely the population of MCC long TAU RS units coded for the gauge during the intertrial period (*Figure 4a*, lower graphs). Single unit analyses confirmed the higher contribution of the long TAU population to gauge encoding (*Figure 4—figure supplement 1*).

## Spiking timescales are anatomically organized in MCC

Spiking timescales measured in MCC and LPFC covered several orders of magnitudes (10–1000 ms; *Figure 2c*). Because single unit recordings spanned large regions, such wide range could reflect anatomical organization of segregated populations with distinct homogeneous intrinsic properties. Such organization has been observed in MCC with human fMRI (*Meder et al., 2017*). We indeed found that average TAU values in MCC were higher in more posterior parts, in particular for RS units (ANOVA on Blom transformed TAU: MCC, monkey A: $F(5,112)=2.8$, p=0.041, monkey H: $F(5,54)=3.09$, p=0.033; linear regression on Blom transformed TAU: MCC, monkey A: $t(1,112)=8.99$, p=0.0067, monkey H: $t(1,54)=2.22$, p=0.28; all p-values are FDR corrected for $n=2$ comparison per monkey) (*Figure 4—figure supplement 2a*). This suggests an antero-posterior gradient or heterogeneity of spiking timescales. No such effect was observed in our LPFC data (ANOVA on Blom transformed TAU: LPFC, monkey A: $F(6,110)=0.34$, p=1, monkey H: $F(6,64)=2.49$, p=0.066; linear regression on Blom transformed TAU: LPFC, monkey A: $t(1,110)=1.09$, p=0.60, monkey H: $t(1,64)=0.25$, p=1; all p-values are FDR corrected for $n=2$ comparison per monkey). Note that the so-called LPFC data covered several subparts of posterior LPFC (see *Stoll et al., 2016*). Similar analyses for LAT revealed no consistent heterogeneity within MCC or LPFC (*Figure 4—figure supplement 2b*).

The consequence of such an organization, knowing the respective functional involvement of units with long and short TAU (*Figure 4a*), should be an antero-posterior functional gradient. We tested this by separating MCC cells in posterior vs. anterior subgroups and tested their contribution to feedback and gauge encoding (*Figure 4—figure supplement 2c*). Indeed, posterior RS units activity contributed to positive encoding of gauge size, preceded in time by encoding of positive feedback (negative estimates) (*Figure 4—figure supplement 2c*, lower and upper right), while anterior RS units showed primarily a contribution to feedback encoding (upper right). Finally, anterior FS units were primarily (in time and in strength) contributing to encoding negative feedback. This remarkable contribution of FS to feedback encoding is studied and discussed further below.

In summary, MCC RS units with relatively short or long TAU contributed to the encoding of task elements relevant over short and long terms, respectively. The spiking timescales seemed to be organized along the rostro-caudal axis in MCC. This suggests a correspondence between cell type, temporal signatures, and their functional involvement in processing specific aspects of cognitive information in different functional subdivisions of cortical regions. The crucial questions thus remain of the mechanistic origin of temporal signatures and of how they relate to cognitive functions.

## Biophysical determinants of temporal signatures in frontal network models

To uncover the source and consequences of distinct temporal spiking signatures in the LPFC and MCC, we designed a fine-grained model of local recurrent frontal networks. This model is unique in combining (1) highly detailed biophysical constraints on multiple ionic channels, synaptic receptors,

and architectural frontal specificities, and (2) the cardinal realistic features of mammals cortical neuro-dynamics including the excitation/inhibition balance, high-conductance state of neuronal activity, and asynchronous irregular regime characterizing the awake state (*Brunel, 2000*; *Destexhe et al., 2003*; *Hennequin et al., 2017*). Our specific goal was to evaluate whether biophysical circuit specificities could mechanistically account for differences in LPFC and MCC temporal signatures. We also assessed whether these specificities induce distinct collective network neurodynamics and functional impli-cations, possibly explaining the empirical relationships between temporal signatures, cell type, and information processing. Note that for modelling purposes we equate FS units to GABAergic interneu-rons and RS units to excitatory neurons while acknowledging that it is a crude simplification.

We first explored, using Hodgkin-Huxley cellular models (see Materials and methods), whether specific frontal temporal signatures may arise from ionic or synaptic properties of individual neurons. Extensive explorations of these models identified the maximal cationic non-specific conductance ($g_{CAN}$) and potassium after-hyperpolarization conductance ($g_{AHP}$) as the sole couple affecting both LAT and TAU (*Figure 5—figure supplement 1a-b*). By contrast, conductance couples setting spiking adaptation, post-inhibitory rebound, and slow synaptic transmission were ineffective in changing LAT and TAU (*Figure 5—figure supplement 2*). However, we could not find any region of the $g_{CAN}$ and $g_{AHP}$ parameter space that yielded reasonable values for both LAT and TAU (*Figure 5—figure supplement 1b*). Therefore, the temporal signature of the monkey dataset (*Figure 5—figure supplement 1c*) was poorly reproduced by the cellular model (*Figure 5—figure supplement 1d*). Thus, we then assessed whether collective dynamics at the level of recurrent networks models could better account for frontal temporal signatures (*Figure 5a–b*, see Materials and methods). One-dimensional explorations of the large parameter space failed to identify single biophysical determinants accounting, alone, for differ-ences between monkey LPFC and MCC (RS and FS) temporal signatures (*Figure 5—figure supple-ment 3*; *Figure 5—source data 1*). However, these explorations targeted four parameters of interest regulating either LAT or TAU confirming those already revealed in cellular models ($g_{CAN}$ and $g_{AHP}$) and uncovering, in addition, NMDA and GABA-B maximal conductance ($g_{NMDA}$ and $g_{GABA-B}$) whose slow time constants strongly affected network dynamics.

Two-dimensional explorations using these key parameters (*Figure 5* and *Figure 6—figure supple-ment 1*) identified a single specific setup which demonstrated network dynamics that reproduced the shift from the LPFC-like temporal signature to that resembling the MCC with striking precision. An increase of both $g_{AHP}$ and $g_{GABA-B}$, in the presence of gCAN, drove the model from an LPFC-like temporal signature (LPFCm) (*Figure 5c–d*; map and contours: bivariate probability density model and monkeys' distributions, respectively) towards that of the MCC (MCCm, *Figure 5e–f*). Specifically, $g_{AHP}$ increased LAT and decreased TAU in excitatory (possibly equivalent to RS) neurons (*Figure 6a* left) and had no effect in inhibitory (putatively FS) neurons (*Figure 6a*, right). Besides, $g_{GABA-B}$ decreased LAT in both excitatory and inhibitory neurons (*Figure 6a*, top) and increased TAU in an intermediate range (*Figure 6a*, bottom). A bivariate similarity measure of probability density (see Materials and methods) revealed that monkey temporal signatures were robustly reproduced by the model in two large contiguous regions in the ($g_{AHP}$, $g_{GABA-B}$) space (from which best fits were drawn), with both conductances increased in the MCCm compared to LPFCm (*Figure 6b*).

Several lines of evidence further indicated the model's relevance. First, spiking statistics were similar to those of monkeys (*Figure 7—source data 1*). Then, the model properly accounted for the larger LAT variability in monkey RS vs. FS units (*Figure 5*). Moreover, it reproduced the complex relations between LAT and first-order latency (inter-spike interval [ISI] distribution latency) remarkably well, and in all populations (*Figure 6c* and *Figure 6—figure supplement 2*). Furthermore, both the firing frequency and input-output gain were lower in MCCm excitatory neurons (*Figure 6d*), because of its higher $g_{AHP}$ (*Naudé et al., 2012*), as found experimentally (*Medalla et al., 2017*).

## Metastable states underlie LPFC and MCC temporal signatures

The asynchronous irregular (presumably chaotic) dynamics of network models was highly structured in time (*Figure 5b*). Hidden Markov models (HMMs) revealed that it organized through collective transitions between the so-called metastable (quasi-stationary) states in model neural populations (*Figure 7a*) or pseudo-populations (*Figure 7—figure supplement 1*; see Materials and methods) in the LPFCm and MCCm, as found in frontal areas (*Abeles et al., 1995*; *Seidemann et al., 1996*; *Xydas et al., 2011*). Moreover, while LPFCm states maximally lasted a few hundred milliseconds (*Figure 7b*,

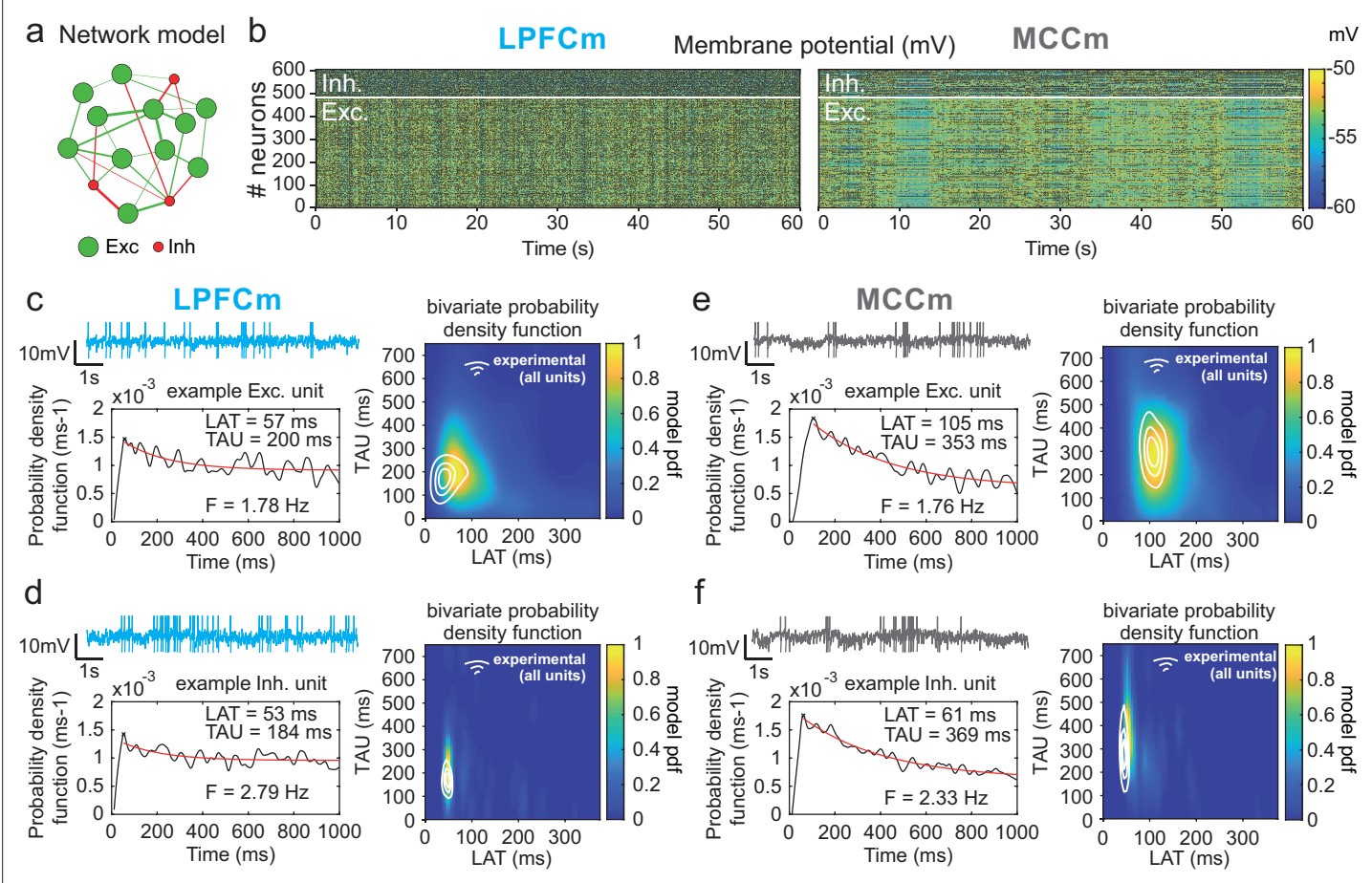

**Figure 5.** Temporal signature of LPFCm and MCCm recurrent network biophysical models. (**a**) Scheme of the frontal recurrent networks modelled, with 80% excitatory (green) and 20% inhibitory (red) neurons and sparsity of synaptic connections. (**b**) Membrane potential in the 484 excitatory (lower part) and 121 inhibitory (upper part) neurons of LPFC and MCC example network models (respectively LPFCm and MCCm ; 'm' for model) with parameter set to approximate LPFC dynamics ($g_{CAN}$ = 0.025 mS·cm$^{-2}$, $g_{AHP}$ = 0.022 mS·cm$^{-2}$, $g_{GABA-B}$=0.0035 mS·cm$^{-2}$; see text and legend of *Figure 6b* for the choice of LPFCm and MCCm standard $g_{AHP}$ and $g_{GABA-B}$ maximal conductances) and MCC dynamics ($g_{CAN}$ = 0.025 mS·cm$^{-2}$, $g_{AHP}$ = 0.087 mS·cm$^{-2}$, $g_{GABA-B}$=0.0143 mS·cm$^{-2}$). (**c**) (Upper left) Membrane potential of an example excitatory neuron of LPFCm. Scaling bars 1 s and 10 mV (spikes truncated). (Lower left) Autocorrelogram of this LPFCm example excitatory neuron (black) and its exponential fit (red, see Materials and methods). (Right) Bivariate probability density distribution of autocorrelogram parameters in LPFCm excitatory neurons. Contour lines at 50%, 75%, and 90% of the maximum of the bivariate probability density distribution in LPFC monkey regular spiking (RS) units. (**d**) Same as (**c**) for LPFCm inhibitory neurons, with contour lines from the bivariate probability density distribution in LPFC monkey fast spiking (FS) units. (**e,f**) Same as (**c,d**), for the MCCm and MCC.

The online version of this article includes the following source data and figure supplement(s) for figure 5:

**Source data 1.** Summary of the effects of the main parameters determining TAU and LAT in the network model.

**Figure supplement 1.** Temporal signature in the pyramidal biophysical neuron model.

**Figure supplement 2.** Temporal signature in the pyramidal neuron model as a function of adaptation and rebound intrinsic and slow synaptic conductances.

**Figure supplement 3.** One-dimensional explorations of key parameters determining TAU and LAT in the network model.

left, blue), MCCm states persisted up to several seconds (*Figure 7b*, grey). This suggested that such a difference in metastability may also parallel the difference of temporal signature in monkey LPFC and MCC areas. Applying HMM to neural pseudo-populations built from experimental data revealed that, as predicted by the model, neural activity was organized as metastable states at slower timescales in the MCC (vs. the LPFC, *Figure 7b*, right). State durations were globally shorter in models (compared to monkeys), as they contained neither temporal task structure nor learning (see Discussion) and were not optimized to fit data.

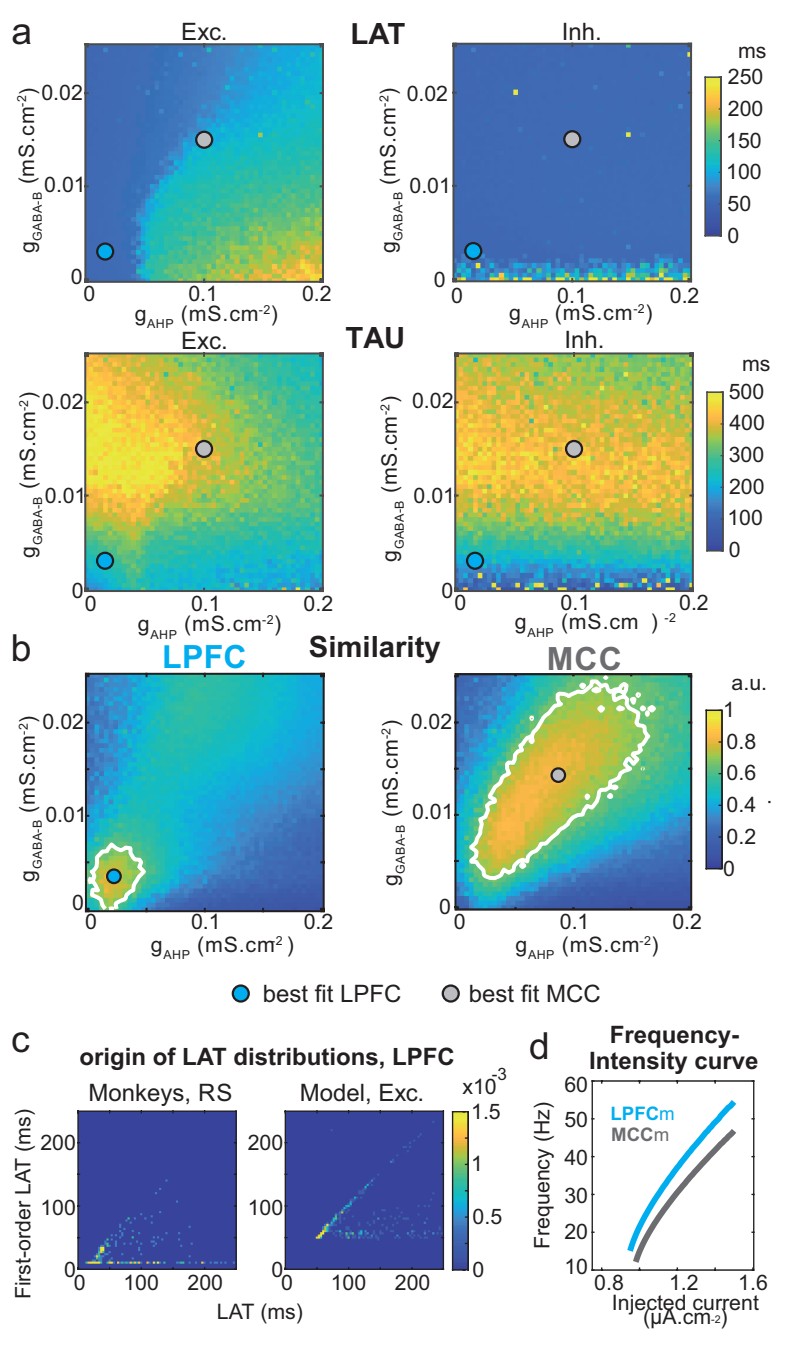

**Figure 6.** Similarity to monkey lateral prefrontal cortex (LPFC) and midcingulate cortex (MCC) temporal signatures critically depends on AHP and GABA$_B$ conductance in the network model. (**a**) Mean population LAT (top) and TAU (bottom) in Exc (left) and Inh (right) neurons, as a function of AHP and GABA-B maximal conductances. Blue and grey dots indicate the ($g_{AHP}$, $g_{GABA-B}$) parameter values of the best fits for LPFCm and MCCm, respectively. (**b**) Similarity of the temporal signature between the network model and monkey data in the LPFC (left) and MCC (right), as a function of AHP and GABA-B maximal conductances (see Materials and methods). In (**a**) and (**b**), the value for each ($g_{AHP}$, $g_{GABA-B}$) is averaged over five simulations. Contour line at 80% of maximum similarity. LPFCm and MCCm ($g_{AHP}$, $g_{GABA-B}$) parameter values calculated as coordinates of the contour delimited area's weighted average. (**c**) Bivariate probability density distribution of the autocorrelogram LAT and first-order latency (the latency of the inter-spike interval [ISI] distribution) in regular spiking (RS) units in monkey LPFC (left) and excitatory neurons in the example LPFCm (right). The model accounts for two distinct neuronal subsets in RS neurons, where LAT is determined by first-order latency solely (due to $g_{AHP}$-mediated refractoriness; diagonal band), or in conjunction with other factors ($g_{GABA-B}$ slow dynamics-mediated burstiness and recurrent synaptic weight variability; horizontal

*Figure 6 continued on next page*

*Figure 6 continued*

band). (**d**) Single excitatory neuron frequency/intensity relationship in LPFCm (blue) and MCCm (grey) in response to a constant injected current.

The online version of this article includes the following figure supplement(s) for figure 6:

**Figure supplement 1.** Two-dimensional explorations in ($g_{AHP}$, $g_{CAN}$) and ($g_{AHP}$, $g_{NMDA}$) spaces.

**Figure supplement 2.** Relationship between autocorrelogram latency and first-order (inter-spike interval [ISI]) latency in lateral prefrontal cortex (LPFC) fast spiking (FS) units/inhibitory (Inh) neurons, and in midcingulate cortex (MCC) regular spiking (RS) units/excitatory (Exc) and MCC FS units/Inh neurons.

---

Long states essentially required high $g_{GABA-B}$ in the MCCm, as they disappeared when $g_{GABA-B}$ was lowered to its LPFCm value (MCCm$_{LPFC\ GABA-B}$ model, *Figure 7b* left, orange curve). In contrast, they only marginally depended on $g_{AHP}$. MCCm and an MCCm with the $g_{AHP}$ derived from that of LPFCm (MCCm$_{LPFC\ AHP}$) showed state duration distributions that were essentially similar, although there was a small increase in the probability of short states at lower $g_{AHP}$ (pink vs. grey curves). In the ($g_{AHP}$, $g_{GABA-B}$) space, $g_{GABA-B}$ systematically proved to be essential in increasing the duration of states, with a border region that clearly separated short states (<0.1 s) from longer states (>1 s) (*Figure 7c*) At this intermediate border, lower $g_{AHP}$ increased the probability of short states (grey vs. pink dots; distributions were even bimodal at lowest $g_{AHP}$ values, not shown), as witnessed by departure from log-normality (*Figure 7c*). As such, the temporal structure of states in the LPFCm was dominated by short and unimodal state duration distributions (*Figure 7c and d*, blue dots), as in monkeys (*Figure 7b*, right) and previous studies (*Abeles et al., 1995*; *Seidemann et al., 1996*). In the MCCm, by contrast, the distribution displayed large durations and a slight departure from log-normality (*Figure 7c and d*, grey dots), resulting in a majority of long states (>1 s) coexisting with short states, as found in data (*Figure 7b*).

State duration, that is, stability, scaled with spatial separation in the neural space of activity (*Figure 7e*, see Materials and methods). Indeed, the shorter states of network models with lower $g_{GABA-B}$ (LPFCm and MCCm$_{LPFC\ GABAB}$, blue and orange dots) were less distant, compared to those of networks models with higher $g_{GABA-B}$ (MCCm and MCCm$_{LPFC\ AHP}$, grey and pink dots). While states were largely intermingled in the LPFCm and MCCm$_{LPFC\ GABAB}$ (*Figure 7f*, upper and middle left), they clearly segregated in the MCC and MCCm$_{LPFC\ AHP}$ (*Figure 7f*, upper and middle right). As predicted by the model, segregation between states was indeed higher in the monkey MCC (*Figure 7e*, large grey triangle, and *Figure 7f*, lower right), compared to the LPFC (*Figure 7e*, large blue triangle, and *Figure 7f*, lower left). This suggests that the higher stability of states in monkey MCC arose from a larger segregation of representations in the space of neural activity.

Altogether, these results suggested that itinerancy between metastable states constitutes a core neurodynamical principle underlying the diversity of computational processes and functions operated in primate frontal areas (*Figure 7g*, see Discussion). From this perspective, the conditions governing transitions between states is critical. We thus evaluated how perturbations of selective neuronal populations would escape ongoing states and reach specified target states (*Figure 7h*). In the MCCm, we substituted the membrane potentials and synaptic opening probabilities of a fraction of excitatory (vs. inhibitory) neurons of the ongoing HMM state by those of a target state. This could mimic the effect of internal chaotic fluctuations or external inputs aimed at reaching that target state. Surprisingly, escaping the ongoing state or reaching the target state remained quite unlikely when substituting excitatory neurons, whatever the fraction (*Figure 7h*, left). By contrast, both probabilities of escaping and reaching scaled with the fraction of substituted inhibitory neurons, with high maximal probabilities (mean: 0.89 and 0.59 for escaping and reaching, respectively – *Figure 7h*, right panel). Interestingly, the probability of escaping a state could attained 0.24 even with as few as 2% of substituted inhibitory neurons, indicating the significant impact of single inhibitory neurons on state itineracy.

Thus, inhibition is a major factor controlling targeted transitions between metastable states in the MCC network model and is also crucial in determining their stability. Excitation had no such role. This result is remarkable, especially considering that MCC FS neurons encoded negative outcomes immediately after feedback onset that triggered behavioural adaptive responses (*Figure 4*). This could reflect the involvement of MCC FS neurons in inducing state changes on feedback associated to behavioural flexibility.

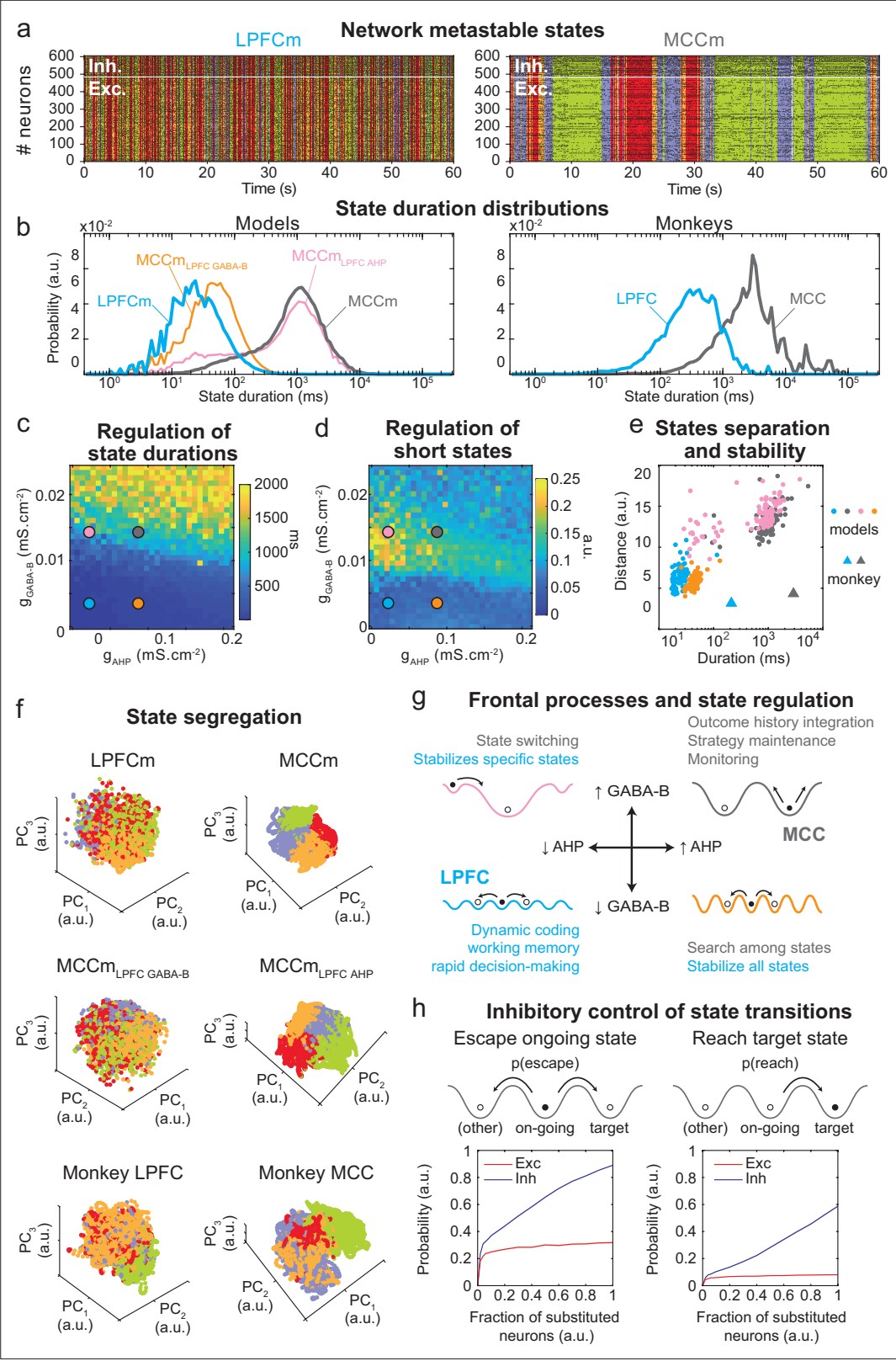

**Figure 7.** Properties of metastable states in the lateral prefrontal cortex (LPFC) and midcingulate cortex (MCC). (**a**) LPFCm and MCCm spiking raster plots (black dots), with Hidden Markov model states (HMM, coloured bands). (**b**) State duration distributions: probability distributions of being in states of given durations in LPFCm (blue), MCCm (grey), MCCm with LPFCm $g_{AHP}$ (MCCm$_{LPFC\ AHP}$ pink), and MCCm with LPFCm $g_{GABA-B}$ (MCCm$_{LPFC}$

*Figure 7 continued on next page*

*Figure 7 continued*

$_{GABA-B}$, orange) models (left) and monkey LPFC (blue) and MCC (grey) areas (right). Each model was simulated 100 times and analysed via HMM, while monkey data was analysed via HMM with 100 different initiation parameter states. Periods above 300 s were excluded. (**c, d**) Regulation of state duration and short states: median state duration (**c**) and Kolmogorov-Smirnov one-sample test statistic or maximal distance of state duration probability distributions to log-normality, as a measure of the over-representation of short states (**d**), as a function of $g_{AHP}$ and $g_{GABA-B}$ maximal conductances. Coloured disks indicate parameter values of models LPFCm, MCCm, MCCm$_{LPFC}$ $_{AHP}$, and MCCm$_{LPFC\ GABA-B}$, respectively. Each point is the average of five simulations. (**e**) Separation between states: average distances between HMM states (averaged pairwise distance between neural centred standardized frequency centroids [temporal averages] of HMM states), as a function of median state durations. Distances calculated over 100 simulations in models and once for monkey LPFC and MCC data. (**f**) State segregation: projection of neural activity on the principal components of the principal component analysis (PCA) space of example model simulations and of monkey data. State colours as in (**a**). (**g**) Frontal processes and state regulation: schematic attractor landscapes in the LPFC and MCC. Horizontal and vertical arrows indicate possible regulations of AHP and GABA$_B$ conductance levels respectively by intrinsic/synaptic plastic processes or neuromodulation in the LPFC and MCC. Likely functional processes operating in these landscapes are indicated in blue for the LPFC and grey for the MCC. (**h**) Inhibitory control of state transitions: probability to escape an ongoing state (left) and to reach a target state (right), when the ongoing state is perturbed by substituting a given proportion of its excitatory (vs. inhibitory) neurons' activity by that of the same neurons in the (perturbing) target state (see Materials and methods). Average (full line), ± s.e.m. (shaded areas, almost imperceptible).

The online version of this article includes the following source data and figure supplement(s) for figure 7:

**Source data 1.** Spiking statistics comparison between monkey and model data.

**Source data 2.** Analysing the causal relationship between neural frequency drift and Hidden Markov model (HMM) state durations in monkey spike data.

**Figure supplement 1.** Hidden Markov model (HMM) of model pseudo-population datasets and analysis of the role of task variable coding in HMM analyses.

**Figure supplement 2.** Analysing the correlational relationship between neural frequency drift and Hidden Markov model (HMM) state durations in monkey spike data.

**Figure supplement 3.** Hidden Markov model (HMM) state durations are distributed exponentially as implied by metastability.

**Figure supplement 4.** Akaike information criterion (AIC) and Bayesian information criterion (BIC) analysis of the number of Hidden Markov model (HMM) states, and its influence on HMM state durations.

## Discussion

We showed LPFC and MCC displayed long population spiking timescales (TAU), with larger values in MCC, consistent with previous observations (*Chaudhuri et al., 2015*; *Murray et al., 2014*). In fact, LPFC and MCC express distinctive and complex temporal organizations of their activity, which cannot be solely captured by the population spiking timescale. The spiking timescale has been used as a measure characterizing intrinsic areal properties and an inter-area temporal hierarchy. However, the spiking timescale of single units varied over two orders of magnitude within each area (*Cavanagh et al., 2018*; *Murray et al., 2014*; *Wasmuht et al., 2018*). The latency of autocorrelogram also demonstrate informative variability, which suggest important underlying functional richness. Our study demonstrates that the temporal signature (TAU and LAT) of single units, measured through spike autocorrelogram metrics and cell-type segregation, can highlight specific local ionic and synaptic mechanisms. Differences in temporal signatures, for instance between LAT (the time lag of the peak of the autocorrelogram) of FS and RS in MCC, and within regions, provide important information on the functional properties of the underlying neural network.

Unravelling the multidimensional nature of LPFC and MCC temporal signatures at the level of individual neurons enabled us to constrain refined biophysical recurrent network models and reveal the local biophysical determinants mechanistically accounting for their specific temporal organization. Moreover, we showed that these determinants control neurodynamical features that constitute core computational foundations for the executive cognitive processes operated by these frontal areas.

## Functional spatio-temporal organization of temporal signatures in frontal areas

The relationship between temporal signatures and behaviour suggests how such biophysical properties could contribute to functional specificities. Such functional relations are still debated. Spiking timescales distributions have been related to persistent activity, choice value, and reward history in the LPFC and MCC (*Bernacchia et al., 2011*; *Cavanagh et al., 2018*; *Meder et al., 2017*; *Wasmuht et al., 2018*), but in a recent study no correlation was observed between task-dependent and intrinsic timescales at the unit level (*Spitmaan et al., 2020*). In all those studies, however, cell types were not considered. Here, we could not estimate task-relevant timescales to correlate with TAU, but we found that the spiking timescales of MCC RS units increased on average during periods of engagement in cognitive performance, likely reflecting the global implication of neural processes in task performance at long timescales. MCC units with different temporal signatures differentially contributed to cognitive processes known to engage MCC, namely feedback/outcome processing and outcome history representations (*Kennerley et al., 2009*; *Quilodran et al., 2008*; *Seo and Lee, 2007*). Outcome processing generally enables rapid – trial by trial – adaptation of control and decisions, while outcome history representations contribute to the long-term – across trials – establishment of values guiding strategy adaptation (*Behrens et al., 2007*; *Karlsson et al., 2012*). Here, population analyses suggested that short spiking timescale units contributed to feedback processing, whereas long spiking timescale units and especially RS units contributed to encode gauge size, which linearly increase with the accumulation of rewards across trials. In MCC, this temporal dissociation coincided with a spatial organization along the antero-posterior axis: anterior units mainly encoded feedback valence, more strongly and earlier than posterior units, whilst posterior units mostly encoded the long-term information related to gauge size. This antero-posterior gradient strikingly resembles that observed in humans (*Meder et al., 2017*).

## Local molecular basis of frontal temporal signatures

Through extensive parameter exploration of constrained biophysical frontal network models, we identified two conductances that precisely reproduced all monkey temporal signatures. In the model, higher TAU (i.e. MCC vs. LPFC, posterior vs. anterior MCC) was accounted for by stronger synaptic GABA-B levels, consistent with reported higher GABA-B receptor densities (*Zilles and Palomero-Gallagher, 2017*), stronger and slower inhibitory currents in the MCC (vs. LPFC) (*Medalla et al., 2017*), and stronger GABA-B receptor densities in the posterior (vs. anterior) MCC (*Palomero-Gallagher et al., 2009*). Excitatory synaptic transmission has been proposed to be a crucial determinant of longer spiking timescales in the temporal cortical hierarchy (*Chaudhuri et al., 2015*). We found that while stronger excitatory transmission increases TAU (possibly accounting for longer MCC TAUs), it also decreases LAT. LAT, however, was longer in the monkey MCC. This inability to reproduce the temporal signature pattern of frontal areas suggests that GABA-B inhibitory – rather than excitatory – transmission is likely the principal causal determinant of longer spiking timescales, at least in the LPFC and MCC. Noticeably, long timescales do not require strong inhibitory-to-inhibitory connections (*Kim and Sejnowski, 2021*) nor specific disinhibition between molecularly identified subnetworks of interneurons (*Wang, 2020*), but of strong slow inhibition to both excitatory and inhibitory neurons. Note also that long timescales naturally emerge from weights variability (see below) and does not require synaptic learning as found elsewhere (*Kim and Sejnowski, 2021*). The model also predicts that higher LAT in the MCC originate from increased refractoriness through higher AHP conductances in RS units (which increases first-order latency). Higher AHP implies lower input-output gains in MCC RS units, compared to the LPFC (*Naudé et al., 2012*), as found empirically (*Medalla et al., 2017*). Finally, reproducing appropriate temporal signatures required the cationic non-specific (CAN) conductance in the areas' RS units. This was observed in RS of rodent medial frontal areas (*Haj-Dahmane and Andrade, 1997*; *Ratté et al., 2018*), where it regulates, together with AHP, cellular bistability and memory, network persistent activity, and computational flexibility (*Compte et al., 2003*; *Papoutsi et al., 2013*; *Rodriguez et al., 2018*; *Thuault et al., 2013*). Our conclusions do not preclude the contribution of other factors to temporal signatures such as different positions in the anatomical hierarchy, different proportion of excitatory to inhibitory neurons, large-scale hierarchical gradients of other neurotransmitter receptor or receptor subunit expression (*Chaudhuri et al., 2015*), distinct neuromodulations

(see below), different extra-regional inputs, or inputs with different spectral contents to LPFC and MCC.

## Frontal temporal signatures uncover metastable dynamics

The LPFC and MCC activity, both in models and in monkeys, was metastable, that is, organized in sequences of discrete, quasi-stationary states in which activity fluctuates around fixed-point attractors (*Abeles et al., 1995*; *La Camera et al., 2019*; *Rich and Wallis, 2016*; *Seidemann et al., 1996*). Such states were robustly found, whether analysing populations or pseudo-populations of neurons (see Materials and methods). As a general rule, the duration of states increases with the stability of their attractor (i.e. the depth/width of their basin of attraction) and decreases with spiking fluctuations. Fluctuations originate from stochastic inputs or chaotic noise (as in our model), and they trigger state transitions. Here, activity was always present as consecutive states occurred, that is, with no interruption, and therefore departed from UP/DOWN dynamics in which the network was either active or silent (*Jercog et al., 2017*).

States were longer in monkeys, likely because extensive training induced attractors that were more stable, whereas models displayed less stable attractors that simply resulted from just random connectivity without learning. Thus metastability genuinely emerged from synaptic heterogeneity and did not require strong network clustering (*La Camera et al., 2019*). We showed that high GABA-B levels are crucial to stabilize states because they amplify the heterogeneity of inhibition and widens attractors, as reflected by higher state separation in the MCCm. In addition, GABA-B's long time constant naturally promotes burstiness, that is, stable discharge episodes. Finally, higher AHP levels, required for higher LAT in MCC RS units, limited the occurrence of the shortest states, limiting frequent transitions between states. AHP conductances have been implied in other computational functions such as in the linearization of neuronal input-output function (*Wang, 1998*), network decorrelation (*Renart et al., 2010*), or the complexity of network dynamics (*Cartling, 1993*). This diversity may emerge from differences in AHP gating dynamics considered, for example, fast (here) vs. slow (*Jercog et al., 2017*) AHP currents.

In monkeys and biophysical models, temporal signatures, which correlate with state stability, actually reflect the underlying temporal organization of neurodynamics into metastable states. Interestingly, state durations (up to >10 s) were longer than spiking timescales (<0.5 s), reconciling the apparent discrepancy between typical spiking timescales in frontal areas (<1 s) and the functional timescales at which those areas operate (up to tens of seconds, *Bernacchia et al., 2011*).

## Functional significance of controlled metastable states in frontal areas

Metastable states can be linked to specific representations in the brain at a variety of levels of abstraction, from stimuli to mental states (*Engel et al., 2016*; *La Camera et al., 2019*; *Mazzucato et al., 2015*; *Mazzucato et al., 2019*; *Rich and Wallis, 2016*; *Taghia et al., 2018*). In general, state transitions contain appreciable randomness, with high transition rates signing internal deliberation, whilst more stable states predicting forthcoming decisions (*La Camera et al., 2019*). We suggest that controlling itinerancy among metastable states constitutes a core neurodynamical process supporting executive functions in frontal areas, which allows to scan choices and strategies, generate deliberation, and solve ongoing tasks.

Specifically, in the MCC (*Figure 7g*, grey landscape) GABA-B-mediated long metastable states underlying long spiking timescales may contribute to the maintenance of ongoing strategies (*Durstewitz et al., 2010*; *Enel et al., 2016*; *Stoll et al., 2016*) and to the integration of outcome history (*Kennerley et al., 2006*; *Meder et al., 2017*; *Seo and Lee, 2007*; *Tervo et al., 2014*). At shorter timescales, short states might instantiate dynamic coding, flexible computations, and rapid decision-making in the LPFC (*Figure 7g*, blue landscape) (*Rich and Wallis, 2016*; *Rigotti et al., 2013*; *Stokes, 2015*). Short states may be lengthened in the LPFC when AHP is increased (*Figure 7g*, orange landscape), favouring longer timescales and a global stabilization of, for instance, working memory processes (*Cavanagh et al., 2018*; *Durstewitz and Seamans, 2008*). Conversely, decreasing GABA-B destabilizes all long states in the MCC model, globally favouring fast transitions (*Figure 7g*, orange landscape). This mechanism might contribute to abandon prior beliefs and to rapid search for adapted representations, for example, in uncertain environments (*Karlsson et al., 2012*; *Quilodran et al., 2008*; *Stoll et al., 2016*). In the LPFC model with increased GABA-B or in the MCC model with

decreased AHP, activity destabilizes certain long states, favouring transitions to remaining long states (*Figure 7g*, pink landscape). Such a configuration might be relevant for flexible behaviours, directed exploration, and switching (*Durstewitz et al., 2010*; *Pasupathy and Miller, 2005*; *Russo et al., 2021*; *Stoll et al., 2016*). Regulating GABA-B and AHP to dynamically adapt computations and temporal signatures could be achieved through neuromodulatory or fast plastic processes (*Froemke, 2015*; *Satake et al., 2008*).

Macroscopic gradients of inhibitions and excitations appear as important determinants of the large scale organization of cortical dynamics (*Wang, 2020*; *Womelsdorf et al., 2014b*). Our results indicate a complementary fundamental dual role of local inhibition in regulating state durations and stability on one hand, and setting the timing and direction of state transitions, on the other. Moreover, transitions can be easily triggered using very few inhibitory neurons. Our study suggests that interneurons and inhibition might be causal in error-driven state transitions in the MCC. Such transitions, initiated by FS neurons immediately after feedback onset (*Figure 4*), would allow escaping currently unsuccessful states, reaching alternatives or exploring new states.

In conclusion, we showed that local ionic and synaptic determinants specify the scale of temporal organization of activity in frontal cortical areas. These determinants might produce the particularly long states observed in monkey MCC dynamics and could explain its contribution to functions operating over extended behavioural periods. More generally, our results suggest that the diversity of spiking timescales observed across the cortical hierarchy reflects the local excitability- and synaptic inhibition-mediated regulation of metastability, which sets the temporal organization of computational processes.

# Materials and methods

## Subjects and materials

This project was conducted with two male rhesus monkeys (*Macaca mulatta*), monkeys A and H. All procedures followed the European Community Council Directive (2010) (*Ministère de l'Agriculture et de la Forêt, Commission nationale de l'expérimentation animale*) and were approved by the local ethical committee (*Comité d'Ethique Lyonnais pour les Neurosciences Expérimentales*, CELYNE, C2EA #42). Electrophysiological data were recorded using an Alpha-Omega multichannel system (AlphaOmega Engineering, Israel).

## Recording sites

Recording chambers (Gray Matter Research, Bozeman, MT) were centred on antero-posterior coordinates of +34.4 and+33.6 relative to ear bars (for monkeys A and H, respectively) (*Stoll et al., 2016*). MCC recording sites covered an area extending over 10 mm (anterior to posterior), and at depths superior to 4 mm from cortical surface (corresponding to the anatomically defined aMCC or functionally defined dACC). Recording sites in LPFC were located between the principalis and arcuate sulcus and just dorsal to the arcuate (areas 6DR, 8B, 8A, and 9/46) and at depths inferior to 2 mm from cortical surface (see supplemental figures in *Stoll et al., 2016*). Reconstructions of cortical surface, of MRI sections perpendicular to recording grids and of microelectrode tracks were performed using neuronavigation. Locations were confirmed with MRI reconstructions and stereotaxic measurements by keeping track of electrophysiological activity during lowering of electrodes.

## Single unit activity

Electrophysiological activity was recorded using epoxy-coated tungsten electrodes (1–2 MOhm at 1 kHz; FHC Inc, Bowdoin, ME) independently lowered using Microdrive guidance (AlphaOmega Engineering). Neuronal activity was sampled at 22 kHz resolution. Single units were sorted offline using a specific toolbox (UltraMegaSort2000, Matlab toolbox, Kleinfeld Lab [*Hill et al., 2011*], University of California, San Diego, CA). Metrics served to verify the completeness and purity of single unit activity. Each single unit activity was selected, recorded, and included in analyses on the basis of the quality of isolation only. We obtained 298 MCC units and 272 LPFC units while monkeys performed a checking task (*Stoll et al., 2016*). A subset of these data has been used in a previous publication (*Stoll et al., 2016*).

## Spike shape clustering

Spike shapes can be clustered in different groups that might correspond to different putative cell populations. For each single unit, we computed the average spike shape on which we extracted the spike width, represented by the time between the peak and the trough (maximal and minimal value, respectively), and the spike amplitude defined by the ratio between the minimum value of the waveform following the peak and the peak. To cluster units we first computed the spike width vs. spike amplitude distance matrix (*dist* function in R). The partitioning led to three clusters, one with narrow spike shapes, one with wide spikes, and one with very wide spikes. Narrow and wide spikes were considered FS and RS, respectively. Although clustering revealed three clusters, no differences were found between the two wide ones, both considered RS neurons (see *Figure 2—figure supplement 1*). To get a statistical confirmation of the numbers of retained clusters, we then fitted the distribution of spike widths using Gaussian Mixture Models (*Mclust* function from the package *MClust* in R, which uses the expectation-maximization algorithm). This method was previously applied for spike clustering (*Torres-Gomez et al., 2020*). We tested the presence of up to three mixture components with variable/unequal variance, comparing the different models using Bayesian information criterion (BIC). In this context BIC values are an approximation to integrated likelihood and should be maximized (*Banfield and Raftery, 1993*; *Scrucca et al., 2016*).

The model which fitted best the spike widths distribution was composed of three Gaussians (BIC unimodal: 255, BIC bimodal: 477, BIC trimodal: 588).

We decided not to use the firing rate for the clustering because we did not have a clear justification for choosing a specific period for firing rate calculation. Yet we found that the so-called FS population we extracted had a higher firing rate than the RS population (*Figure 2—figure supplement 2a*). This difference is in adequation with the literature and supported our decision to cluster units solely based on spike shape/duration. This difference of firing rate is reported when computed from the whole recording. Firing rate computed from different periods of the recordings (when monkeys are engaged in the task, taking a break or around key task events, etc.) are correlated but variable. Actually, in our recordings the correlation is lowest when considering firing rates between pauses and the fore period of the task, two of the periods which could have been logical candidates for a firing rate of reference.

## Spike count autocorrelogram and timescale

The primary analysis of timescales was based on *Murray et al., 2014*. Spike counts (sc) were measured in 14 successive bins of 50 ms from the pre-cue period (700 ms) of each trial, when the monkey is in a controlled, attentive state awaiting stimulus onset. We first calculated the cross-trial bin cross-correlations. Each vector of spike counts from the 50 ms bin *t* was correlated with vectors of spike counts at subsequent bins (*t*+1, *t*+2, etc.) generating an autocorrelation matrix. Autocorrelograms were computed for negative and positive lags, producing a histogram symmetric along the zero axis. Timescales were computed using the autocorrelation defined over positive time lags. The autocorrelogram data was then fitted using non-linear least square (*nls* function in R) to a function of the form:

$$R \sim A * e^{\frac{-t}{scTAU}} + B$$

where *R* is the correlation coefficient and *t* the bin time. scTAU, representing the decay of the exponential function and thus the timescale, and *A*, a scaling constant, were obtained from the fit. We computed scTAU both at the population level, by using a global fit on all recorded units from a given area (as in *Murray et al., 2014*), and at the single unit level.

However, the above method cannot resolve the fine dynamics of neuronal activity at short time lags because it is based on counts pooled across trials and from coarse-grained time bins (50 ms). Moreover, the large variability of unit discharge resulted in a high variability of autocorrelograms, which could not be fitted in many cases (47.5% failures), as in other studies (52.1% and 48.4% failures in *Wasmuht et al., 2018* and *Cavanagh et al., 2018*, respectively). Finally, tracking the causal determinants of LPFC and MCC temporal signatures in terms of local cellular and/or network dynamics requires a high temporal precision, because they rely on intrinsic and synaptic time constants, which often lie below the coarse time bin of the spike count method. To prevent these shortcomings, we directly computed the autocorrelogram of individual neurons from spike times, allowing for high temporal precision in parameter estimation. For this we leveraged all the data recorded for each neuron to reduce the large noise present at the level of individual neurons.

## Spike autocorrelogram analysis

To capture the dynamics of neuronal activity, we computed autocorrelograms from individual unit spike time series and extracted their latencies (LAT; the time lag of the peak of the autocorrelogram) and time constants (TAU). The same method was applied to units from in vivo recordings and neurons from network models. To do so, we computed the lagged differences between spike times up to the 100th order, that is, the time differences between any spike and its *n* successors (up to *n* = 100) at the unit level. The lagged differences were then sorted in 3.33 ms bins from 0 to 1000 ms. The resulting counts allowed to build the probability density function of the autocorrelogram (AC) that we multiplied by the inverse of the time bin width so that it peaked at 1 and is graphically more understandable (as in *Figure 2*). We then smoothed the AC by local non-linear regression (*loess* method, with span 0.1; to filter high-frequency noise and correctly detect the peaks, see below) after removing its first 10 ms, to eliminate potential source data contaminations, such as ISIs shorter than the absolute refractory period. We defined the peak of the AC as its maximum, except when the maximum was the very first bin, in which case the peak was defined as the first local maximum after the first bin. The latency of that peak, LAT, was considered in further analyses as a structural parameter of the AC characterizing the temporal signature of the neuron/unit spiking set. For each AC, a global mono-exponential fit (GLOBAL fit) was then performed on the part of the AC situated after the peak using the Levenberg-Marquardt algorithm (*nlsLM function in R*) for monkey data or von-Neumann-Karmarkar interior-point algorithm (*fmincon in Matlab*) for network models (we checked that either algorithm on the same spiking sets gave similar results), as follows:

$$AC \sim Ae^{-t/TAU} + B$$

TAU, the time constant of the AC fit characterized the temporal signature of the neuron. *A*, the amplitude of the exponential, and *B*, the offset, are positive constants. Note that this mono-exponential fitting equation is strictly equivalent to that of *Murray et al., 2014*, *B* here corresponding to *AB* in the Murray method. Choosing one or the other equation did not affect the resulting fit and we kept the present form as it is easier to interpret. Fits on each AC were performed 50 times, with random initial guesses in the range $[0, 2(max(AC) - min(AC))]$ for *A*, $[0, 2min(AC)]$ for *B*, and [0, 1000] ms for TAU, from which the best fit was kept.

In a minority of cases (<3% of neurons), the AC following the peak (as defined above and denoted below the 1st peak) could present a shape that diverged from a simple exponential decay, because the first peak was followed by: a fast and large dip, then a second peak (local maximum), then the slower, final exponential decay. In this case, we developed a pipeline aiming at consistently choosing the peak (i.e. 1st or 2nd) from which the fit started. We defined the AC as having a dip if the first local minimum in the 100 ms after the 1st peak was below 75% of the global range of the AC, $max(AC) - min(AC)$ (to avoid modelling local troughs due to noise as dips). When two peaks were detected, the second peak was defined as the maximum of the AC after the dip. Two additional mono-exponential fits of the AC were then performed, one from the first peak to the dip (FAST fit) and a second one from the second peak to the end of the AC (SLOW fit).

To be valid, any individual fit (whether of the GLOBAL, FAST or SLOW type) had to display positive *A*, *B*, and TAU values. When neurons had a valid GLOBAL fit, two possibilities were considered. First, the valid GLOBAL fit was kept when (1) at least one of the FAST and SLOW fits were not valid or when (2) it was the best (i.e. its root-mean-square error was inferior to that of the sum of the valid FAST and SLOW fits). Neurons that did not have a valid GLOBAL fit were excluded from further analysis. Thus, while FAST and SLOW fits were de facto systematically excluded from further analysis, they were only used to ensure the quality of GLOBAL exponential fits. Note again that excluding <3% of neurons, this complex procedure was very conservative and designed for the sake of fitting performance.

All codes are freely available (*Fontanier et al., 2020*).

## Statistical analysis

All analyses were performed using R (version 3.6.1) with the RStudio environment (*R core team, 2014*).

*BLOM transformation*. As some timescale measures are non-normally distributed, analyses required a robust non-parametric test. We opted for the BLOM transformation which is a subcase of rank-based inverse normal transformations (*Beasley et al., 2009*). Basically, the data is ranked and then

back-transformed to approximate the expected normal scores of the normal distribution according to the formula:

$$Y_i = \Phi^{-1} \frac{r_i - c}{N - 2c + 1}$$

where $r_i$ is the ordinary rank and $Y_i$ the BLOM transformed value of the $i$th case among the $N$ observations. $\Phi - 1$ is the standard normal quantile (or probit) function and $c$ a constant set to 3/8 according to **Blom, 1958**. Regular parametric analyses can then be performed on the transformed data. Since $z$-scores of the transformed data are normally distributed and differences are expressed in standard errors, main effects and interactions can easily and robustly be interpreted. As sanity checks we also ran more classical non-parametric tests (Wilcoxon test) on non-normally distributed data leading to the same conclusions.

## Behaviour and context-dependent modulations

*Behavioural task.* Monkeys were trained to perform a dual task involving rule-based and internally driven decisions (**Stoll et al., 2016**). Monkeys performed the task using a touch screen. In each trial they could freely choose whether to perform a rewarded categorization task or to check their progress towards a large bonus juice reward (**Figure 3a**). Upon checking (selection of a disk-shaped lever) progress was indicated by the onset of a visual 'gauge' (an evolving disk inside a fixed circle). Choosing the categorization task (selection of an inverted triangle lever) started a delayed response task in which an oriented white bar (cue) was briefly presented, followed by a delay at the end of which two bars oriented 45° leftward and rightward where presented. Selecting the bar matching the cue orientation led to a juice reward. An incorrect response led to no reward delivery. The gauge increased based on correct performance in the categorization task following seven steps to reach the maximum size. If the animal checked while the gauge was full, the bonus reward was delivered, and the gauge reset to step 1. The full gauge was reached after either 14, 21, 28, or 35 correct trials (=number of trials to complete the seven steps, pseudo-randomly chosen in each block). Thus, the gauge could increase at one of four different speeds.

*Pause vs. engage periods.* As each trial was self-initiated by the animal, monkeys could decide to take a break in their work. We defined pauses as periods of at least 60 s without trial initialization. Monkeys made on average 3.4±2.57 pauses per session (mean ± sd, monkey A: 3.44±2.55, monkey H: 3.34±2.63; see **Figure 3b**). We extracted spike times during the defined pause and engage time segments for each unit. To control for a time-on-task confound on timescale modulation in this analysis, we contrasted pauses with engaged periods that occurred at the same time of the session (after the first pause). Because engage periods were as long as pause periods for one monkey (monkey H, 53 sessions, MDengage = 392 s, MDpause = 396 s, Wilcoxon-paired test: $V$=790, p=0.51) and roughly twice as long for the other (monkey A, 96 sessions, MDengage = 638 s, MDpause = 372 s, Wilcoxon-paired test: $V$=406, p=2.19e-12), we decided not to further segment the data to avoid resampling biases. This analysis was conducted on units for which TAU could be extracted for both periods ($n_{MCC-FS}$=19, $n_{MCC-RS}$=80, $n_{LPFC-FS}$=21, $n_{LPFC-RS}$=97).

*Fast vs. slow-paced blocks.* We defined 14 and 21 correct trials blocks to be fast blocks and 28 and 35 correct trials blocks as slow blocks (**Figure 3a**, bottom). We considered neuronal activity from the first-time monkeys checked in a block until the end of the block. We excluded pause periods from this analysis. We extracted spike timing from the segments and computed timescales as previously, keeping only units with successful timescale extraction for both periods ($n_{MCC-FS}$=33, $n_{MCC-RS}$=165, $n_{LPFC-FS}$=46, $n_{LPFC-RS}$=165).

*Emptier vs. fuller gauge size seen.* In each block, monkeys used the gauge size observed upon checking to regulate their future decisions to check. The checking frequency increased with gauge size with a marked increase at steps >4. We thus compared neuronal activity in periods in which monkeys saw gauges of size <4, with periods in which they saw gauges >4, excluding the very beginning of blocks when monkeys have not seen the gauge yet, and pauses periods. We performed this analysis on 430 units ($n_{MCC-FS}$=30, $n_{MCC-RS}$=178, $n_{LPFC-FS}$=47, $n_{LPFC-RS}$=175).

To test whether current block speed had an influence on TAU at the unit level, we computed a modulation index for each unit: log(TAU$_{slow}$)/log(TAU$_{fast}$). Similarly, to test whether gauge filling state had an influence on TAU at the unit level, we computed a modulation index for each unit: log(TAU$_{empty}$)/

log(TAU$_{full}$), where TAU$_{full}$ corresponds to TAU calculated on the spike data recorded during the time in blocks where the gauge was superior of equal to the 4th level.

## Task-related analyses

*Single unit activity.* Each unit's spikes were counted in sliding bins of 200 ms overlapping by 50 ms from feedback onset to 800 ms post-feedback and during the intertrial interval from 400 ms before the end of trial signal onset to 2000 ms after its onset.

*Group analyses using a glmm.* We used a *glmm* using a Poisson family. p-Values were corrected for multi-comparison with the false discovery rate algorithm with the number of comparisons being the number of timebins (p.adjust function in R).

The mixed models used were of the form:

$$y = \beta_0 + \beta_1.\text{CheckWork} + \beta_2.\text{Gauge} + \beta_3.\text{Previousfeedback} + \gamma.Z + \epsilon$$

where $\gamma{\cdot}Z$ is the random term, and CheckWork, Gauge, and PreviousFeedback are the fixed effects describing the Check vs. Work decision (0/1), the gauge size (1–7) and the feedback in the previous trial (0/1) with their respective parameters ($\beta$). In the *glmm*, the single unit identity was used as a random factor.

A persistent problem with Poisson models in biology is that they often exhibit overdispersion. Not accounting for overdispersion can lead to biased parameter estimates. To deal with overdispersion we used observation-level random effects, which model the extra variation in the response variable using a random effect with a unique level for every data point.

*Median splits.* To test the hypothesis that units with different timescales may encode feedback differently, we divided the units into two groups based on the median of the timescale metric. We computed the median of the metric (e.g. peak latency or TAU) in all the units of a given cell type. Then we put units with a metric value below the median into the 'short' group and units with a metric value above the median into the 'long' group. These splits led to the following population of units: LPFC: FS short: 37, FS long: 18, RS short: 148, FS long: 54 – MCC: FS short: 10, FS long: 29, RS short: 67, RS long: 161.

To assess differences between short and long TAU population coding for a given area and cell type, we have constructed null distributions of coding (*z*-values) by permuting TAU group allocation of units. Such permutations allowed us to retain differences in sampling (e.g. the population of MCC RS with long TAU is larger than the short TAU one). This procedure was performed 100 times for each area and cell type. We then compared the position of the true data relative to the cumulative distribution of the permutations and set a statistical threshold at $\alpha$=0.05. Outcomes are shown as raster above x-axes in panel in *Figure 4*.

*Single unit approach.* To investigate whether each single unit activity encoded the different key variables of the task, we analysed variations of spike counts measured in each trial using a glm (using the libraries MASS and ggplot2 for graphics under R software) (see also *Stoll et al., 2016*). Spike counts were measured on successive bins of 200 ms moved smoothly by 50 ms around key event times in each trial. Because of the statistical properties of count data, the glm were applied using a Poisson regression (Poisson error structure). We checked for overdispersion by dividing residual deviance by the degree of freedom. In case of overdispersion, we applied a negative binomial regression using the glm.nb() function in R. To validate this choice for each set of data, we statistically compared the two models (Poisson and negative binomial) fitted for each set (likelihood ratio test, w2-test).

Proportions of significant single units were extracted from the sliding glm if they significantly discriminated the factor of interest for four consecutive bins (covering a time period of 350 ms).

We used also used the wPEV (computed with the function *anova_stats* from the sjstats package in R) as a statistical measure to quantify the extent to which the variability in neural firing rate was determined by feedback valence and gauge state. We then quantified the time-resolved proportion of cells coding for the task variables (p<0.05 for at least four successive bins) and the wPEV in populations of units with short or long TAU (median split by cell type). To assess differences between short and long TAU populations, we built null distributions by permuting 1000 times the TAU group allocation of units. We then compared the position of the true data relative to the cumulative distribution of the permutations and set a statistical threshold at $\alpha$=0.05.

## Timescale and coding variations along the antero-posterior axis

We considered the genu of the arcuate sulcus as an anatomical landmark from which we computed distances of recording location along the anterior-posterior axis from MRI reconstructions.

We questioned TAU antero-posterior variability keeping recording locations covering the same range in both monkeys. We ordered locations from the most posterior site for each area. We excluded FS units from statistical analysis due to their disparateness (RS units, monkey A: $n_{MCC}$ = 112, $n_{LPFC}$ = 110; monkey H: $n_{MCC}$ = 54, $n_{LPFC}$ = 64). This analysis was conducted separately between monkeys to account for inter-subject anatomical variability.

To test variation in population coding along the antero-posterior axis, we divided single units into a posterior and anterior group based on the range of locations of each area (posterior MCC from 4.5 to 7 mm, $n_{MCCRSpost}$ = 84, $n_{MCCFSpost}$ = 14; anterior MCC from 7 to 9.5 mm, $n_{MCCRSant}$ = 82, $n_{MCCFSpost}$ = 16; posterior LPFC from 2.5 to 6 mm, $n_{LPFCRSpost}$ = 77, $n_{LPFCFSpost}$ = 19; anterior LPFC from 6 to 8.5 mm, $n_{LPFCRSant}$ = 97, $n_{LPFCFSant}$ = 19). Population coding analysis is described in task-related analyses.

## Cellular model of pyramidal neurons in frontal areas

We built a generic biophysical Hodgkin-Huxley model of the detailed dynamics of membrane potential and of ionic and synaptic currents of individual pyramidal neurons in frontal areas. The model was generic, being endowed with a large set of ionic voltage- and calcium-dependent conductances, to encompass the wide possible repertoire of spiking discharge patterning encountered in vivo. In the model, the membrane potential followed

$$C\frac{dV}{dt} = -\left(I_{Ionic} + I_{Syn}\right)$$

where $C$ is the specific membrane capacity and the membrane ionic current writes

$$I_{Ionic} = I_L + I_{Na} + I_K + I_{CaL} + I_{CAN} + I_{AHP} + I_{CaT} + I_H$$

in which the leak current is

$$I_L = \bar{g}_L \left(V - V_L\right)$$

and action potential (AP) currents ($I_{Na}$, $I_K$) are taken from a previous model we devised to reproduce spike currents of frontal pyramidal regular-spiking neurons (**Naudé et al., 2012**). The high-threshold calcium current was

$$I_{CaL} = \bar{g}_{CaL}\, p_{CaL}^2 \left(V - V_{CaL}\right)$$

where the activation followed first-order kinetics

$$\frac{dp_{CaL}}{dt} = \left(p_{CaL}^\infty\left(V\right) - p_{CaL}\right)/\tau_{CaL}\left(V\right)$$

with a voltage-dependent time constant

$$\alpha_{CaL} + \beta_{CaL}V$$

$$\tau_{CaL}\left(V\right) = 10$$

where $\alpha_{CaL}$ and $\beta_{CaL}$ were fitted from in vitro data (**Helton et al., 2005**). The infinite activation followed

$$p_{CaL}^\infty\left(V\right) = 1/\left(1 + exp\left(-\left(V - V_{1/2,CaL}\right)/k_{CaL}\right)\right)$$

where $V_{1/2,CaL}$ and $k_{CaL}$ respectively denote the half-activation potential and e-fold slope of the Boltzmann activation voltage dependence, estimated from in vitro data (**Helton et al., 2005**). The cationic non-selective ($I_{CAN}$) current and the medium after-hyperpolarization ($I_{AHP}$) current, responsible for frequency adaptation in pyramidal neurons were taken as in **Rodriguez et al., 2018**, with

$$I_{CAN} = \bar{g}_{CAN}\, p_{CAN} \left(V - V_{CAN}\right)$$

and

$$I_{AHP} = \bar{g}_{AHP}\, p_{AHP}^2 \left(V - V_{AHP}\right)$$

The activation of both currents, $p_x$ ($x \in \{CAN,\ AHP\}$) followed,

$$\frac{dp_x}{dt} = \left(p_x^\infty\left(Ca\right) - p_x\right)/\tau_x\left(Ca\right)$$

with

$$\tau_x\left(Ca\right) = 1/\left(\alpha_x Ca + \beta_x\right)$$

and

$$p_x^\infty\left(Ca\right) = \alpha_x/\left(\alpha_x Ca + \beta_x\right)$$

where $\alpha_x$ and $\beta_x$ respectively denote activation and deactivation kinetic constants consistent with experimental data in layer 5 PFC pyramidal neurons (**Faber and Sah, 2007**; **Haj-Dahmane and Andrade, 1997**; **Villalobos et al., 2004**). The low-threshold calcium ($I_{CaT}$) and hyperpolarization-activated ($I_H$) currents were from reference **Ritter-Makinson et al., 2019**. To account for autocorrelogram parameters, we employed different versions of the model that contained distinct subsets of ionic currents, which have been implicated in adaptation and bursting ($I_{CaL}$, $I_{AHP}$), rebound ($I_{CaT}$, $I_H$), and regenerative and bistable discharge ($I_{CaL}$, $I_{CAN}$, $I_{AHP}$) in cortical pyramidal neurons. Calcium concentration dynamics resulted from the inward influx due to $I_{CaL}$ and $I_{CaT}$ and first-order buffering or extrusion (**Rodriguez et al., 2018**) through:

$$\frac{dCa}{dt} = -\left(1/2F\right)\left(S/V\right)\left(I_{CaL} + I_{CaT}\right) + \left(Ca_0 - Ca\right)/\tau_{Ca}$$

where $F$ is the Faraday constant, $Ca_0$ is the basal intracellular calcium concentration, $\tau_{Ca}$ is the buffering time constant, and

$$S/V = r_1^{-1}\left(1 - r_1/r_0 + r_1^2/\left(3r_0^2\right)\right)^{-1}$$

is the surface area to volume ratio of an idealized intracellular shell compartment of thickness $r_1$ situated beneath the surface of a spherical neuron soma of radius $r_0$.

The synaptic current ($I_{Syn}$) mimicked in vivo conditions encountered by neurons in the asynchronous irregular regime, summing random synaptic excitatory inputs, through AMPA and NMDA receptors, and inhibitory inputs, through GABA$_A$ and GABA$_B$ receptors. Thus,

$$I_{Syn} = I_{AMPA} + I_{NMDA} + I_{GABA_A} + I_{GABA_B}$$

For AMPA, GABA$_A$, and GABA$_B$,

$$I_x = \bar{g}_x\, p_x\left(V - V_x\right)$$

where $p_x$ is the opening probability of channel receptors and $V_x$ the reversal potential of the current. The NMDA current followed

$$I_{NMDA} = \bar{g}_{NMDA}\, p_{NMDA}\, x_{NMDA}\left(V\right)\left(V - V_{NMDA}\right)$$

incorporating the magnesium block voltage dependence modelled (**Jahr and Stevens, 1990**) as

$$x_{NMDA}\left(V\right) = \left(1 + \left[Mg^{2+}\right]e^{-0.062\,V}/3.57\right)^{-1}$$

To simulate fluctuations encountered in vivo, all opening probabilities followed Ornstein-Uhlenbeck processes (**Destexhe and Paré, 1999**)

$$\frac{dp_x}{dt} = \frac{\left(m_x - p_x\right)}{\tau_x^{decay}} + \sigma_x\varepsilon\left(t\right)$$

where $\varepsilon\left(t\right)$ is a Gaussian stochastic process with zero mean and unit standard deviation and $m_x$ and $\sigma_x$ are the mean and standard deviation of the opening probabilities. For AMPA and GABA$_A$, the

mean was taken as the steady-state value of first-order synaptic dynamics described in the network model (see below):

$$m_x = \left(1 + \tau_x^{decay-1} \Delta p_x^{-1} f_{Syn}^{-1} n_{Syn}^{-1}\right)^{-1}$$

with $n_{Syn}$ pre-synaptic neurons firing at a frequency $f_{Syn}$ (with $Syn \in \{Exc, \ Inh\}$ , depending on the type of current considered), an instantaneous increase $\Delta p_x$ of opening probability upon each pre-synaptic spike and first-order decay dynamics with time constant $\tau_x^{decay}$ between spikes. For NMDA and GABA_B, the mean was taken as the steady-state value of second-order synaptic dynamics described in the network model (see below):

$$m_x = \left(1 + \tau_x^{decay-1} \alpha^{-1} \left(1 + \tau_x^{rise-1} \Delta p_x^{-1} f_{Syn}^{-1} n_{Syn}^{-1}\right)\right)^{-1}$$

For all currents, standard deviations were taken as $\sigma_x = 0.5 m_x$ . Feed-forward excitatory and inhibitory currents were balanced (*Xue et al., 2014*), according to the driving forces and the excitation/inhibition ratio, through

$$\begin{cases} \bar{g}_{GABA_A} = g_{GABA_A} \ \frac{-(V_{mean} - V_{Exc})}{(V_{mean} - V_{GABA_A})} \ \frac{n_{Exc}}{n_{Inh}} \\ \bar{g}_{GABA_B} = g_{GABA_B} \ \frac{-(V_{mean} - V_{Exc})}{(V_{mean} - V_{GABA_B})} \ \frac{n_{Exc}}{n_{Inh}} \end{cases}$$

## Model similarity to monkey data

The bivariate probability density distribution of neuronal TAU and LAT autocorrelogram parameters was estimated in RS and FS units in monkey in both the LPFC and MCC, using bivariate normal kernel density functions. For cellular models, similarity maps to monkey data was determined as following: for each model parameter couple of the map, the similarity to the considered cortex (LPFC or MCC) was defined as the probability density of that cortex to display the TAU and LAT parameters produced by the model. Cellular models with mean firing frequency superior to 20 Hz were considered to discharge in an unrealistic fashion, compared to data, and were discarded. In network models, for each parameter value (one-dimensional explorations) or model parameter couples of the map (two-dimensional explorations), the similarity (*S*) was defined as the normalized Frobenius inner product between the bivariate probability density distributions of units in monkeys (*U*) and that of neurons in the network model (*N*), following

$$S_{U,N} = \frac{<U,N>_F}{\|U\|_F \|N\|_F}$$

In order to account for the TAU and LAT autocorrelogram parameters for both RS and FS populations, the similarity was calculated separately as RS with Exc and FS with Inh. Seeing as excitatory neurons represent $p_{Exc} = 0.8$ of the neurons in cortex (*Beaulieu et al., 1992*), the overall similarity was then calculated as

$$S = p_{Exc} \ S_{RS,Exc} + p_{Inh} \ S_{FS,Inh}$$

## Parametric explorations in the pyramidal neuron model

Biophysical properties of neurons can affect autocorrelation parameters in several ways. In principle, increasing the refractory period (through increased hyperpolarizing ionic conductance) shifts the distribution of first-order lags (ISIs), thus increasing LAT (*Figure 5—figure supplement 1a* i). Increasing burstiness of the spike discharge (through increased depolarizing conductance-mediated positive feedback) also increases the latency, because higher-order lag distributions are more peaked. Moreover, conductances with slow time constants (including many bursting-mediating conductances) increase that of the autocorrelogram itself. Finally, all these factors may interact in complex ways in vivo to set the spiking pattern that shapes autocorrelations.

We first explored these alternatives with a detailed biophysical Hodgkin-Huxley model of a generic frontal pyramidal cortical neuron, simulated in in vivo conditions. Pyramidal neurons display a huge electrophysiological diversity set by ionic channels, which, together with synaptic inputs, influences spiking patterns. Two conductances, that is, CAN and AHP, were the sole couple able to affect both

the LAT and TAU of the autocorrelation (compare *Figure 5—figure supplement 1a* ii and iii). Interestingly, these conductances are prominent in monkey LPFC and MCC, as well as rodent prefrontal pyramidal neurons where they control regenerative discharge, bistability, and burstiness (*Haj-Dahmane and Andrade, 1997*; *Medalla et al., 2017*; *Ratté et al., 2018*; *Rodriguez et al., 2018*; *Yang et al., 1996*). Within physiological ranges, (1) the autocorrelogram LAT essentially increased with the maximal $g_{AHP}$ conductance, while (2) TAU increased in an intermediate range of $g_{AHP}$ and increased with $g_{CAN}$ (*Figure 5—figure supplement 1b*), possibly accounting for differences between LPFC and MCC in monkeys. Remarkably, the low-threshold calcium (CaT), high-threshold calcium (CaL), and hyperpolarization-activated H conductances, which are ubiquitous and govern spiking patterns through spiking adaptation and rebound, as well as NMDA and GABA-B synaptic input conductances, which display long time constants, were all ineffective in adequately modulating autocorrelation parameters (*Figure 5—figure supplement 2*).

Computing an estimation of the bivariate probability density distribution of neuronal autocorrelogram parameters for LPFC and MCC RS units (*Figure 5—figure supplement 1c*) allowed to build a map of the similarity of the cellular model to RS units temporal signatures in monkey LPFC and MCC, defined as the bivariate probability density observed for the LAT and TAU yielded by the cellular model, given a ($g_{CAN}$, $g_{AHP}$) couple of parameters (see *Model similarity to monkey data*). We found that the model displayed large (i.e. sub-maximal) similarity to the LPFC in a substantial region of ($g_{CAN}$, $g_{AHP}$) parameters (*Figure 5—figure supplement 1d*). By contrast, this was not true for the MCC (*Figure 5—figure supplement 1d*), because the model was unable to generate LAT in the 100–150 ms range that characterizes the MCC (*Figure 5—figure supplement 1b*), even when exploring large ranges of CAN and AHP conductance kinetic parameters.

## Model of local recurrent neural networks in frontal areas

We built a biophysical model of a generic local frontal recurrent neural network, endowed with detailed biological properties of its neurons and connections. The network model contained $N$ neurons that were either excitatory ($E$) or inhibitory ($I$) (neurons projecting only glutamate or GABA), respectively (*Dale, 1935*), with probabilities $p_E$ and $p_I = 1 - p_E$, respectively, and $p_E/p_I = 4$ (*Beaulieu et al., 1992*). Connectivity was sparse (i.e. only a fraction of all possible connections exists *Thomson et al., 2002*) with no autapses (self-connections) and EE connections (from E to E neurons) drawn to insure the over-representation of bidirectional connections in cortical networks (four times more than randomly drawn according to a Bernoulli scheme; *Song et al., 2005*). The synaptic weights $w_{(i,j)}$ of existent connections were drawn identically and independently from a log-normal distribution of parameters $\mu_w$ and $\sigma_w$ (*Song et al., 2005*). To cope with simulation times required for the massive explorations ran in the parameter space, neurons were modelled as leaky integrate-and-fire neurons, that is, the AP mechanism was simplified, compared to the cellular model (see above). Moreover, leveraging simulations at the cellular level, we only considered the $I_{CAN}$ and $I_{AHP}$ amongst the ionic currents of the cellular model (see above). Thus, the membrane potential followed

$$\begin{cases} \frac{dV_{(j)}}{dt} = -\left( I_{Ionic(j)} + I_{Syn.Rec(j)} + I_{Syn.FF(j)} \right) \\ V_{(j)} > \theta \rightarrow V_{(j)} = V_{rest} \end{cases}$$

where repolarization occurred after a refractory period $\Delta t_{AP}$. The ionic current followed

$$I_{Ionic(j)} = I_{L(j)} + I_{CAN(j)} + I_{AHP(j)}$$

with parameters and gating dynamics of ionic currents identical to the cellular model. The intra-somatic calcium concentration Ca evolved according to discrete spike-induced increments and first-order exponential decay:

$$\frac{dCa_{(j)}}{dt} = \frac{Ca_0 - Ca_{(j)}}{\tau_{Ca}} + \Delta Ca\, \delta\left( t - t_{(j)}^k \right)$$

where $t_{(j)}^k$ is the time of the $k$th spike in the spike train of neuron $j$, $\delta$ the Dirac delta function, $\tau_{Ca}$ the time constant of calcium extrusion, $Ca_0$ the basal calcium, and $\Delta Ca$ a spike-induced increment of

calcium concentration. The recurrent synaptic current on post-synaptic neuron $j$, from – either excitatory or inhibitory – pre-synaptic neurons (indexed by ), was

$$I_{Syn.Rec(j)} = \sum_i \left( I_{AMPA(i,j)} + I_{NMDA(i,j)} + I_{GABA_A(i,j)} + I_{GABA_B(i,j)} \right)$$

The delay for synaptic conduction and transmission, $\Delta t_{syn}$ , was considered uniform across the network (**Brunel and Wang, 2001**). Synaptic recurrent currents followed

$$I_{x(i,j)} = \bar{g}_x \, w_{(i,j)} \, p_{x(i)} \left( V_{(j)} - V_x \right)$$

where $w_{(i,j)}$ is the synaptic weight, $p_{x(i)}$ the opening probability of channel receptors, and $V_x$ the reversal potential of the current. The NMDA current followed

$$I_{NMDA(i,j)} = \bar{g}_{NMDA} \, w_{(i,j)} \, p_{NMDA(i)} \, x_{NMDA}\left(V_{(j)}\right) \left( V_{(j)} - V_{NMDA} \right)$$

with $x_{NMDA}(V)$ the magnesium block voltage dependence (see cellular model). AMPA and GABA$_A$ rise times were approximated as instantaneous (**Brunel and Wang, 2001**) and bounded, with first-order decay

$$\frac{dp_{x(i)}}{dt} = -\frac{p_{x(i)}}{\tau_x^{decay}} + \Delta p_x \left( 1 - p_{x(i)} \right) \, \delta\left( t - t_{(i)}^k \right)$$

To take into account the longer NMDA (**Wang et al., 2008**) and GABA-B (**Destexhe et al., 1998**) rise times, opening probabilities followed second-order dynamics (**Brunel and Wang, 2001**)

$$\begin{cases} \frac{dq_{x(i)}}{dt} = -\frac{q_{x(i)}}{\tau_x^{rise}} + \Delta q_x \left( 1 - q_{x(i)} \right) \delta\left( t - t_{(i)}^k \right) \\ \frac{dp_{x(i)}}{dt} = -\frac{p_{x(i)}}{\tau_x^{decay}} + \alpha_x \, q_{x(i)} \, \left( 1 - p_{x(i)} \right) \end{cases}$$

Recurrent excitatory and inhibitory currents were balanced in each post-synaptic neuron (**Xue et al., 2014**), according to driving forces and excitation/inhibition weight ratio, through

$$\begin{cases} \bar{g}_{GABA_A} = g_{GABA_A} \frac{-(V_{mean} - V_{Exc})}{(V_{mean} - V_{GABA_A})} \frac{\sum_{i \in Exc} w_{(i,j)}}{\sum_{i \in Inh} w_{(i,j)}} \\ \bar{g}_{GABA_B} = g_{GABA_B} \frac{-(V_{mean} - V_{Exc})}{(V_{mean} - V_{GABA_B})} \frac{\sum_{i \in Exc} w_{(i,j)}}{\sum_{i \in Inh} w_{(i,j)}} \end{cases}$$

with $V_{mean} = (\theta + V_{rest})/2$ approximating the average membrane potential.

The feed-forward synaptic current $I_{Syn.FF(j)}$ (putatively arising from cortical and sub-cortical inputs) consisted of an AMPA component

$$I_{Syn.FF(j)} = \bar{g}_{AMPA_{FF}} \, p_{AMPA_{FF}} \left( V_{(j)} - V_{AMPA} \right)$$

with a constant opening probability $p_{AMPA_{FF}}$ .

## Numerical integration and parameters of the models

Models were simulated and explored using custom developed code under MATLAB and were numerically integrated using the forward Euler method with time-steps $\Delta t = 0.1$ ms in cellular models and $\Delta t = 0.5\ ms$ in network models. Computational models are freely accessible on the Zenodo repository (https://doi.org/10.5281/zenodo.5707884) (**Fontanier et al., 2020**).

Unless indicated in figure legends, standard cellular parameter values were as follows. Concerning ionic currents, $C = 1\ \mu F.cm^{-2}$ , $\bar{g}_L = 0.05\ mS.cm^{-2}$ , $V_L = -70\ mV$, $\bar{g}_{Na} = 30\ mS.cm^{-2}$ , $V_{Na} = 50\ mV$, $\bar{g}_K = 2\ mS.cm^{-2}$, $V_K = -90\ mV$, $\bar{g}_{CaL} = 0.01\ mS.cm^{-2}$, $V_{CaL} = 150\ mV$, $\bar{g}_{CAN} = 0.05\ mS.cm^{-2}$ , $V_{CAN} = 30\ mV$ , $\alpha_{CAN} = 0.0015\ \mu M^{-1}.ms^{-1}$ , $\beta_{CAN} = 0.005\ ms^{-1}$ , $\bar{g}_{AHP} = 0.1\ mS.cm^{-2}$ , $V_{AHP} = -90\ mV$ , $\alpha_{AHP} = 0.025\ \mu M^{-1}.ms^{-1}$ , $\beta_{AHP} = 0.025\ ms^{-1}$ , $\bar{g}_{CaT} = 0\ mS.cm^{-2}$ , $V_{CaT} = 120\ mV$ , $\bar{g}_H = 0\ mS.cm^{-2}$ , $V_H = -40\ mV$ , $V_{\tau H1/2} = -105\ mV$ , $k_{\tau H} = 10\ mV$ , $\tau_{H,min} = 1000\ ms$ , $\tau_{H,max} = 6000\ ms$ , $Ca_0 = 0.1\ \mu M$ , $\tau_{Ca} = 25\ ms$ , $F = 96500\ mol.s^{-1}.A^{-1}$ , $r_0 = 4 \cdot 10^{-4}\ cm$ , $r_1 = 0.25 \cdot 10^{-4}\ cm$. Concerning synaptic currents, $\bar{g}_{AMPA} = 0.02\ mS.cm^{-2}$ , $\tau_{AMPA}^{decay} = 2.5\ ms$ , $\bar{g}_{NMDA} = 0.03\ mS.cm^{-2}$ , $\alpha_{NMDA} = 0.275\ ms^{-1}$ ,

$\tau_{NMDA}^{rise} = 4.65\ ms$ , $\tau_{NMDA}^{decay} = 75\ ms$ , $V_{AMPA} = V_{NMDA} = 0\ mV$ , $g_{GABA_A} = 0.0063\ mS.cm^{-2}$ , $\tau_{GABA_A}^{decay} = 10\ ms$ , $V_{GABA_A} = -70\ mV$ , $g_{GABA_B} = 3.125 \cdot 10^{-4}\ mS.cm^{-2}$ , $\alpha_{GABA_B} = 0.015\ ms^{-1}$ , $\tau_{GABA_B}^{rise} = 90\ ms$ , $\tau_{GABA_B}^{decay} = 160\ ms$ , $V_{GABA_B} = -90\ mV$ , $\Delta x_{AMPA} = \Delta x_{NMDA} = \Delta x_{GABA_A} = \Delta x_{GABA_B} = 0.1$ , $V_{mean} = -57.5\ mV$ , $n_{Exc} = 484$ , $n_{Inh} = n_{Exc}/4 = 121$ , $f_{Exc} = 7Hz$ , $f_{Inh} = 7Hz$ , $\left[Mg^{2+}\right] = 1.5\ mM$ .

Unless indicated in figure legends, standard parameter values in network models were identical to cellular model parameters, except for the following. Concerning the network, $N = n_{Exc} + n_{Inh} = 605$ neurons, $p_{Exc} = 0.8$, so that $n_{Exc} = Np_{Exc} = 484$ and $n_{Inh} = Np_{Inh} = 121$. Concerning the weight matrix, $\mu_w = 0.03$, $\sigma_w = 0.02$, $p_{EE} = p_{EI} = p_{II} = 0.3$, $p_{IE} = 0.55$. Concerning integrate-and-fire neuron properties and intrinsic currents, $V_{rest} = -65\ mV$ , $\theta = -50\ mV$ , $V_{mean} = \left(V_{rest} + \theta\right)/2 = -57.5\ mV$ , $\Delta t_{AP} = 3\ ms$ , $\Delta Ca = 0.2\mu M$ , $\bar{g}_{CAN} = 0.025\ mS.cm^{-2}$ . Concerning synaptic currents, $\Delta t_{syn} = 0.5\ ms$ , $\tau_{AMPA_{FF}} = 2.5\ ms$ , $\Delta x_{AMPA_{FF}} = 0.1$, $\bar{g}_{AMPA} = \bar{g}_{AMPA_{FF}} = 0.23\ mS.cm^{-2}$ , $\bar{g}_{NMDA} = 0.35\ mS.cm^{-2}$ , $g_{GABA_A} = 0.4\ mS.cm^{-2}$ , $p_{AMPA_{FF}} = 0.101$ a.u.

## Parametric explorations in the network model

We first assessed whether variations of a single biophysical parameter could explain the differences in the temporal signature of the MCC, compared to the LPFC: an increased TAU for RS and FS units and an increased LAT for RS (but not for FS) units. To do so, we tested many biophysical parameters determining the architectural, synaptic, and intrinsic properties of the network, but none were able to account for these differences between frontal areas (not shown).

However, these explorations unraveled four model parameters of interest that were able, when varied within their physiological range (i.e. realistic regimes of network activity), to (1) affect either LAT in Exc (but not Inh) neurons, TAU in Exc neurons or TAU in Inh neurons, and (2) do so in a gradual fashion, that is, allowing some possible form of (developmental, homeostatic, or plastic) regulatory control. Indeed, several other parameters could vary TAU or LAT, but they did so abruptly, because their effects occurred at the vicinity of network bifurcations, where network activity dramatically saturated or was silenced, that is, in non-physiological regimes.

The parameters of interest were the maximal conductances, on the one hand, of two membrane ionic currents setting the intrinsic excitability and spiking pattern of cortical pyramidal neurons ($g_{CAN}$ and $g_{AHP}$) and, on the other hand, of two neurotransmitter-gated channels that set slow synaptic neurotransmission in cortical networks ($g_{NMDA}$ and $g_{GABA-B}$).

Firstly, decreasing $g_{CAN}$, the maximal conductance of the spike-triggered calcium-dependent cationic current, which controls regenerative discharge and spiking bistability in pyramidal neurons (**Haj-Dahmane and Andrade, 1997**), gradually increased LAT in Exc neurons, although this occurred at low values where the network was nearby silence and displayed very low firing frequency (**Figure 5—figure supplement 3a**, upper left). The CAN current is absent in Inh neurons (**Gorelova et al., 2002**), so that changing $g_{CAN}$ (i.e. only in Exc neurons) left LAT constant in Inh neurons (**Figure 5—figure supplement 3a**, lower left). Besides, increasing the CAN maximal conductance gradually increased TAU in Exc neurons (**Figure 5—figure supplement 3a**, upper right). This arose because $g_{CAN}$ increases burstiness, a factor that can increase the autocorrelogram time constant (**Bar-Gad et al., 2001**). However, it had no effect on TAU in Inh neurons, where it is absent (**Figure 5—figure supplement 3a**, lower right). Thus, while $g_{CAN}$ possibly accounted for LAT in frontal areas (increased LAT in MCC Exc neurons, no change in Inh neurons), as well as for the increased TAU in MCC Exc neurons, it could not explain the TAU difference in Inh neurons (**Figure 5—figure supplement 3**). Moreover, accounting for LAT and TAU in the MCC required incompatible $g_{CAN}$ ranges of values and this was the same for LPFC (**Figure 5—figure supplement 3**).

Secondly, the maximal conductance of the medium AHP current ($g_{AHP}$), a spike-triggered calcium-dependent potassium current, which balances the CAN current in the patterning of spiking in pyramidal neurons, increased LAT in Exc neurons (**Figure 5—figure supplement 3b**, upper left); this arose because AHP increases the refractory period, which can increase LAT (**Bar-Gad et al., 2001**). Similarly to CAN, the AHP current is absent in Inh neurons (**Gorelova et al., 2002**), so changing $g_{AHP}$ (i.e. in Exc neurons) left LAT constant in Inh neurons (**Figure 5—figure supplement 3b**, lower left). Besides, although $g_{AHP}$ is largely known for its effect on firing frequency adaptation, an important determinant of discharge temporal patterning, it displayed an extremely weak effect on TAU (**Figure 5—figure supplement 3b**, right). Thus, while $g_{AHP}$ possibly accounted for LAT in frontal areas (increased LAT in

MCC Exc neurons, no difference in Inh neurons), it could not explain differences in TAU (*Figure 5— figure supplement 3*).

Together, these effects of intrinsic conductances at the network scale shared important trends with those in the cellular model, inasmuch as $g_{AHP}$ increased LAT and $g_{CAN}$ increased TAU in Exc neurons.

Thirdly, the NMDA receptor maximal conductance, $g_{NMDA}$, displayed no effect on LAT (*Figure 5— figure supplement 3c*, left), but it increased TAU (*Figure 5—figure supplement 3c*, right) in Exc and, to a lesser extent, in Inh neurons, because of its slow synaptic action (decay time constant, 75 ms) on both neuronal types. However, these effects on TAU occurred at high $g_{NMDA}$ values where the network was nearby saturation and displayed unrealistic high-frequency activity. Thus, while $g_{NMDA}$ possibly accounted for TAU in frontal areas and for the absence of change in LAT in Inh neurons, it could not explain the difference in LAT in Exc neurons between LPFC and MCC (*Figure 5—figure supplement 3*).

Fourthly, the GABA$_B$ receptor maximal conductance, $g_{GABA-B}$, as for the NMDA current, displayed no effect on LAT (*Figure 5—figure supplement 3d*, left), but it increased TAU (*Figure 5—figure supplement 3d*, right) both in Exc and Inh neurons, because of its slow synaptic action (rise and decay time constants, 90 and 160 ms, respectively) on both neuronal types. Thus, while $g_{GABA-B}$ possibly accounted for TAU differences in frontal areas and for the absence of change in LAT in Inh neurons, it could not explain the difference in LAT in Exc neurons between LPFC and MCC (*Figure 5—figure supplement 3*).

Interestingly, both NMDA and GABA-B currents, which had no effect at the individual level, were essential at the network scale, suggesting that the influence of slow synaptic transmission on recurrent collective network dynamics are central in determining the time constant TAU in frontal areas.

In summary, one-dimensional network explorations showed that: (1) $g_{AHP}$ was the sole biophysical parameter that changed LAT in Exc but not in Inh, while keeping network activity within the physiological regime ($g_{CAN}$ also changed LAT in Exc, but at the border of network silencing); (2) $g_{GABA-B}$ was the sole biophysical parameter that changed TAU in both Exc and Inh ($g_{CAN}$ and $g_{NMDA}$ also changed TAU, mainly in Exc, but at the border of unrealistic regimes, that is, network silence and saturation).

Together, these results pointed to $g_{AHP}$ and $g_{GABA-B}$ as major candidates, with the idea that their combined effect in the ($g_{AHP}$, $g_{GABA-B}$) space could account for the differences in temporal signature between the LPFC and the MCC. However, because ionic and synaptic conductances typically display strong non-linear interactions whereby some forms of counter-intuitive compensatory or amplificatory effects can emerge, we nevertheless conducted two-dimensional explorations in the ($g_{AHP}$, $g_{CAN}$), ($g_{AHP}$, $g_{NMDA}$), and ($g_{AHP}$, $g_{GABA-B}$) spaces, with the idea that the relative balance of $g_{AHP}$, which affects LAT, on the one hand, and of either $g_{CAN}$, $g_{NMDA}$, or $g_{GABA-B}$, which affect TAU, on the other hand, could synergistically account for both the larger LAT and larger TAU observed in the MCC, compared to the LPFC (*Figure 5—figure supplement 1* and *Figure 6—figure supplement 1*).

Exploring the ($g_{AHP}$, $g_{CAN}$) space, we found that combined increases of $g_{AHP}$ and $g_{CAN}$ could both (1) increase LAT in Exc neurons but not in Inh neurons (*Figure 6—figure supplement 1a*, top) and (2) increase TAU in Exc neurons (*Figure 6—figure supplement 1a*, bottom), which translated, quantitatively, as two domains of smaller and larger ($g_{AHP}$, $g_{CAN}$) parameter values that displayed higher similarity to LPFC and MCC data, respectively (*Figure 6—figure supplement 1b*). However, increasing $g_{CAN}$ and $g_{AHP}$ only very weakly varied TAU in Inh neurons (as in one-dimensional explorations), one of the three major changes observed in FS units in the MCC (together with higher LAT and TAU in RS units). While this incapacity marginally reflected in the similarity measure (which integrates similarity in Exc (RS) and FS (Inh) neurons proportional to their relative abundance, that is, 0.2 for Inh neurons), the model therefore revealed qualitatively insufficient to account for the differences in LPFC and MCC temporal signatures.

The exploration of the ($g_{AHP}$, $g_{NMDA}$) space indicated a situation where combined increases of $g_{AHP}$ and $g_{NMDA}$ could increase LAT in Exc neurons but not in Inh neurons (*Figure 6—figure supplement 1c*, top) but could hardly reproduce TAU in Exc and Inh neurons (*Figure 6—figure supplement 1c*, bottom), so the qualitative agreement was weak. As a result, quantitatively, the domain of largest MCC similarity displayed modest similarities (*Figure 6—figure supplement 1d*). Moreover, as in one-dimensional explorations, $g_{NMDA}$ increased TAU mostly at high values ($g_{NMDA}$ ~ 1), where the network model was near saturation (unrealistic high frequency), while intracellular recordings show

no difference in Exc post-synaptic current amplitudes between MCC and LPFC in monkeys (*Medalla et al., 2017*).

Contrarily to explorations in ($g_{AHP}$, $g_{CAN}$) and ($g_{AHP}$, $g_{NMDA}$) spaces, exploration in the ($g_{AHP}$, $g_{GABA-B}$) provided two domains of high similarity to LPFC and MCC that indicated a strong quantitative agreement of the model to monkey data (see main text). Moreover, these domains were large (relative to the mean values of $g_{AHP}$ and $g_{GABA-B}$ in said domains) indicating robustness to the inherent biological variability present in frontal cortical structures. Finally, qualitative agreement was present, in addition to the quantitative agreement revealed by the similarity measure, in the sense that all three main qualitative differences between LPFC and MCC (with higher LAT and TAU in RS units and a higher TAU in Inh neurons) were well reproduced in this setup.

## HMM analysis

We used HMM to map the spiking set of neural network models and unit populations in monkeys onto discrete states of collective activity, based on previously established methods (*Abeles et al., 1995*; *Seidemann et al., 1996*). Monkey recordings occurred in separate sessions from which samples of typically three to four (maximum five) neurons were recorded simultaneously. HMMs were impractical on such small samples, so it was necessary to create pseudo-populations that could substitute for large simultaneous recordings of the population, to improve the amount of data leveraged and maximize the information extracted in terms of neural states. However, creating pseudo-populations ignores correlated activity between neurons, globally degrading the accuracy of state detection. In principle this degradation affects absolute quantitative state estimations in the LPFC and MCC equally. Therefore, it should not affect the conclusions drawn here, as they are based on *relative* comparisons between order of magnitudes of state estimates in the LPFC and MCC. Thus, applying HMM to large simultaneous populations would most likely not change the conclusions drawn from experimental data. This point is supported by the fact that we qualitatively obtained similar HMM results in populations and pseudo-populations in the model (see Results), as well as by the general observation that the amount of information that can be time-decoded in associative areas (such as in the LPFC) is globally the same, when using pseudo-populations or simultaneously recorded neurons (*Anderson et al., 2007*; *Meyers et al., 2008*). Extended results presented in supplementary materials were obtained using LPFC and MCC pseudo-populations built of groups of neurons drawn from simulations of distinct random neural networks (i.e. synaptic matrices), with group statistics (numbers of groups and numbers of excitatory and inhibitory neurons within each group) similar to those of monkey LPFC and MCC data.

HMM methods allow to determine the probability $p\left(S_k\left(t\right)\right)$ of the network to be in state $S_k$, $k \in \{1...n_S\}$ at time $t$. Typically, we found that, as previously shown in frontal areas, population activity organized into periods that lasted in the range $\sim 10\,\mathrm{ms} - 10\,\mathrm{s}$, that is, transition probabilities were small and states were quasi-stationary. When all probabilities of being in a state $p\left(S_k\right) < 0.8$, the network was considered to be in the null state $S_0$, signifying that the network was not in any of the states. Periods in the $S_0$ state were typically short (mean: LPFCm =16 ms, MCCm =36 ms, not shown). Thus, when immediately preceded and followed by two periods in the same state $S_k$, periods in $S_0$ were attributed the state $S_k$. For each network spiking set assessed, we pooled the durations of all periods in all the states of the HMM, to build the overall probability distribution of period durations $p\left(d\right)$. We then used this probability distribution to compute

$$p_t\left(d\right) = \frac{p\left(d\right) d}{\int_{u=0}^{+\infty} p\left(u\right) u\, du}$$

that is, the proportion of time spent in state periods of duration $d$, that is, the probability, at any given instant in time, of being in state periods of duration $d$. We could not find any generic univocal suitable method of stably determining the number of states $n_S$. However, as a low number of states is more parsimonious in terms of data interpretation (*Pohle et al., 2017*) in general and because the task structure contains a low number of possible states in terms of actions (four), reward on the last trial (incorrect trial, first correct trial, correct trial after previous correct trials) and behavioural states (exploration, exploitation), we arbitrarily fixed $n_S = 4$. We checked that this choice was sound and evaluated Akaike information criterion (AIC) and BIC values for different numbers of states in both cortical areas and in both monkey and model data (*Figure 7—figure supplement 4*). We observed

that AIC and BIC did not substantially change when varying the number of states in the range 2–10. We therefore kept the chosen value $n_S = 4$.

Each HMM analysis was conducted on a spiking set lasting 600 s, both in neural network models and unit populations in monkeys. For each monkey area, the activity of all neurons was pooled, regardless of their recording session. This was mandatory because the number of neurons simultaneously recorded in each session was typically inferior to 5, so that HMMs were inefficient in detecting states. Pooling all neurons allowed the detection of global states that corresponded to the combination of collective dynamics recorded during distinct sessions, that is, that were not time-locked together (phase information lost across sessions) and causally independent. Although chimeric, these HMM states were nevertheless able to indirectly capture the underlying temporal structure of collective spiking discharges in frontal areas in a similar way and thus allowed comparing LPFC and MCC collective temporal structure. In control HMMs, both the timing and neuron assignment of all spikes were randomly shuffled. The initial estimation of the average state duration across all periods in a given state was taken at a high value (300 ms), which was suggested to give better log-likelihood scores and converge to similar states across repetitions of the HMM (*Seidemann et al., 1996*). The time bin was $\Delta t = 0.5$ ms. We checked that state durations were not caused by eventual drifts in data firing frequency (*Figure 7—figure supplement 2*; *Figure 7—source data 2*). We also checked that state durations were distributed exponentially, lending credence to the metastable nature of HMM states (*Figure 7—figure supplement 3*).

## Principal component analysis

The principal component analysis (PCA) of LPFC and MCC of monkeys' units and neural network models' neurons spiking activity was computed from firing frequencies, in order to better visualize and characterize collective dynamics. PCA was achieved on the set of the spiking frequency vectors of all units/neurons in each case. Spiking frequency was estimated through convolution of spiking activity with a normalized Gaussian kernel with standard deviation $\sigma = 100$ ms, as average frequencies were typically $< 10$ Hz in both areas. For each neuron, frequencies were then centred and standardized for optimal PCA. Cells with average frequencies <0.5 Hz were removed for the experimental data and for the model data, to avoid abnormal standardized frequencies when the neuron's average frequency was too low (at most six cells per area).

## Perturbation protocol for state transitions

We assessed the contribution of excitatory and inhibitory neural populations to the stability of HMM states. To do so, we estimated the probability to stay in a given ongoing (or perturbed, see below) HMM state or to switch towards a distinct target (or perturbing) state in response to specified perturbations. The perturbation was achieved by substituting the value of neural variables (membrane potential, spiking state, calcium concentration, downstream channel opening probabilities) of a random subset of excitatory (respectively inhibitory) neurons of the ongoing state by those of the same neurons taken from the (distinct) target state. Specifically, starting from an initial (unperturbed) 600 s simulation, perturbations were achieved by substituting state variables 50 ms after the onset of a randomly chosen period of a specified perturbed state by those taken 50 ms after the onset of a randomly chosen period of a distinct perturbing state and the resulting network states used as initial conditions for further 'perturbation simulations'. For each perturbation simulation, the network was simulated from the perturbation time to the end of the period when the network was not perturbed and the HMM state was determined as the posterior state probability based on HMM transition and emission matrices obtained from the entire initial unperturbed simulation. The probability to escape the ongoing state (*Figure 7*, left) and to reach the target state (*Figure 7*, right) were then computed as the proportion of time spent, during the ongoing period, in a HMM state different from the ongoing perturbed state (escape ongoing state probability), and in the target perturbing state (reach target state probability), respectively. The effects of perturbations were tested by replacing either excitatory or inhibitory populations, where proportions of replaced neurons systematically varied in the range 0–1. For each neuron type and proportion tested, the perturbation protocol was applied and results averaged for 50 random combinations of periods (with period durations >100 ms), for each of the 12 possible pairs of the four HMM states (excluding pairs of repeated states), over 20 different randomly initialized MCCs. Probabilities were offset and normalized to remove the basal probability of escaping

the ongoing (0.09) and reaching the target (0.01) states when no perturbation was applied (such transitions were due to random selection of simultaneous spikes when initiating the HMM analysis).

## Acknowledgements

We thank C Nay for administrative support and C Wilson and J Naudé for helpful discussions and proofreading the manuscript.

## Additional information

### Funding

| Funder | Grant reference number | Author |
|---|---|---|
| Fondation pour la Recherche Médicale | DEQ20160334905 | Emmanuel Procyk |
| Agence Nationale de la Recherche | ANR-10-SVSE4-1441 | Emmanuel Procyk |
| Agence Nationale de la Recherche | ANR-16-NEUC-0006-01 | Bruno Delord |
| Agence Nationale de la Recherche | ANR-11-LABX-0042 | Emmanuel Procyk |
| Fondation pour la Recherche Médicale | FDT201904008187 | Vincent Fontanier |

The funders had no role in study design, data collection and interpretation, or the decision to submit the work for publication.

### Author contributions

Vincent Fontanier, Matthieu Sarazin, Conceptualization, Data curation, Formal analysis, Investigation, Methodology, Resources, Software, Validation, Visualization, Writing – original draft, Writing – review and editing; Frederic M Stoll, Investigation, Writing – review and editing; Bruno Delord, Emmanuel Procyk, Conceptualization, Data curation, Formal analysis, Funding acquisition, Investigation, Methodology, Project administration, Resources, Software, Supervision, Validation, Visualization, Writing – original draft, Writing – review and editing

### Author ORCIDs

Vincent Fontanier http://orcid.org/0000-0003-2742-0542
Frederic M Stoll http://orcid.org/0000-0002-8642-8133
Bruno Delord http://orcid.org/0000-0002-2912-3405
Emmanuel Procyk http://orcid.org/0000-0001-7486-4993

### Ethics

All procedures followed the European Community Council Directive (2010) (Ministère de l'Agriculture et de la Forêt, Commission nationale de l'expérimentation animale) and were approved by the localethical committee (Comité d'Ethique Lyonnais pour les Neurosciences Expérimentales, CELYNE, C2EA683 #42).

### Decision letter and Author response

Decision letter https://doi.org/10.7554/eLife.63795.sa1
Author response https://doi.org/10.7554/eLife.63795.sa2

## Additional files

### Supplementary files
• MDAR checklist

## Data availability

All spike time series from monkey recordings, scripts for temporal signatures extraction and scripts of computational models are freely accessible on the Zenodo repository (https://doi.org/10.5281/zenodo.5707884).

The following dataset was generated:

| Author(s) | Year | Dataset title | Dataset URL | Database and Identifier |
|---|---|---|---|---|
| Fontanier V, Sarazin M, Stoll FM, Delord B, Procyk E | 2022 | Code for spike analyses and biophysical modelling for 'Inhibitory control of frontal metastability sets the temporal signature of cognition" | https://zenodo.org/record/5707884 | Zenodo, 10.5281/zenodo.5707884 |

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
