## [Editor Report]

The paper investigates the temporal signatures of single-neuron activity (the autocorrelation timescale and latency) in two frontal areas, MCC and LPFC. These signatures differ between the two areas and cell classes, and form an anatomical gradient in MCC and, moreover, the intrinsic timescales of single neurons correspond with their coding of behaviorally relevant information on different timescales. The authors develop a detailed biophysical network model which suggests that after-hyperpolarization potassium and inhibitory GABA-B conductances may underpin the potential biophysical mechanism that explains diverse temporal signatures observed in the data. The proposed relationship between the intrinsic timescales, coding of behavioral timescales, and anatomical properties (e.g., the amount of local inhibition) in the two frontal areas is novel, the use of the biophysically detailed model is creative and interesting and the claims are convincingly supported by the data.

---

## [Decision Letter]

**Decision letter after peer review:**

Thank you for submitting your article "Inhibitory control of frontal metastability sets the temporal signature of cognition" for consideration by *eLife*. Your article has been reviewed by 2 peer reviewers, and the evaluation has been overseen by a Reviewing Editor and Timothy Behrens as the Senior Editor. The reviewers have opted to remain anonymous.

The reviewers have discussed the reviews with one another and the Reviewing Editor has drafted this decision to help you prepare a revised submission.

As the editors have judged that your manuscript is of interest, but as described below

both reviewers found this study's findings to be interesting and novel, and they appreciated the integration of intrinsic timescale analysis, coding of behavioral signals, and exploration of mechanisms in a biophysical circuit model. However, they both raised serious concerns about methodological and interpretational aspects of the analyses which would need to be satisfactorily addressed in subsequent review. In consultation, both reviewers were in agreement with all points raised in the other's review. I view the two reviewers' requests and suggestions as appropriate and complementary, and all of them needed response. Given the quality of the reviewers' comments and their consensus about what needs to be addressed, I am appending both reviewers' full reviews at the end of this letter, to guide the authors' revision and response.

I highlight here two of the concerns raised by the reviewers (but all raised by the reviewers merit addressing). First is the need for intrinsic timescale analysis to not be confounded by slowly varying changes in firing rate coding task variables (Point 1.2 of Reviewer 2). This is important for interpretation on intrinsic timescale and its correlation with task variable coding as a major result. Second is the interpretation and implementation of Hidden Markov Models to non-simultaneously recorded neurons (Points 6 of Reviewer 1 and 3 of Reviewer 2). Here again the HMM states may be driven by task variable coding which is not corrected for, which could confound interpretation of results as in terms meta-stable states and its link to the circuit model. Furthermore, HMM analysis of circuit model does not match its methodology for experimental data, but could be through non-simultaneous model spike trains, which the manuscript does not justify. I will add my own suggestion that perhaps both of these methodological data analysis concerns could potentially be addressed through comparison to null model of a non-homogeneous Poisson process with firing rate given by the variable-coding PSTH. I do not consider it necessary to study a network model that performs the task, as suggested for consideration by Reviewer 1.

*Reviewer #1:*

This is an interesting manuscript which covers an important topic in the field of computational neuroscience – the 'temporal signatures' of individual neurons. The authors set out to address several important questions using a single-neuron electrophysiology dataset, recorded from monkeys, which has previously been published. The behavioural paradigm is well designed, and particularly well suited to investigating the functional importance of different temporal signatures – as it simultaneously requires the subjects to monitor feedback across a short timescale, as well as integrate multiple outcomes across a longer timescale. The neural data are of high quality, and include recordings from lateral prefrontal cortex (LPFC) and mid-cingulate cortex (MCC). First, the authors modify an existing method to quantify the temporal signatures of individual neurons. This modification appears helpful, and an improvement on similar previously published methods, as the authors are able to capture the temporal signatures of the vast majority of neurons they recorded from. The temporal signatures differ across brain regions, and according to the neurons' spike width. The authors argue that the temporal signatures of a subset of neurons are modified by the subjects' degree of task engagement, and that neurons with different temporal signatures play dissociable roles in task-related encoding. However, I have several concerns about these conclusions which I will outline below. The authors then present a biophysical network model, and show that by varying certain parameters in their model (AHP and GABAB conductances) the temporal signatures of the monkey data can be reproduced. Although I cannot comment on the technical specifics of their models, this seems to be an important advance. Finally, they perform a Hidden-Markov Model analysis to investigate the metastability of activity in MCC, LPFC, and their network model. However, there are a few important differences between the model and experimental data (e.g. neurons recorded asynchronously, and the network model not performing a task) that limit the interpretation of these analyses. Overall, I found the manuscript interesting – and the insights from the biophysical modelling are exciting. However, in its current form, the conclusions drawn from the experimental data are not supported by sufficient evidence.

1. The authors use a hierarchical clustering algorithm to divide neurons into separate groups according to their spike width and amplitude (Figure 1C). There are three groups: FS, RS1, and RS2. The authors ultimately pool RS1 and RS2 groups to form a single 'RS' category. They then go on to suggest that RS neurons may correspond to pyramidal neurons, and FS neurons to interneurons. I have a few concerns about this. Firstly, the suggestion that spike width determined from extracellular recordings in macaques can be used as an indicator of cell type is controversial. A few studies have presented evidence against this idea (e.g. Vigneswaran et al. 2011 JNeuro; Casale et al. 2015 JNeuro). The authors should at least acknowledge the limitation of the inference they are making in the Discussion section. Secondly, visualising the data alone in Figure 1C, it is far from clear that there are three (or two) relatively distinct clusters of neurons to warrant treating them differently in subsequent analyses. In the methods section, the authors mention some analyses they performed to justify the cluster boundaries. However, this data is not presented. A recent study approached this problem by fitting one gaussian to the spike waveform distribution, then performing a model comparison to a 2-gaussian model (Torres-Gomez et al. 2020 Cer Cortex). Including an analysis such as this would provide a stronger justification for their decision to divide cells based on spike waveform.

2. The authors conclude that the results in Figure 3 show that MCC temporal signatures are modulated by current behavioural state. However, this conclusion seems a bit of a stretch from the data currently presented. I can understand why the authors used the 'pause' periods as a proxy for a different behavioural state, but the experiment clearly was not designed for this purpose. As the authors acknowledge, there is only a very limited amount (e.g. a few minutes) of 'pause' data available for the fitting process compared with 'engage' data. Do the authors observe the same results if they constrain the amount of included 'engage' data to match the length of the 'pause' data? Also, presumably the subjects are more likely to 'pause' later on in the behavioural session once they are tired/sated. Could this difference between 'pause' and 'engage' data be responsible for the difference in taus? For instance, there may have been more across-session drift in the electrode position by the time the 'pause' data is acquired, and this could possibly account for the difference with the 'engage' data. Is the firing rate different between 'pause' and 'engage' periods – if so, this should be controlled for as a covariate in the analyses. Finally, it is not really clear to me, or more importantly addressed by the authors, as to why they would expect/explain this effect only being present in MCC RS neurons (but not FS or LPFC neurons).

3. At many points in the manuscript, the authors seem to be suggesting that the results of Figure 4 demonstrate that neurons with longer (shorter) timescales are more involved in encoding the task information which is used across longer (shorter) behavioural timescales (e.g. "long TAU were mostly involved in encoding gauge information", and "population of MCC RS units with short TAU was mostly involved in encoding feedback information"). However, I disagree that this conclusion can be reached based on the way the analysis has currently been performed. A high coefficient simply indicates that the population is biased to be more responsive depending on a particular trial type / condition – i.e. the valence of encoding. This does not necessarily tell us how much information the population of neurons is encoding, as the authors suggest. For instance, every neuron in the population could be extremely selective to a particular parameter (i.e. positive feedback), but if half the neurons encode this attribute by increasing their firing but the other half of neurons encode it by decreasing their firing, the effects will be lost in the authors' regression model (i.e. the β coefficient would equal 0). I would suggest that the authors consider using an alternative analysis method (e.g. a percentage of explained variance or coefficient of partial determination statistic for each neuron) to quantify coding strength – then compare this metric between the high and low tau neurons.

4. Similarly, in Figure 4 the authors suggest that the information is coded differently in the short and long tau neurons. However, they do not perform any statistical test to directly compare these two populations. One option would be to perform a permutation test, where the neurons are randomly allocated into the 'High TAU' or 'Low TAU' group. A similar comment applies to the different groups of neurons qualitatively compared in panel Figure 4C.

5. The authors make an interesting and well-supported case for why changing the AHP and GABA-B parameters in their model may be one mechanism which is sufficient to explain the differences in temporal signatures they observed between MCC and LPFC experimentally. However, I think in places the conclusions they draw from this are overstated (e.g. "This suggests that GABA-B inhibitory – rather than excitatory – transmission is the causal determinant of longer spiking timescales, at least in the LPFC and MCC."). There are many other biophysical differences between different cortical regions – which are not explored in the authors' modelling – which could also account for the differences in their temporal signatures. These could include differences in extra-regional input, the position of the region in an anatomical hierarchy, proportion of excitatory to inhibitory neurons, neurotransmitter receptor/receptor subunit expression, connectivity architecture etc. I think the authors should tone down the conclusions a little, and address some more of these possibilities in more detail in their discussion.

6. For the Hidden Markov Model, I think there are a couple of really important limitations that the authors only touch upon very briefly. Firstly, the authors are performing a population-level analysis on neurons which were not simultaneously recorded during the experiment (only mentioned in the methods). This really affects the interpretation of their results, as presumably the number of states and their duration is greatly influenced by the overall pattern of population activity which the authors are not able to capture. At this stage of the study, I am not sure how the authors can address this point. Secondly, the experimental data is compared to the network model which is not performing any specific task (i.e. without temporal structure). The authors suggest this may be the reason why their predictions for the state durations (Fig7b) are roughly an order of magnitude out. Presumably, the authors could consider designing a network model which could perform the same task (or a simplified version with a similar temporal structure) as the subjects perform. This would be very helpful in helping to relate the experimental data to the model, and may also provide a better understanding of the functional importance of the metastability they have identified in behaviour.

7. It is not clear to me how many neurons the authors included in their dataset, as there appear to be inconsistencies throughout the manuscript (Line 73, Figure 1A-b: MCC = 140, LPFC = 159; line 97-98: MCC = 294, LPFC=276; Figure 2: MCC = 266, LPFC = 258; Methods section line 734-735 and Fig2S2: MCC = 298, LPFC = 272). While this is likely a combination of typos and excluding some neurons from certain analyses, this will need to be resolved. It will be important for the authors to check their analyses, and also add a bit more clarity in the text as to which neurons are being included/excluded in each analysis, and justify this.

*Reviewer #2:*

The paper investigates the temporal signatures of single-neuron activity (the autocorrelation timescale and latency) in two frontal areas, MCC and LPFC. These signatures differ between the two areas and cell classes, and form an anatomical gradient in MCC. Moreover, the intrinsic timescales of single neurons correspond with their coding of behaviorally relevant information on different timescales. The authors develop a detailed biophysical network model which suggests that after-hyperpolarization potassium and inhibitory GABA-B conductances may underpin the potential biophysical mechanism that explains diverse temporal signatures observed in the data. The results appear exciting, as the proposed relationship between the intrinsic timescales, coding of behavioral timescales, and anatomical properties (e.g., the amount of local inhibition) in the two frontal areas is novel. The use of the biophysically detailed model is creative and interesting. However, there are serious methodological concerns undermining the key conclusions of this study, which need to be addressed before the results can be credited.

1. One of the key findings is the correspondence between the intrinsic timescales of single neurons and their coding of information on different behavioral timescales (Figure 4). However, the method for estimating the intrinsic timescales has serious problems which can undermine the finding.

1.1. The authors developed a new method for estimating autocorrelograms from spike data but the details of this method are not specified. It is stated that the method computes the distribution of inter-spike-intervals (ISIs) up to order 100, which was "normalized", but how it was normalized is not described. The correct normalization is crucial, as it converts the counts of spike coincidences (ISI distribution) into autocorrelogram (where the coincidence counts expected by chance are subtracted) and can produce artifacts if not performed correctly.

1.2. The new method, described as superior to the previous method by Murray et al., 2014, appears to have access to more spikes than the Murray's method (Figure 2). Where is this additional data coming from? While Murray's method was applied to the pre-cue period, the time epoch used for the analysis with the new method is not stated clearly. It seems that the new method was applied to the data through the entire trial duration and across all trials, hence more spikes were available. If so, then changes in firing rates related to behavioral events contribute to the autocorrelation, if not appropriately removed. For example, the Murray's method subtracts trail-averaged activity (PSTH) from spike-counts, similar to shuffle-correction methods. If a similar correction was not part of the new method, then changes in firing rates due to coding of task variables will appear in the autocorrelogram and estimated timescales. This is a serious confound for interpretation of the results in Figure 4. For example, if the firing rate of a neuron varies slowly coding for the gauge size across trials, this will appear as a slow timescale if the autocorrelogram was not corrected to remove these rate changes. In this case, the timescale and GLM are just different metrics for the same rate changes, and the correspondence between them is expected. Before results in Figure 4 can be interpreted, details of the method need to be provided to make sure that the method measures intrinsic timescales, and not timescales of rate changes triggered by the task events. This is an important concern also because recent work showed that there is no correlation between task dependent and intrinsic timescales of single neurons, including in cingulate cortex and PFC (Spitmaan et al., PNAS, 2020).

2. The balanced network model with a variety of biophysical currents is interesting and it is impressive that the model reproduces the autocorrelation signatures in the data. However, we need to better understand the network mechanism by which the model operates.

2.1. The classical balanced network (without biophysical currents such as after-hyperpolarization potassium) generates asynchronous activity without temporal correlations (Renart et al., Science, 2010). The balanced networks with slow adaptation currents can generate persistent Up and Down states that produce correlations on slow timescales (Jercog et al., *eLife*, 2017). Since slow after-polarization potassium current was identified as a key ingredient, is the mechanism in the model similar to the one generating Up and Down states, or is it different? Although the biophysical ingredients necessary to match the data were identified, the network mechanism has not been studied. Describing this network mechanism and presenting the model in the context of existing literature is necessary, otherwise the results are difficult to interpret for the reader.

2.2. Does the model operate in a physiologically relevant regime where the firing rates, Fano factor etc. are similar to the data? It is hard to judge from Figure 5b and needs to be quantified.

2.3. The latency of autocorrelation is an interesting feature in the data. Since the model replicates this feature (which is not intuitive), it is important to know what mechanism in the model generates autocorrelation latency.

3. HMM analysis is used to demonstrate metastability in the model and data, but there are some technical concerns that can undermine these conclusions.

3.1. HMM with 4 states was fitted to the data and model. The ability to fit a four-state HMM to the data does not prove the existence of metastable states. HMM assumes a constant firing rate in each "state", and any deviation from this assumption is modeled as state transitions. For example, if some neurons gradually increase/decrease their firing rates over time, then HMM would generate a sequence of states with progressively higher/lower firing rates to capture this ramping activity. In addition, metastability implies exponential distributions of state durations, which was not verified. No model selection was performed to determine the necessary number of states. Therefore, the claims of metastable dynamics are not supported by the presented analysis.

3.2. HMM was fit to a continuous segment of data lasting 600s, and the data was pooled across different recording sessions. However, different sessions have potentially different trial sequences due to the flexibility of the task. How were different trial types matched across the sessions? If trial-types were not matched/aligned in time, then the states inferred by the HMM may trivially reflect a concatenation of different trial types in different sessions. For example, the same time point can correspond to the gauge onset in one session and to the work trial in another session, and vice versa at a different time. If some neurons respond to the gauge and others to the work, then the HMM would need different states to capture firing patterns arising solely from concatenating the neural responses in this way. This confound needs to be addressed before the results can be interpreted.

---

## [Author Response]

I highlight here two of the concerns raised by the reviewers (but all raised by the reviewers merit addressing).First is the need for intrinsic timescale analysis to not be confounded by slowly varying changes in firing rate coding task variables (Point 1.2 of Reviewer 2). This is important for interpretation on intrinsic timescale and its correlation with task variable coding as a major result.

We agree with the reviewers that the relationship between firing rate and timescale variation is not trivial and could constitute an important confound. We tackled this concern in the initial submission (*Figure 2 —figure supplement 2*) by showing that there is no linear correlation between the firing rate and TAUs. However, and as pointed out by the reviewers, this relationship depends on how the firing rate is calculated. In the analysis reported in the initial manuscript, we aimed to extract the firing rate when the units are the most active as we thought it would be a good measure of the physiological firing rate and this would avoid average issues. In response to the reviewers, we decided to go further into this analysis and we described TAUs and firing rate relationships between different recording periods, when monkeys are engaged, taking breaks or around key task events. We found little correlation around key task events (feedback and decision). These however were not reliable as segments used for data were short (2s for feedback, 4s for decision) resulting in noisy ISI computations leading to less spike-autocorrelogram being fitted and more variability in TAUs.

Conversely there were no limitations in firing rate computation (number of spikes given a duration) for larger segments (data during pauses or engage periods, or the whole data set). Using such large segments, we showed that there was no linear correlation between the firing rates computed at feedback or during decisions and TAUs extracted from the whole recording:

Consistent with the idea that TAUs measures are stable and descriptive of a unit’s properties, we found that TAUs extracted from different periods were quite correlated with each other, when data allowed for such correlations. Also, as pointed out in the manuscript, TAUs extracted from the spike autocorrelogram and from the spike-count method were correlated across units for which both measures were available.

The spike-count method to measure TAUs requires a repetition of task trials from which an autocorrelation can be computed between each time bin. In their supplementary materials, Murray and colleagues (Murray et al. 2014) demonstrated that the autocorrelation subtracts the mean spike-count of each time bin, thus controlling for firing rate fluctuation. Nevertheless, this does not remove within trials fluctuations of firing rate. In fact in our dataset we found a slight correlation between scTAUs and firing rate extracted from the precue period (p-value = 0.031, rho=0.14):

**Author response image 1. sa2fig1:** 

Reviewer #2 suggested to remove the average firing rates (across trials) by normalization when extracting TAUs in the spike autocorrelogram. This however might not be relevant as the spike autocorrelogram method is not trial-based but based on the ISIs distribution. Some confusion might arise from the term autocorrelation which refers to a true mathematical autocorrelation in the spikecount method, whereas it refers to the construction of spike autocorrelograms from the ISIs distribution in our method. We clarify the description of the method in the Result section.In conclusion, firing rate in our analyses appears to not be a confound for the analyses performed in this paper. We provide other more specific responses to reviewers below.

Second is the interpretation and implementation of Hidden Markov Models to non-simultaneously recorded neurons (Points 6 of Reviewer 1 and 3 of Reviewer 2). Here again the HMM states may be driven by task variable coding which is not corrected for, which could confound interpretation of results as in terms meta-stable states and its link to the circuit model. Furthermore, HMM analysis of circuit model does not match its methodology for experimental data, but could be through non-simultaneous model spike trains, which the manuscript does not justify. I will add my own suggestion that perhaps both of these methodological data analysis concerns could potentially be addressed through comparison to null model of a non-homogeneous Poisson process with firing rate given by the variable-coding PSTH. I do not consider it necessary to study a network model that performs the task, as suggested for consideration by Reviewer 1.

We have followed both of your propositions, regarding non-simultaneous model spike trains (i.e. built and run HMM analyses of pseudo-populations on model data) and task variable-coding (i.e. using non-homogeneous Poisson with firing rate given by the task variable-coding PSTH) to disambiguate these possible issues. These additional analyses clearly show that (1) pseudo-populations in experimental data vs populations in the model is not an issue in drawing conclusions on the fact that neural states display larger durations in the MCC, compared to the LPFC in both monkey and model activity data, and (2) task variable coding is not responsible for longer meta-stables states in the MCC (compared to the LPFC) in monkey data (see *Specific responses to reviewers*).

Reviewer #1:This is an interesting manuscript which covers an important topic in the field of computational neuroscience – the 'temporal signatures' of individual neurons. The authors set out to address several important questions using a single-neuron electrophysiology dataset, recorded from monkeys, which has previously been published. The behavioural paradigm is well designed, and particularly well suited to investigating the functional importance of different temporal signatures – as it simultaneously requires the subjects to monitor feedback across a short timescale, as well as integrate multiple outcomes across a longer timescale. The neural data are of high quality, and include recordings from lateral prefrontal cortex (LPFC) and mid-cingulate cortex (MCC). First, the authors modify an existing method to quantify the temporal signatures of individual neurons. This modification appears helpful, and an improvement on similar previously published methods, as the authors are able to capture the temporal signatures of the vast majority of neurons they recorded from. The temporal signatures differ across brain regions, and according to the neurons' spike width. The authors argue that the temporal signatures of a subset of neurons are modified by the subjects' degree of task engagement, and that neurons with different temporal signatures play dissociable roles in task-related encoding. However, I have several concerns about these conclusions which I will outline below. The authors then present a biophysical network model, and show that by varying certain parameters in their model (AHP and GABAB conductances) the temporal signatures of the monkey data can be reproduced. Although I cannot comment on the technical specifics of their models, this seems to be an important advance. Finally, they perform a Hidden-Markov Model analysis to investigate the metastability of activity in MCC, LPFC, and their network model. However, there are a few important differences between the model and experimental data (e.g. neurons recorded asynchronously, and the network model not performing a task) that limit the interpretation of these analyses. Overall, I found the manuscript interesting – and the insights from the biophysical modelling are exciting. However, in its current form, the conclusions drawn from the experimental data are not supported by sufficient evidence.

We thank the reviewer for their clear appreciation of the gains provided by our study. We would like first to note that the dataset presented in the current study is an extension of the previous one published in Stoll et al. 2016, containing additional recordings from the same animals.

1. The authors use a hierarchical clustering algorithm to divide neurons into separate groups according to their spike width and amplitude (Figure 1C). There are three groups: FS, RS1, and RS2. The authors ultimately pool RS1 and RS2 groups to form a single 'RS' category. They then go on to suggest that RS neurons may correspond to pyramidal neurons, and FS neurons to interneurons. I have a few concerns about this. Firstly, the suggestion that spike width determined from extracellular recordings in macaques can be used as an indicator of cell type is controversial. A few studies have presented evidence against this idea (e.g. Vigneswaran et al. 2011 JNeuro; Casale et al. 2015 JNeuro). The authors should at least acknowledge the limitation of the inference they are making in the Discussion section.

We thank the reviewer for this comment which indeed raised an important point. We acknowledge that there is no one to one relationship between spike shape and cell type and that although several studies presented data in that sense, others, sometimes in different structures, showed that in some instances Fast Spiking (FS) units could be related to pyramidal cells. Also, several excitatory and inhibitory cell types can be found in the primate prefrontal cortex, with possibly different functional roles (Wang Nat Neursc Rev 2020), making the picture more complicated than just a binary dissociation. Yet, Trainito and colleagues found that (from a database of 2500 waveforms) several waveform clusters (4) could be dissociated and linked to different coding properties (Trainito et al. 2019 Curr Biol.) confirming that, using one of the only accessible cell specific features from extracellular recordings, one might extract some basic biological characteristics. Our more limited dataset revealed 3 clusters. Their data and the literature suggest that ‘spike waveform is a sufficiently sensitive and specific marker to dissociate more than two cell classes from extracellular recordings’ (Trainito et al. 2019). Other past studies have succeeded in separating FS and RS units using waveforms (in particular spike width) and observing clustered physiological properties (Nowak et al. 2003 J Neurophysiol, Krimer et al. 2005 J Neurophysiol). Combining spike width to other physiological measures obtained in vitro or with intracellular recordings are without doubt much better for isolating FS and RS. Krimer et al. (2005) found that 100% of pyramidal neurons were RS and 100% of chandelier cells (gabaergic interneurons) were FS, but some other intermediate interneuron types were found to be either RS or FS. Unfortunately, one can’t yet access such measures in behaving monkeys. Although spike waveform is a quite good parameter to separate unit types, it is not perfect. In our dataset a few elements suggest this dissociation might be relevant. First the proportion of FS and RS units observed in our database is 20/80 % which fits relatively well with the estimated proportions of interneurons vs excitatory neurons. Also, as mentioned in the litterature, the firing rate is slightly higher on average for interneurons than for pyramidal neurons, as in our models, and as observed for FS and RS units.

We now acknowledge the inherent pitfalls but also the potential of spike waveform clustering in the text and provide references according to the reviewer’s point (see Results, page 4). We also reformulated the text to clarify that we only hypothesized such equivalence for modeling purposes knowing this is a simplification, as in our spike segregation and modeling only 2 cell types were considered.

Secondly, visualising the data alone in Figure 1C, it is far from clear that there are three (or two) relatively distinct clusters of neurons to warrant treating them differently in subsequent analyses. In the methods section, the authors mention some analyses they performed to justify the cluster boundaries. However, this data is not presented. A recent study approached this problem by fitting one gaussian to the spike waveform distribution, then performing a model comparison to a 2-gaussian model (Torres-Gomez et al. 2020 Cer Cortex). Including an analysis such as this would provide a stronger justification for their decision to divide cells based on spike waveform.

We agree with the reviewers on the confusion about the clustering methods and the lack of justification for the number of clusters to be formed. For the sake of clarity, we decided to remove from the methods section the mention of analyses (PCA procedure) that were not shown as they were redundant with the hierarchical clustering. We now use the method from Torres-Gomez et al. 2020 Cereb Cortex suggested by the reviewer in order to obtain an objective statistical number of clusters formed from the spike widths distribution. We discuss this further below and have modified the methods section accordingly (see *Spike Shape Clustering*).

Furthermore, we decided not to use the firing rate for the clustering because we did not have a clear justification for choosing a specific period for firing rate calculation. Yet we found that the so-called FS population we extracted had a higher firing rate than the RS population (Figure 2 —figure supplement 2a). This difference is in adequation with the literature and supported our decision to cluster units solely based on spike shape/duration. This difference of firing rate is reported when computed from the whole recording. Firing rate computed from different periods of the recordings (when monkeys are engaged in the task, taking a break or around key task events…) are correlated but variable. Actually, in our recordings the correlation is lowest when considering firing rates between pauses and the foreperiod of the task, two of the periods which could have been logical candidates for a firing rate of reference.

**Author response image 2. sa2fig2:** Firing rate correlations computed across different periods.

We finally clustered units according to their spike width and V2/PiK. We first computed the spike width vs. V2/PiK Euclidean distance matrix (dist function in R). Then we performed hierarchical clustering using Ward’s method (hclust function in R). The number of retained clusters was statistically confirmed by fitting one-dimensional gaussian mixture models from 1 to 3 models with variable/unequal variance on the spike widths distribution using the function Mclust from the package Mclust in R which uses the expectation-maximization algorithm. The model that best fitted the spike widths distribution was composed of 3 gaussians (as shown in figure 1c).

2. The authors conclude that the results in Figure 3 show that MCC temporal signatures are modulated by current behavioural state. However, this conclusion seems a bit of a stretch from the data currently presented. I can understand why the authors used the 'pause' periods as a proxy for a different behavioural state, but the experiment clearly was not designed for this purpose. As the authors acknowledge, there is only a very limited amount (e.g. a few minutes) of 'pause' data available for the fitting process compared with 'engage' data. Do the authors observe the same results if they constrain the amount of included 'engage' data to match the length of the 'pause' data? Also, presumably the subjects are more likely to 'pause' later on in the behavioural session once they are tired/sated. Could this difference between 'pause' and 'engage' data be responsible for the difference in taus? For instance, there may have been more across-session drift in the electrode position by the time the 'pause' data is acquired, and this could possibly account for the difference with the 'engage' data. Is the firing rate different between 'pause' and 'engage' periods – if so, this should be controlled for as a covariate in the analyses. Finally, it is not really clear to me, or more importantly addressed by the authors, as to why they would expect/explain this effect only being present in MCC RS neurons (but not FS or LPFC neurons).

Indeed, pauses are a natural phenomena that reflects the animal's voluntary decision to engage or not. The distribution of pauses is indeed biased toward the second half of a session. We now provide a figure for both monkeys (Figure 3b). The across session drift is possible but unlikely to explain everything as we checked for each session the stability of spike shape and the approximate firing rate for each unit. A time on task effect is also possible and this is indeed a very interesting suggestion. Other data from our lab revealed time-on-task changes in frontal neural activity at the level of β oscillation power (Stoll et al. 2016) and at the level of LFP and unit activity (Goussi et al., in preparation).

To control for a time-on-task effect on timescale modulation, we now contrast pause periods with engaged periods that occurred at similar times within sessions (i.e. considering only engaged periods occuring after the first pause, so excluding the first segment of data where no pause occurred – See limits in figure 3b, yellow marks; see changes in Methods – Behaviour and context-dependant modulations).

Because engage periods were as long as pause periods for one monkey (Monkey H, 53 sessions, MDengage=392s, MDpause=396s, Wilcoxon-paired test: V=790, p=0.51) and roughly twice as long for the other (Monkey A, 96 sessions, MDengage=638s, MDpause=372s, Wilcoxon-paired test: V=406, p=2.19e-12), we decided not to further segment the data to avoid resampling biases. The contrast between TAU engage and pause led to results similar to those already reported in the article, with increased TAUs in engage relative to pause periods only in MCC RS population, thus discarding a confounding effect of time on task (nMCCFS=19, nLPFCFS=21, nMCCRS=80, nLPFCRS=97, Wilcoxon signed-ranks test with Bonferroni correction, only significant for MCC RS: MD=1.06, V=2467, p=3.9e-7). Because this analysis is based on a subpart of recordings, the number of units with correctly extracted timescale for those periods is not sufficient to conduct appropriate statistical tests separately for the two monkeys (several groups have n<10). We have however verified that the distribution of TAUs extracted from the whole recordings are similar for the two monkeys, with longer TAUs in the MCC (Linear model on BLOM transformed TAU, region x unit type, Monkey A (332 units), interaction: ns, unit type: t=1,89, p=0,059, region: t=-3,25, p=0,0013 ; Monkey H (189 units), interaction: ns, unit type: ns, region: t=-2,60, p=0,010).

The reviewer finally asked whether one would have an interpretation on why such effect engage/pause is observed for RS units in MCC and not for FS or for LPFC units. We do not have a definitive answer but the differentiation might be related to multiple factors including the large range of timescales available for RS units in MCC (compared to FS or LPFC RS units) which might be regulated by specific mechanisms including neuromodulation by noradrenaline or dopamine. The differential density of neuromodulatory fibers in MCC versus LPFC might be an influential factor. Finally, the different data for LPFC compared to MCC might also relate to the areas covered by recordings which include the most posterior parts of the LFPC (described in Stoll et al. 2016).

3. At many points in the manuscript, the authors seem to be suggesting that the results of Figure 4 demonstrate that neurons with longer (shorter) timescales are more involved in encoding the task information which is used across longer (shorter) behavioural timescales (e.g. "long TAU were mostly involved in encoding gauge information", and "population of MCC RS units with short TAU was mostly involved in encoding feedback information"). However, I disagree that this conclusion can be reached based on the way the analysis has currently been performed. A high coefficient simply indicates that the population is biased to be more responsive depending on a particular trial type / condition – i.e. the valence of encoding. This does not necessarily tell us how much information the population of neurons is encoding, as the authors suggest. For instance, every neuron in the population could be extremely selective to a particular parameter (i.e. positive feedback), but if half the neurons encode this attribute by increasing their firing but the other half of neurons encode it by decreasing their firing, the effects will be lost in the authors' regression model (i.e. the β coefficient would equal 0). I would suggest that the authors consider using an alternative analysis method (e.g. a percentage of explained variance or coefficient of partial determination statistic for each neuron) to quantify coding strength – then compare this metric between the high and low tau neurons.

We now supplement the population coding analysis with a more common single-unit coding analysis as reported in Stoll et al. 2016. The detail is in the Materials and methods section ‘Task-related analyses – Spike analyses’ and results are in figure 4 supplement 1. As suggested by the reviewer, we used the weighted proportion of variance explained (wPEV) as a statistical measure to quantify the extent to which the variability in neural firing rate was determined by feedback valence and gauge state. We then quantified the time-resolved proportion of cells coding for the task variables (p<0.05 for at least 4 successive bins) and the wPEV in populations of units with short or long TAU (median split by cell type). To assess differences between short and long TAU populations we built null distributions by permuting 1000 times the TAU group allocation of units. We then compared the position of the true data relative to the cumulative distribution of the permutations and set a statistical threshold to a=0.05. The ‘single-unit’ analyses overall revealed similar dynamics for RS long TAU units in MCC and for FS involvement in Feedback than the population coding shown in the main Figure 4 (see Figure 4 supplement 1). It thus suggests that the glmm population analysis conserved some freedom (thanks to Unit as a Random term) to fit the different changes in firing rates. The single-unit analysis however, while revealing (as for the glmm) a higher contribution of long TAU RS MCC units at feedback onset, did not show a bias for short TAU RS for feedback encoding during inter-trial. Thus the two approaches provided complementary outcomes that we now summarize in the Results section (see Result section Temporal signatures are linked to cognitive processing):

“We used data from the whole recordings (all periods) and classed both FS and RS units as either short or long TAU units using a median split. We used a time-resolved generalized mixed linear models (glmm; Figure 4a) to reveal notable dissociations between these populations that we complemented using a more classical approach at the single unit level, using Poisson glm and weighted proportion of variance explained (wPEV; Figure 4 – supplement 1).

Early phases of feedback encoding recruited MCC long TAU populations for both FS and RS units (Figure 4a, upper graphs). This discrepancy was confirmed by a difference between early feedback coding in short and long TAU population at the single unit level (Figure 4 —figure supplement 1). Interestingly FS units in the MCC were mostly engaged in the first second after feedback onset, with a strong bias toward encoding negative feedback (Figure 4a, upper left, positive estimates). Effects were more transient and involved short TAU units in the LPFC (Figure 4a).

During the intertrial interval, feedback valence was represented in different directions between short and long TAU RS populations in MCC, coding being positive for short TAU populations (higher activity for incorrect feedback) and becoming negative for long TAU populations (higher activity for correct feedback). Conversely, solely the population of MCC long TAU RS units coded for the gauge during the intertrial period (Figure 4a, lower graphs). Single unit analyses confirmed the higher contribution of the long TAU population to gauge encoding (Figure 4 – supplement 1).”

Altogether, the previous and new analyses point to differentiations between short/long neuronal timescales and the encoding of behavioral information.

4. Similarly, in Figure 4 the authors suggest that the information is coded differently in the short and long tau neurons. However, they do not perform any statistical test to directly compare these two populations. One option would be to perform a permutation test, where the neurons are randomly allocated into the 'High TAU' or 'Low TAU' group. A similar comment applies to the different groups of neurons qualitatively compared in panel Figure 4C.

We thank the reviewer for this suggestion. To assess how short and long TAU population codings for a given area and cell type differ from null, we have now constructed null distributions of coding (zvalues) by permuting TAU group allocation of units. We performed the permutations for each area and cell type 100 times, this allowed us to conserve differences in sampling (e.g. the population of MCC RS with long TAU is larger than the short TAU one).

We then compare the position of the true data relative to the cumulative distribution of the permutations and set a statistical threshold a=0.05. This is shown now in panel (a) in figure 4, significant statistics being represented by rasters below each curve.

The permutation tests revealed results complementary to the more direct *glmm* modeling and to wPEV analyses. Glmm and wPEV show that MCC FS cells encode feedback early after its onset, and earlier for long TAU FS than short ones. Note however that the glmm results do differ from the null distributions. The effects of gauge encoding by MCC RS cells is observed for long TAU compared to short Tau cells in a period just before lever onset (glmm and wPEV). Comparing glmm effects to null distribution reveal effects outside of this period.

5. The authors make an interesting and well-supported case for why changing the AHP and GABA-B parameters in their model may be one mechanism which is sufficient to explain the differences in temporal signatures they observed between MCC and LPFC experimentally. However, I think in places the conclusions they draw from this are overstated (e.g. "This suggests that GABA-B inhibitory – rather than excitatory – transmission is the causal determinant of longer spiking timescales, at least in the LPFC and MCC."). There are many other biophysical differences between different cortical regions – which are not explored in the authors' modelling – which could also account for the differences in their temporal signatures. These could include differences in extra-regional input, the position of the region in an anatomical hierarchy, proportion of excitatory to inhibitory neurons, neurotransmitter receptor/receptor subunit expression, connectivity architecture etc. I think the authors should tone down the conclusions a little, and address some more of these possibilities in more detail in their discussion.

We thank the reviewer for pointing out additional mechanisms which could play a role in the differences in temporal signatures. Regarding excitation transmission parameters, exhaustive 1D and 2D explorations (i.e. AMPA and NMDA maximal conductance and time constants) did not allow to find a space region accounting for network activity with realistic statistics and the whole pattern of temporal signatures (timescale and latency of the autocorrelogram in the LPFC and MCC). This was the same for all other parameters or combinations of parameters tested, except for the GABA-B / AHP plane, where it was possible to find such regions in a robust manner. This contrast was striking in our explorations. Nevertheless, we have toned down our conclusion (“Local molecular basis of frontal temporal signatures’ subsection) and adopted a more nuanced formulation. We have also introduced all of the other dimensions rightfully proposed (extra-regional input, position in the hierarchy, etc.; “Local molecular basis of frontal temporal signatures” subsection).

6. For the Hidden Markov Model, I think there are a couple of really important limitations that the authors only touch upon very briefly. Firstly, the authors are performing a population-level analysis on neurons which were not simultaneously recorded during the experiment (only mentioned in the methods). This really affects the interpretation of their results, as presumably the number of states and their duration is greatly influenced by the overall pattern of population activity which the authors are not able to capture. At this stage of the study, I am not sure how the authors can address this point.

We applied Hidden Markov Models (HMMs) to the activity of populations of neurons that were not recorded simultaneously (i.e. pseudo-populations) to analyze overall LPFC and MCC dynamics in terms of neural states for three reasons. First, HMMs were impractical when limited to the groups of simultaneously recorded monkey neurons, as that contained too few neurons (3 or 4 in general, 5 at maximum); therefore, working on pseudo-populations was the best way to improve the amount of data employed and to maximize the information extracted in terms of neural states. Second, whereas working on pseudo-populations degrades quantitative estimations of states (by ignoring a part of correlated activity between neurons), this effect in principle affects absolute quantitative state estimations in the LPFC and MCC equally and likely does not affect the conclusions drawn here, as they are based on relative comparisons between order of magnitudes of state estimates in the LPFC and MCC. Third, previous studies have shown that the amount of information that can be time-decoded in associative areas (such as in the LPFC) is globally similar when using pseudo-populations or simultaneously recorded neurons (Anderson et al., Exp Brain Res 2007; Meyers et al., J Neurophysiol. 2008). We acknowledge that these considerations were absent in the original manuscript and have now introduced them in the Materials and methods (“Hidden Markov Model (HMM) analysis” subsection).

In our results, a similar trend was observed in both monkey and model neural activity: the duration of states detected by HMM was larger in the MCC, compared to the LPFC, with a MCC/LPFC ratio of mean state durations of ~30 in model populations and of ~7.5 in monkey pseudo-populations. To better interpret and strengthen our results, with respect to the question raised by reviewer’s #1, we have now additionally applied HMM analysis to neural activity in the model, using pseudo-populations built of groups of neurons drawn from LPFC and MCC simulations of distinct random neural networks, with group statistics (number of recording sessions and numbers of simultaneously recorded excitatory and inhibitory neurons within each recording session) similar to those of monkey LPFC and MCC data (Figure 7 – supplement 1). The rationale here is to examine whether a higher state duration MCC/LPFC ratio is still present, i.e. assess whether the model still predicts and accounts for higher state durations in monkey MCC pseudo-populations, when using model pseudo-population activity.

We found that, although the MCC/LPFC ratio is decreased when switching to pseudopopulation activity, state duration is still higher in the MCC, compared to the LPFC, with a ratio of ~1.5. In order to better understand the origin of this difference, we performed controls, where we randomly shuffled spikes according to the two orthogonal dimensions of spiking data, i.e. neurons and time, separately from one another. Shuffling spikes by randomly reattributing each individual spike of each neuron to a randomly chosen neuron (neuron shuffle control, repeated 10 times) also gave a similar ratio of ~1.92, showing that the neural identity of spikes did not account for the MCC/LPFC state duration ratio. However, randomly shuffling the timing of individual spikes within each neuron’s activity, or the time shuffle control, led to a ratio of ~1. The combination of both results indicated the MCC/LPFC difference originated from the global temporal structure of the collective network activity (rather than per-neuron temporal structure). To ensure the global temporal structure truly corresponded to spike timing rather than other trivial confounding variables, we controlled for variables we thought could affect the global temporal structure other than spike timing, e.g. the number or the firing frequency of LPFC/MCC neurons. Using the same number of neurons of each type for both LPFC and MCC model pseudo-populations areas (those of the LPFC) gave a ratio of ~1.52, while removing randomly chosen spikes from LPFC activity to lower its mean firing frequency to a value similar to that of the MCC gave a ratio of ~2.03. These controls evidenced that the ~1.5 MCC/LPFC ratio emerged from, and represented, a robust trait of model LPFC and MCC activity temporal structure and not a side effect of possible confounding factors (frequency, neural discharge identity, group statistics). Furthermore, the MCC/LPFC difference is structurally stable, showing strong robustness even to control procedures which strongly degrade model data.

Altogether, these results indicate that, while information is degraded when pseudo-population is used, a part of the information of neural activity’s temporal structure is preserved, resulting in a similar – although decreased – relative state duration ratio between MCC and LPFC. A fortiori, this additional analysis strengthens our results, as it indicates that MCC to LPFC state duration ratios should be higher in real monkey neural populations than the 7.5 ratio found here in pseudo-populations activity (given the 30 vs 1.5 ratios obtained with population vs pseudo-population model activity, respectively). In conclusion, these results indicate that using pseudo-populations in experimental data vs populations in the model is not an issue in drawing conclusions on the fact that neural states display relatively larger durations in the MCC, compared to the LPFC, in both monkey and model activity.

We have now added these extended results as supplementary materials (Figure 7 —figure supplement 1a) to better support our conclusions. We also explicitly mention in the Results that HMM are applied to model populations and monkey pseudo-populations (“Metastable states underlie LPFC and MCC temporal signatures” subsection), and point to specific explanations of this question in Materials and methods (“Hidden Markov Model (HMM) analysis” subsection) and Supplementary Materials (Figure 7 —figure supplement 1). We also mention this issue in the Discussion (“Frontal temporal signatures uncover metastable dynamics” subsection).

Secondly, the experimental data is compared to the network model which is not performing any specific task (i.e. without temporal structure). The authors suggest this may be the reason why their predictions for the state durations (Fig7b) are roughly an order of magnitude out. Presumably, the authors could consider designing a network model which could perform the same task (or a simplified version with a similar temporal structure) as the subjects perform. This would be very helpful in helping to relate the experimental data to the model, and may also provide a better understanding of the functional importance of the metastability they have identified in behaviour.

Building a biophysical neural network realizing the monkey behavioral task, even employing engineer-based learning plasticity rules, is out of scope of the present study. However, to address the concern raised, we devised a specific version of the HMM analysis to assess the possible role of the temporal structure of the task in the difference observed in the MCC compared to the LPFC in experimental data. Specifically, we simulated neural spiking activity of each neuron in LPFC and MCC areas as a non-homogeneous Poisson model (NHPM) whose instantaneous firing rate was given by the post-stimulus time histogram (PSTH) of that neuron upon each epoch type across all possible task events (e.g., reward, go signal, etc.). This simulated activity was only dependent on the coding of task variable averaged across trials, and did not contain more specific fine-grained activity temporal structure within each trial. Applying HMM analysis to such transformed population activity showed no difference of state durations between LPFC and MCC. This indicated that task variable coding was not responsible for the state duration difference between LPFC and MCC but relied on the precise temporal structure of spiking, independent of task epochs. This also indicated that accounting for neural states in monkeys with a biophysical model devoid of task structure was not an issue *per se*. We have now added this result as Supplementary Materials (Figure 7 —figure supplement 1b) to show that task coding is not essential in understanding the difference in state duration between MCC and LPFC in monkeys and can be accounted for by a model devoid of task temporal structure.

7. It is not clear to me how many neurons the authors included in their dataset, as there appear to be inconsistencies throughout the manuscript (Line 73, Figure 1A-b: MCC = 140, LPFC = 159; line 97-98: MCC = 294, LPFC=276; Figure 2: MCC = 266, LPFC = 258; Methods section line 734-735 and Fig2S2: MCC = 298, LPFC = 272). While this is likely a combination of typos and excluding some neurons from certain analyses, this will need to be resolved. It will be important for the authors to check their analyses, and also add a bit more clarity in the text as to which neurons are being included/excluded in each analysis, and justify this.

We thank the reviewer for this remark, as the reported numbers were indeed confusing. This is because neurons might not meet the criteria used to ensure that analyses contained enough data when performing specific tests, like limiting the extraction of Taus and latencies on specific behavioral periods. We have checked every sample size mentioned in the text and tried to better explain why and when those changes occurred.

Reviewer #2:The paper investigates the temporal signatures of single-neuron activity (the autocorrelation timescale and latency) in two frontal areas, MCC and LPFC. These signatures differ between the two areas and cell classes, and form an anatomical gradient in MCC. Moreover, the intrinsic timescales of single neurons correspond with their coding of behaviorally relevant information on different timescales. The authors develop a detailed biophysical network model which suggests that after-hyperpolarization potassium and inhibitory GABA-B conductances may underpin the potential biophysical mechanism that explains diverse temporal signatures observed in the data. The results appear exciting, as the proposed relationship between the intrinsic timescales, coding of behavioral timescales, and anatomical properties (e.g., the amount of local inhibition) in the two frontal areas is novel. The use of the biophysically detailed model is creative and interesting. However, there are serious methodological concerns undermining the key conclusions of this study, which need to be addressed before the results can be credited.

We thank the reviewer for their positive appreciation of our study. We address below all the methodological concerns.

1. One of the key findings is the correspondence between the intrinsic timescales of single neurons and their coding of information on different behavioral timescales (Figure 4). However, the method for estimating the intrinsic timescales has serious problems which can undermine the finding.1.1. The authors developed a new method for estimating autocorrelograms from spike data but the details of this method are not specified. It is stated that the method computes the distribution of inter-spike-intervals (ISIs) up to order 100, which was "normalized", but how it was normalized is not described. The correct normalization is crucial, as it converts the counts of spike coincidences (ISI distribution) into autocorrelogram (where the coincidence counts expected by chance are subtracted) and can produce artifacts if not performed correctly.

We thank the reviewer for pointing out the lack of clarity. We now make a more explicit and detailed explanation of the normalization procedures and the methods for spike autocorrelation in general.

We normalized by calculating the probability distribution function. This doesn’t affect the value of TAU exponentially fitted. In any case, there is a confusion between our measure (spike autocorrelation, calculated on ISIs) and auto-correlation coefficient (calculated across trials). The latter is the classical definition used by (Murray et al., 2014) and requires a specific normalization, i.e. subtracting the mean (across trials) and dividing by the standard deviation (across trials). Our method is based on the classic spike autocorrelogram computing the distribution of lags from each spike and the surrounding emitted spikes.

To clarify the approach, the methods section has been rewritten. It now reads: "To do so, we computed the lagged differences between spike times up to the 100th order, i.e. the time differences between any spike and its n successors (up to n = 100) at the unit level. The lagged differences were then sorted in \Δ_t=3.33ms bins from 0 to 1000ms. Given N the count of lagged differences within bins, we compute the probability distribution function as N/(\sum(N)*\Δ_t). …" in the Spike Autocorrelogram Analysis section of Methods.

1.2. The new method, described as superior to the previous method by Murray et al., 2014, appears to have access to more spikes than the Murray's method (Figure 2). Where is this additional data coming from? While Murray's method was applied to the pre-cue period, the time epoch used for the analysis with the new method is not stated clearly. It seems that the new method was applied to the data through the entire trial duration and across all trials, hence more spikes were available. If so, then changes in firing rates related to behavioral events contribute to the autocorrelation, if not appropriately removed. For example, the Murray's method subtracts trail-averaged activity (PSTH) from spike-counts, similar to shuffle-correction methods. If a similar correction was not part of the new method, then changes in firing rates due to coding of task variables will appear in the autocorrelogram and estimated timescales. This is a serious confound for interpretation of the results in Figure 4. For example, if the firing rate of a neuron varies slowly coding for the gauge size across trials, this will appear as a slow timescale if the autocorrelogram was not corrected to remove these rate changes. In this case, the timescale and GLM are just different metrics for the same rate changes, and the correspondence between them is expected. Before results in Figure 4 can be interpreted, details of the method need to be provided to make sure that the method measures intrinsic timescales, and not timescales of rate changes triggered by the task events. This is an important concern also because recent work showed that there is no correlation between task dependent and intrinsic timescales of single neurons, including in cingulate cortex and PFC (Spitmaan et al., PNAS, 2020).

The reviewer’s comments are very important and suggest that we did not explain the method clearly enough. In the new version we have tried to be more explicit.

In short, the method is based on the distribution of spike emitted after each spike, what has been called in the neurophysiological literature *spike autocorrelogram*. As explained in the method and main text, the spike autocorrelogram allows extraction of 2 temporal signatures: the latency at the peak, and the decay (as measured by an exponential fit). Author response image 3 illustrates these two measures for 1 example unit:

**Author response image 3. sa2fig3:** 

Contrary to the Murray method it is not based on the autocorrelation of binned spike counts measured on a particular time epoch and across trials. As the reviewer understood, the method requires a spike time timeseries sufficiently long to extract a non noisy spike autocorrelogram. It does not rely on a trial structure imposed by a task. One can measure the temporal signature on data recorded at rest. Importantly, and in contrast to studies using the Murray method, our analyses allowed us to include the large majority of units in our database, and even to seperate by unit types (FS and RS). Note that since in many studies the success rates of fitting an exponential were less than 55%, it followed that about half of the recorded data were excluded from analyses. We think that it could lead to serious sampling biases if for example one dissociates unit types as we did in this study.The reviewer is concerned about a potential relationship between task-induced variations of firing rate and the timescale measured on the spike autocorrelogram, and that this might explain the link observed between encoding of task variables and TAU median-splitted populations of units. As explained above (response to editor) we have made several controls to answer this concern.

We first asked whether TAU and firing average rate are related across units. Because there is no absolute method to measure firing rate, we tested measures for several periods of data (whole recording, feedback epoch, etc..). We found little if any correlation.

In addition, we would like to emphasize the following:

– In Stoll et al. we observed using *glmm* on spike counts that the gauge size was not encoded continuously during the task (i.e. across trials and during trials of the main task), but only at the time of decisions (see figure 5 in Stoll et al. Nat Comm. 2016). Thus gauge-related changes in firing rates were not structured in a way that it would impact the spike autocorrelograms.

– The difference in time scales between long and short timescale populations are of the order of a few hundreds milliseconds. In contrast the trial structure of the task, the gauge changes and the feedback deliveries, evoked by the reviewer is of the order of several tens of seconds. The two are thus incompatible and it is unlikely that task-related fluctuations explain differences in timescales measures with our methods.

Besides, as mentioned in our response to Reviewer 1 comment 6, we devised a specific version of the HMM analysis to assess the possible role of the temporal structure of the task in the difference observed in the MCC compared to the LPFC in experimental data (see Figure 7 —figure supplement 1). Specifically, we simulated neural spiking activity of each neuron in LPFC and MCC areas as a non-homogeneous Poisson model (NHPM) whose instantaneous firing rate was given by the post-stimulus time histogram (PSTH) of that neuron upon each epoch type across all possible task events (e.g., reward, go signal, etc.). This simulated activity was only dependent on the coding of taskvariable across trials but did not (as opposed to in original data) contain the specific fine-grained activity temporal structure within individual trials (i.e. at the level of individual ISIs) that is internally generated by frontal areas. Applying HMM analysis to such transformed population activity showed that the ratio between LPFC and MCC state duration distributions was lost. This indicated that task variable coding was not responsible for the difference of temporal structure between LPFC and MCC spiking activity, but that it relied on the precise temporal structure of spiking, independent of task epochs. So it is unlikely that the slow timescales arise from some slow firing rate variation due to the structure of the task. This makes sense, as the duration of task epochs was statistically similar when recording in the LPFC and MCC (i.e. the monkey performed the same task).

Regarding the Spitmaan et al. study, one must note that this study, like all other preceding investigations of timescales, could not take into account cell type differences which, according to our results, are a key element discriminating between task dependent properties and temporal signatures. This element put aside, our analyses did not directly test a correlation between estimated behavioral timescales and estimated intrinsic timescales. We simply observed the functional dissociation in cell type and functional properties on median split cell populations. To follow the reviewer’s concern we now mention this in the discussion:

“Spiking timescales distributions have been related to persistent activity, choice value and reward history in the LPFC and MCC (Bernacchia et al. Nat. Neurosci, 2011; Cavanagh et al., Nature Comm 2018; Meder et al., Nature Comm 2017; Wasmuht et al., Nat Comm. 2018), but in a recent study no correlation was observed between task dependent and intrinsic timescales at the unit level (Spitmaan et al., 2020). In all those studies, cell types were not considered. Here, we could not estimate taskrelevant timescales to correlate with TAU, but we found that the spiking timescales of MCC RS units increased on average during periods of engagement in cognitive performance”.

2. The balanced network model with a variety of biophysical currents is interesting and it is impressive that the model reproduces the autocorrelation signatures in the data. However, we need to better understand the network mechanism by which the model operates.2.1. The classical balanced network (without biophysical currents such as after-hyperpolarization potassium) generates asynchronous activity without temporal correlations (Renart et al., Science, 2010). The balanced networks with slow adaptation currents can generate persistent Up and Down states that produce correlations on slow timescales (Jercog et al., eLife, 2017). Since slow after-polarization potassium current was identified as a key ingredient, is the mechanism in the model similar to the one generating Up and Down states, or is it different? Although the biophysical ingredients necessary to match the data were identified, the network mechanism has not been studied. Describing this network mechanism and presenting the model in the context of existing literature is necessary, otherwise the results are difficult to interpret for the reader.

In the present study, the network model displays collective dynamics that peregrinate between distinct metastable states, each of which implying (and captured by the HMM as) a subset of strongly active neurons. At any time, as consecutive states occur, activity is present in the network, thus being permanent, i.e. with no interruption or ‘DOWN’ state. Such dynamics therefore strongly depart from UP and DOWN dynamics in a recurrent neural network (Jercog et al., *eLife*, 2017), in which all neurons are either all active or all silent. Moreover, while our network model is aimed at reproducing collective dynamics in frontal areas of awake monkeys, that of Jercog et al. 2017 is designed to capture spiking in anesthetized rodent cortical networks. Engel et al., 2016 have shown that primate visual networks can switch between two states that correspond to ON and OFF attentional states of awake monkeys. These states may resemble UP and DOWN states in other preparations/animals/areas. However, they differ from UP and DOWN states in that OFF state activity corresponded to a weak but non-null activity, whereas OFF states in anesthetized animals or in vitro are defined by the absence of any spiking. Therefore, data from Engel et al. rather likely correspond to switching between two metastable states fulfilling specific computations required by a binary attentional demand. Besides, the main parameters setting the timescale and duration of states in the present study are actually the maximal conductance of GABA-B receptors and the heterogeneity of synaptic weights. By contrast, the AHP maximal conductance is essential in setting the latency of the autocorrelogram, the other dimension of the temporal signature (as well as limiting the amount of fast transitions). AHP conductances have been implied in many possible computational functions such as network decorrelation (Renart et al., Science, 2010), the complexity of network dynamics (Cartling, International Journal of Neural Systems, 1996) or the linearization of the input-output function of neurons (Wang, J Neurophysiol, 1998). While possibly arising in different functional contexts or cooperating together in cortical networks, such diversity of AHP-dependent computational functions may emerge from differences in gating dynamics of AHP currents. Here, for instance, we consider fast AHP (20 ms time constant), while Jercog et al., 2017 consider slow AHP conductance (500 ms). We now clarify these elements in the Discussion (“Frontal temporal signatures uncover metastable dynamics” subsection).

2.2. Does the model operate in a physiologically relevant regime where the firing rates, Fano factor etc. are similar to the data? It is hard to judge from Figure 5b and needs to be quantified.

We agree that this comparison was lacking. We now provide (see Table 2) a quantitative evaluation of whether spiking statistics is similar in monkeys and model data in LPFC and MCC areas. We find that in both cases spiking frequency is below 5 Hz, CV above 1, and CV2 and LV (i.e. local (in the temporal domain) measures of ISI variability) and Fano factors around 1. An exact quantitative match between model and empirical was not sought for being (1) extremely complex to achieve, given the large amount of other essential constraints applied to the model (e.g. connective patterns, AMPA/NMDA ratios, E/I balance, temporal timescales and latencies of both excitatory and inhibitory neurons) and (2) not crucial, as the exact value of firing rates (and second order spiking moments) is not a determinant of timescales (see Figure 7 —figure supplement 1) but a function of biophysical parameters (GABA-B, AHP). In this context, the general quantitative consistency between spiking dynamics in model and monkey data indicates that our recurrent networks qualitatively fit with physiological statistics of frontal areas in the awake regime.

2.3. The latency of autocorrelation is an interesting feature in the data. Since the model replicates this feature (which is not intuitive), it is important to know what mechanism in the model generates autocorrelation latency.

In the model, we find that the principal mechanism responsible for the latency of the autocorrelation is the maximal conductance of the AHP current in excitatory neurons. Indeed, in these neurons, AHP currents are spike-evoked and their presence increases refractoriness of neurons, i.e. first-order latency, which is a major contributor of latency. This was detailed in supplementary material but has now also been stated in the Discussion (“Frontal temporal signatures uncover metastable dynamics” subsection).

3. HMM analysis is used to demonstrate metastability in the model and data, but there are some technical concerns that can undermine these conclusions.3.1. HMM with 4 states was fitted to the data and model. The ability to fit a four-state HMM to the data does not prove the existence of metastable states. HMM assumes a constant firing rate in each "state", and any deviation from this assumption is modeled as state transitions. For example, if some neurons gradually increase/decrease their firing rates over time, then HMM would generate a sequence of states with progressively higher/lower firing rates to capture this ramping activity.

This is a interesting point and we thank reviewer #2 for this remark. Network neural activity in behaving monkeys displays a dynamical complexity that exceeds that of HMMs, with non-stationary, continuous variations of activity operating at many timescales. Thus, analyzing neural data with HMMs therefore implies a form of simplification due to the representation of non-stationary continuous activity dynamics into stationary discrete states. However, some evidence indicates that HMMs constitute robust models (compared to PCA and PSTH analysis, e.g.), when applied to transient (i.e. non stationary) forms of coding (Jones et al., PNAS, 2007). Actually, HMMs have long been applied in frontal areas in contexts where neural activity is not stationary but evolves with ongoing task constraints and cognitive demands (perceptual events, decision processes, actions, e.g.; La Camera et al., Curr Opin Neurobiol 2019). These studies have evidenced that although non-stationary in essence, neural activity switches between well-separated states of activity – where it dwells most of the time –, within which firing rates are approximately stationary. A large corpus of theoretical studies has shown that states may correspond to attractors such as stable activity state within strongly connected Hebbian assemblies (Hopfield, PNAS, 1982; Brunel et Wang, Journal of Computational Neuroscience 2001; LaCamera et al., Curr Opin Neurobiol 2019). It has also largely been demonstrated that in highdimensional chaotic dynamics (as found in awake cortical recurrent networks), these attractors are not stable in all directions (i.e. they form “pseudo-attractors” or “attractor ruins”), which accounts for transitions between (metastable or quasi-stationary) states (see chaotic itinerancy (Tsuda, Behav Brain Sci 2001), dynamics along heteroclinic channels (unstable directions; Rabinovich et al. Science + PloS CB, 2008) or noise transitions between connectivity-related attractors (La Camera et al., Curr Opin Neurobiol 2019 or Miller, Curr Opin Neurobiol, 2016, e.g.)). In this context, HMM are interesting, as they allow highlighting the property of residing in preferential regions (e.g. Hebbian attractor) (as opposed to erratic wandering with uniform probability) in the neural state-space. Also, they are of particular precision in the temporal dimension, compared to traditional methods (Seidemann et al., J Neurosc 1996; Jones et al., PNAS, 2007).

Besides, we understand that systematic drifts in neuron’s firing frequencies at large time scales may bias HMMs towards modeling drifts as state transitions or bias the relative importance (duration) of specific states at different times. This bias might be important in the monkey recorded data HMM analyses (the neural frequency in our model networks being stationary by design). To specifically answer the issue and disambiguate the possibility that the observed states are reflecting drifts in data frequency, we assessed whether average neural frequency drift and average HMM state durations are correlated (Figure 7 —figure supplement 2). To do so, we repeated the HMM analysis on temporal segments of 50s of the 600s monkey data (i.e. the HMM was performed on spike data from 0-50s, 50100s, etc. all the way up to 550-600s). Moreover, this procedure was repeated for different segment durations (i.e. segments of 50, 60, 75s, 100, 120, 150, 200 300, 600s), to assess many different time scales over which neural frequency drifts might strongly bias HMM state durations, if at all. In each case, the average neural frequency drift was measured as the absolute difference in neural frequency between the first and second half of the data segment (e.g. 0-25 vs 25-50s for first 50s segment), averaged across all neurons.

We found that average neural frequency drifts were correlated to average HMM state durations within each cortical area when pooling all segment durations (LPFC: rho ~0.54, p-val ~9.9*10-5; MCC: rho ~0.38, p-val ~8.3*10-3; only a few correlations survived when looking at individual segment durations; LPFC, 60s: rho ~-0.68, p-val ~4.3*10-2; LPFC, 100s: rho ~0.85, p-val ~3.3*10-2; MCC, 120s: rho ~0.95, p-val ~1.3*10-2). However, state duration and frequency drift were never correlated when pooling data from both cortical areas, be it within or across temporal subdivisions. This was because the drifts in frequency were equivalent in the LPFC and MCC, whereas state durations were much longer in the MCC, compared to the LPFC. Thus, this indicates that average neural frequency drift does not account for the observed difference in state durations between LPFC and MCC.

This conclusion was confirmed by additional analysis of the relation between frequency drift and state durations on the 600s segment. Dividing neurons into two halves – “most” or “least” drifting – according to the amplitude of their frequency drifts across time (calculated similarly as before for each neuron on data from 0-600s), we then analyzed the spiking activity of each group via HMM. The frequency drift was ~6.6x higher (in LPFC and MCC) in the “most drifting” neurons, compared to “least drifting” neurons, indicating a strong difference in frequency drift between both groups across areas. However, the HMM state duration only increased by ~1.75x in “most drifting” vs. “least drifting” neurons across areas (~1.7x for LPFC, 1.8x for MCC). In contrast, the ratio of MCC vs. LPFC average state duration across neuron groups was much larger than 1.75x, being instead around ~5.75x (~5.5x for “least drifting” and ~6x for “most drifting” neurons). This confirms that neural frequency drift was not a major cause in the difference between LPFC and MCC average state durations, similarly to the previous analysis. These analyses are now provided in Figure 7 —figure supplement 2 and Table 3 in the article.

In addition, metastability implies exponential distributions of state durations, which was not verified.

We have now verified that distributions of state durations are exponential. To do so, we performed linear regressions on the logarithm of the count probability (binned in 50ms bins) of HMM state durations for each of the 100 HMM analyses, and reported the slope coefficient p-value and R2, for both cortical areas in both monkey and model data. We observed overall that R2 scores were above ~0.6, with significantly non-zero slopes (p<0.05) in all monkey HMM analyses, in all MCCm models and in the immense majority of PFCm models (98/100). Thus, state duration distributions appear exponential, lending credence to the metastable nature of HMM states. We have now added this analysis as Figure 7 —figure supplement 3.

No model selection was performed to determine the necessary number of states. Therefore, the claims of metastable dynamics are not supported by the presented analysis.

We understand this criticism and have now performed model selection to determine the necessary number of states. To do so we evaluated Akaike Information Criterion (AIC) and Bayesian Information Criterion (BIC) values for different numbers of states in both cortical areas and in both monkey and model data. We observed that AIC and BIC did not substantially change when varying the number of states in the range 2-10. Other methods for determining the number of states were explored (pseudoresidual error analysis, visual analysis and autocorrelation of HMM-generated spike rasters, and analysis of the structure of individual states) and none indicated a strong preference for any given number of states (not shown). More importantly, even when changing the number of states, HMM state duration results did not substantially differ, compared to those with 4 states. While LPFC and MCC state durations slightly decreased with more states (since the same time period is divided into a greater number of states), MCC state durations remained 1-2 orders of magnitude longer than LPFC state durations. Thus, choosing a number of states in the range 2-10 was equivalent according to this analysis. Since (1) a low number of states is more parsimonious in terms of data interpretation (Pohle et al., 2017) in general, (2) the task structure contains a low number of possible states in terms of actions (four), reward on the last trial (incorrect trial, first correct trial, correct trial after previous correct trials) and behavioral states (exploration, exploitation), and (3) increasing the number of HMM states substantially increased computational time of HMMs, we kept the low number of states chosen, in this instance 4 states. We have now added this analysis as Figure 7 —figure supplement 4.

3.2. HMM was fit to a continuous segment of data lasting 600s, and the data was pooled across different recording sessions. However, different sessions have potentially different trial sequences due to the flexibility of the task. How were different trial types matched across the sessions? If trial-types were not matched/aligned in time, then the states inferred by the HMM may trivially reflect a concatenation of different trial types in different sessions. For example, the same time point can correspond to the gauge onset in one session and to the work trial in another session, and vice versa at a different time. If some neurons respond to the gauge and others to the work, then the HMM would need different states to capture firing patterns arising solely from concatenating the neural responses in this way. This confound needs to be addressed before the results can be interpreted.We previously showed that the temporal structure of the task had no effect on the LPFC/MCC state duration difference (see Reviewer #1, major point #6), and that pseudo-populations preserve global spike-timing structure (rather than spike-timing structure within individual recording sessions). As such, trial types do not affect HMM state duration, and sessions were thus aggregated without any trial-type alignment.

We have now assessed this issue by two complementary analyses.

First, we have showed that the temporal structure of the task had no effect on the LPFC/MCC state duration difference (Figure 7 – supplement 1b). To do so, we have simulated neural spiking activity of each neuron in LPFC and MCC areas as a non-homogeneous Poisson model (NHPM) with instantaneous firing rate given by its post-stimulus time histogram (PSTH) during behavioral epoch type between all possible task events (e.g., reward, go signal, etc.). This simulated activity only depended on task-variable coding and did not contain more specific fine-grained spiking temporal structure. Applying HMM analysis to this surrogate activity showed no difference of state durations between LPFC and MCC. This indicated that task variable coding was not responsible for the state duration difference between LPFC and MCC but relied on the precise temporal structure of spiking, independent of task epochs. This also indicated that accounting for neural states in monkeys with a biophysical model devoid of task structure was not an issue per se.

Besides, we have also now applied HMM analysis to neural activity in the model, using pseudopopulations built of groups of neurons drawn from LPFC and MCC simulations of distinct random neural networks, with group statistics (number of recording sessions and numbers of simultaneously recorded excitatory and inhibitory neurons within each recording session) similar to those of monkey LPFC and MCC data (Figure 7 – supplement 1a) and showed that pseudo-populations gathered from different sessions preserve global spike-timing structure (rather than spike-timing structure within individual recording sessions).

A more detailed presentation of these supplementary analyses can be found in response to reviewer 1 (major point 6).

Altogether, these new analyses indicate that trial types do not affect HMM state duration, and that sessions can thus be aggregated without any trial-type alignment.